# Integrated in vivo combinatorial functional genomics and spatial transcriptomics of tumours to decode genotype-to-phenotype relationships

Marco Breinig [1,12] ✉, Artem Lomakin[2,11,12], Elyas Heidari[2,3,12], Michael Ritter[4,5], Gleb Rukhovich[2], Lio Böse[1], Luise Butthof[1], Lena Wendler-Link[1], Hendrik Wiethoff [1], Tanja Poth[6], Felix Sahm [4,5], Peter Schirmacher[1], Oliver Stegle[3], Moritz Gerstung [2,7,8,9] ✉ & Darjus F. Tschaharganeh[1,10] ✉

Advancing spatially resolved in vivo functional genomes will link complex genetic alterations prevalent in cancer to critical disease phenotypes within tumour ecosystems. To this end, we developed PERTURB-CAST, a method to streamline the identification of perturbations at the tissue level. By adapting RNA-templated ligation probes, PERTURB-CAST leverages commercial 10X Visium spatial transcriptomics to integrate perturbation mapping with transcriptome-wide phenotyping in the same tissue section using a widely available single-readout platform. In addition, we present CHOCOLAT-G2P, a scalable framework designed to study higher-order combinatorial perturbations that mimic tumour heterogeneity. We apply it to investigate tissue-level phenotypic effects of combinatorial perturbations that induce autochthonous mosaic liver tumours.

Cancer, like many other complex diseases, is caused by a combination of multiple genetic alterations[1,2]. Transitioning from portraying these genetic changes to understanding their phenotypic consequences by comparing human samples is, however, constrained by environmental influences, genetic diversity between patients, pervasive epistasis and the complexity of multicellular tissue structure[3–5]. Consequently, it remains poorly understood how combinations of alterations repro-gramme cells and their interactions with the tissue environment.

Genetic screens conducted in model systems have proven valuable for decoding genotype–phenotype relationships in controlled settings[6,7] However, the presently available approaches for cancer-relevant in vivo functional genomics mostly investigate the effect of singular or pairwise alterations on proliferation and tumorigenesis without considering spatial niches in which cancer cells competitively develop and grow[8–12]. Although emerging studies are beginning to explore the impact of individual tumour alterations on their immunological microenvironments[13–15], gaps remain in our understanding of how genetic changes jointly rewire tumour cells and their surrounding cellular ecosystems through epistasis. Overcoming this challenge calls for experimental approaches and modelling

[1]Institute of Pathology, University Hospital Heidelberg, Heidelberg, Germany. [2]Artificial Intelligence in Oncology, German Cancer Research Center (DKFZ), Heidelberg, Germany. [3]Computational Genomics and Systems Genetics, German Cancer Research Center (DKFZ), Heidelberg, Germany. [4]Department of Neuropathology, University Hospital Heidelberg, Heidelberg, Germany. [5]Clinical Cooperation Unit Neuropathology, German Consortium for Translational Cancer Research (DKTK), German Cancer Research Center (DKFZ), Heidelberg, Germany. [6]Center for Model System and Comparative Pathology, Institute of Pathology, University Hospital Heidelberg, Heidelberg, Germany. [7]Robert Bosch Center for Tumor Diseases, Stuttgart, Germany. [8]Medical Faculty, Eberhard Karls University, Tübingen, Germany. [9]Universty Hospital Tübingen, Tübingen, Germany. [10]'Cell Plasticity and Epigenetic Remodeling' Helmholtz-University Group, German Cancer Research Center (DKFZ), Heidelberg, Germany. [11]Present address: Stanford Cancer Institute, School of Medicine, Stanford University, Stanford, CA, USA. [12]These authors contributed equally: Marco Breinig, Artem Lomakin, Elyas Heidari. ✉e-mail: marco.breinig@gmail.com; moritz.gerstung@dkfz.de; d.tschaharganeh@dkfz.de

strategies that leverage tissue-level analyses and can effectively scale to handle higher-order combinations[3]. For instance, testing all combinations of four alterations in standard rodent models with four mice per group requires 64 animals. This need escalates rapidly: six perturbations would demand 256 and eight perturbations would require >1,000 animals.

Here, to meet this challenge, we introduce a scalable experimental framework designed to facilitate the functional exploration of complex genotype–phenotype associations at the tissue level, which we termed charting higher-order combinations leveraging analysis of tissue to investigate genotype-to-phenotype relationships (CHOCOLAT-G2P; hereafter referred to as C-G2P). C-G2P is based on a mouse model of autochthonous tumour development, where combinations of barcoded perturbation plasmids randomly integrate into cells within their native environment (Random Unique Barcode Integration Combinatorics, RUBIX). RUBIX thus generates mosaics of genetically heterogeneous tumour clones in a single tissue. To streamline C-G2P spatially resolved in vivo functional genomics, we further developed Perturbation Barcode Capture Spatial Transcriptomics (PERTURB-CAST), a method that seamlessly integrates perturbation mapping with the standardized and commercially available 10X Visium spatial transcriptomics platform. We applied C-G2P to investigate phenotypic effects of eight combinatorial perturbations that induce liver tumours sampled from 256 possible genotypes.

## Results

### A framework to spatially map engineered tumour heterogeneity

Human tumours frequently present combinations of genetic alterations. With the aim to link complex genetic alterations prevalent in cancer to critical disease phenotypes within tumour ecosystems, we developed C-G2P. C-G2P allows induction and mapping of combinatorial perturbations in murine tissue and simultaneous characterization of the resulting neoplastic phenotypes on the same sample using a single spatial transcriptomics readout platform. Therefore, C-G2P merges and advances available technologies, including multiplexed perturbation in vivo functional genomics, molecular barcoding and spatial omics[5,7,9,14,16] (Fig. 1).

Previous in vivo approaches to spatially map perturbations relevant to cancer employed ex vivo-manipulated cells that were subsequently injected into animals[14,15,17]. For C-G2P, we aimed to leverage an in vivo setting that more closely resembles tumour heterogeneity and tumorigenesis by direct genetic modification of cells embedded in their native tissue environment[7]. We therefore modified an autochthonous murine mosaic liver cancer model[18–20] to allow for the creation of coexisting genetically diverse tumours. This approach (RUBIX) relies on hydrodynamic-tail-vein (HDTV) injection of pooled molecular-barcoded plasmids and sleeping beauty transposon-based methods to stably integrate traceable higher-order combinations of genetic alterations in hepatocytes within their tissue context. Consequently, RUBIX offers the possibility to generate mosaics of genetically heterogeneous tumour clones in a single tissue (Fig. 1a(top) and Methods).

Presently available approaches to spatially map perturbations within tissue engage custom protocols and orthogonal readouts to also obtain transcriptomics-based phenotypic profiles, such as sequential antibody-based barcode detection and 10X Visium spatial transcriptomics on an additional tissue sample[14]. We developed PERTURB-CAST to address the constraints of existing methods and streamline the identification of perturbations and comprehensive tissue-level phenotypic information. PERTURB-CAST leverages spatial transcriptomics based on targeted transcript capture via RNA-templated ligation (RTL) probes[21] that are commercially available with 10X Visium for formalin-fixed paraffin-embedded (FFPE) samples. To detect the introduced perturbations, PERTURB-CAST

engages perturbation plasmids extended with 50-nucleotide (nt) barcodes amenable to RTL-probe capture (Fig. 1a(bottom)). Importantly, to ensure immediate compatibility with default commercial kits and circumvent modifications to the standard protocol, we redeployed 10X Visium RTL probes capturing chemosensory receptor transcripts that are not expressed in mouse liver for barcode identification. Specifically, we exploited 50-nt RTL-probe capture sequences of olfactory-, taste- and vomeronasal-receptor transcripts as molecular barcodes (Fig. 1b, Supplementary Fig. 1 and Methods). To achieve robustness, triplet barcode arrays were included in each perturbation construct to enable detection by three individual RTL probes that are pre-existing components of 10X Visium spatial transcriptomics kits (Fig. 1b, Extended Data Fig. 1, Supplementary Fig. 1 and Methods).

### RUBIX generates higher-order combinatorial perturbations

Related to the genetic complexity in human liver cancer, we found that approximately 30% of tumours simultaneously present seven established cancer-driving alterations (including gain- and loss-of-function mutations as well as somatic copy number alterations such as amplifications and heterozygous losses; Fig. 2a and Supplementary Fig. 2) that can reveal varying combinatorial patterns in individual tumour samples (Fig. 2b)[1]. For C-G2P proof-of-concept, we therefore modelled complex cancer genetics by not solely focusing on loss-of-function mutations and multiplex CRISPR knockouts that may induce chromosomal rearrangements and cellular toxicity due to multiple double-stranded breaks[8]. We instead concentrated on combinations of alterations associated with liver cancer, including overexpression of oncogenic drivers (*Myc*, mutant *Ctnnb1* (mtCtnnb1), *Vegfa* and NICD) and silencing of tumour suppressors (*Trp53*, *Pten* and *Kmt2c*) with short hairpin RNA (shRNA) alongside a frequently used *Renilla* luciferase (shRen)-targeting control construct[13,22]. We used RUBIX with a mix of eight barcoded perturbation plasmids for HTDV injection to generate a spectrum of combinatorial alterations relevant to liver cancer (Fig. 2b,c) and waited until tumours were palpable. Defined by combinations of these eight perturbations, we consequently anticipated testing $2^8 = 256$ possible cancer-driving genotypes in a single experiment (Fig. 1a). In our pilot experiments, we distributed a total of 38 redeployed barcodes amenable to RTL-probe capture across eight perturbation plasmids (including variations in position and promotors) and further included complementary barcodes (for example, peptides)[14,23] for orthogonal readouts (Extended Data Fig. 1 and Methods).

### PERTURB-CAST hijacks probe-based transcript capture for barcode mapping

C-G2P liver samples were collected ten weeks after HTDV injection and processed (FFPE; Extended Data Fig. 2a). For spatial transcriptomics, six topographically separated regions of interest were selected based on histopathological assessment of haematoxylin and eosin (H&E)-stained sections (Fig. 3a and Extended Data Fig. 2a–e). Both 10X Visium and 10X CytAssist were conducted for a total of 12 samples covering these six regions, including serial sections as technical replicates (Extended Data Fig. 2f–h). A total of 324 tumour nodules were identified across the six segregated sections (Fig. 3a). Notably, spatial transcriptomics helped distinguish overlapping nodules that seemed to be single lesions from the histopathological perspective (Supplementary Fig. 3).

Assessing the feasibility of PERTURB-CAST and our barcoding strategy, we observed that most barcode signals were readily detected in the C-G2P liver samples. Strikingly, barcode signals were spatially confined and closely tracking the areas of microscopic tumour nodules, as revealed by H&E staining (Fig. 3b). In contrast, we observed that none of the 38 redeployed barcode sequences were detected in publicly available murine liver 10X Visium datasets[24] (Fig. 3b(left)).

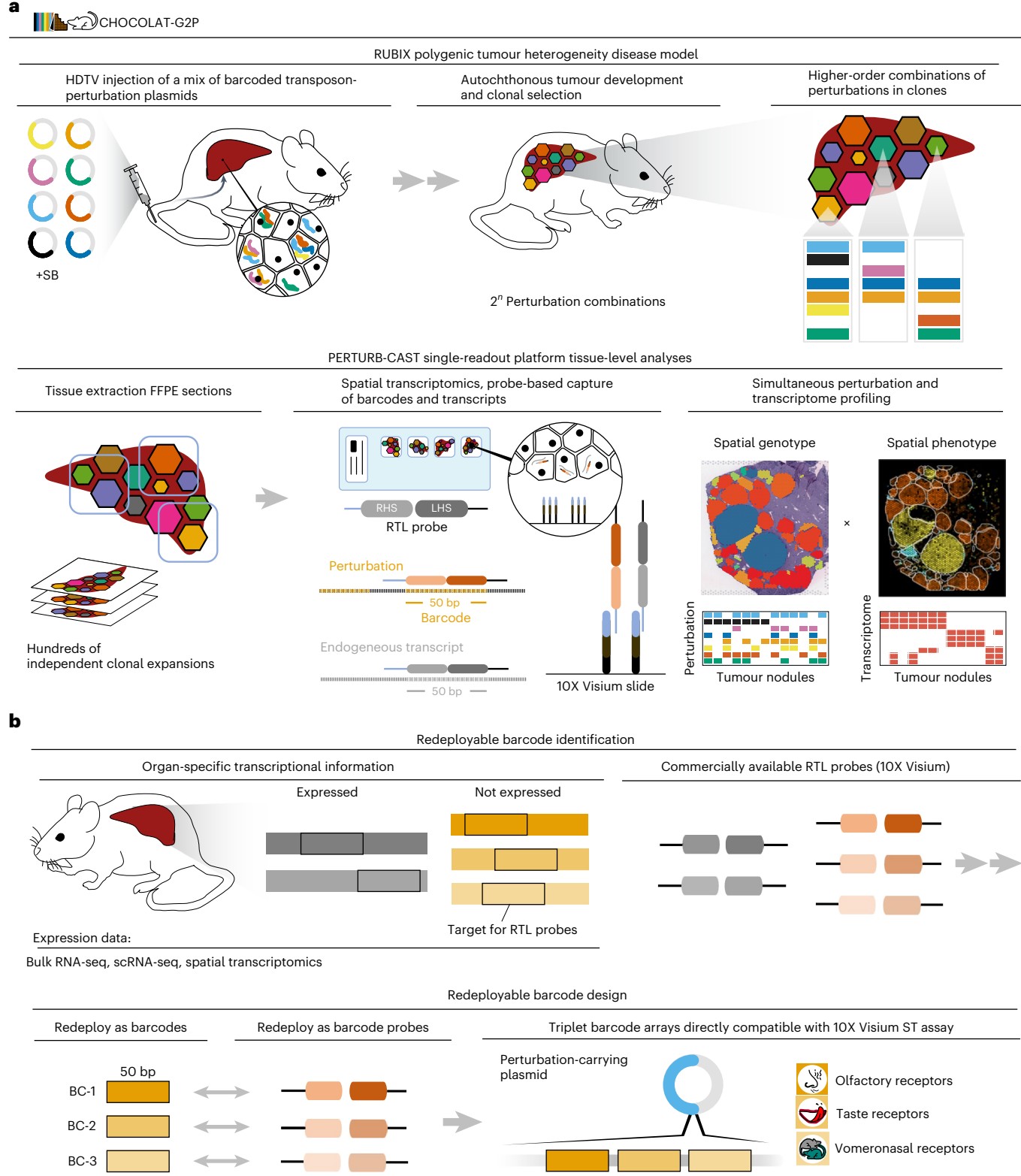

**Fig. 1 | A framework to spatially map engineered tumour heterogeneity.**
**a**, The C-G2P framework. Mice are HDTV injected with a pool of barcoded perturbation plasmids leading to sleeping beauty (SB)-transposon-mediated stable integration into the genome of hepatocytes. Higher-order combinatorial perturbations drive mosaic liver tumour development in a conceptual $2^n$ combination space for clonal selection (RUBIX). Direct barcode identification is achieved by linking perturbations to 50-nt barcode sequences that are captured and identified by RTL probes as embedded in the 10X Visium for FFPE platform (PERTURB-CAST). Endogenous transcripts are captured alongside barcodes, hence enabling simultaneous mapping of genotypes (as defined by the presence of perturbations) and phenotypes (as defined by transcriptional signatures) on the same tissue section. **b**, PERTURB-CAST barcode selection. Transcripts not expressed in murine liver are identified using public databases. Their respective 50-nt RTL-probe capture sequences are used as barcodes detected by redeployed commercially available RTL probes provided with the 10X Visium for FFPE mouse kit (Methods). Barcodes derived from chemosensory receptor transcripts are embedded in perturbation plasmids as triplet arrays.

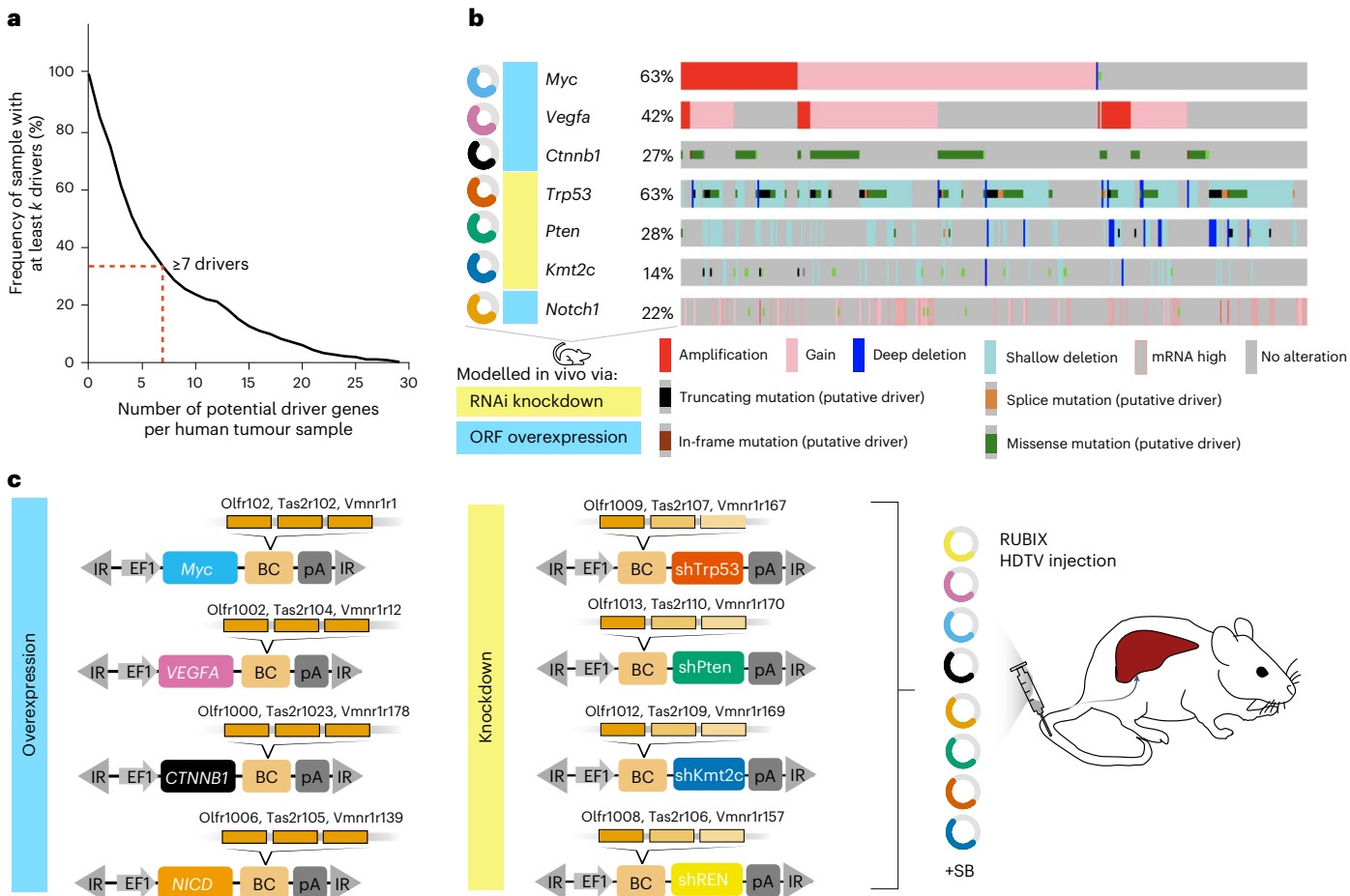

**Fig. 2 | Modelling tumour heterogeneity. a**, Frequency of liver tumour samples with at least *k* potential driver mutations per sample in The Cancer Genome Atlas Program (TCGA) HCC dataset. Potential drivers were defined as either amplification or fusion of known COSMIC oncogenes, or homozygous deletion, nonsense mutation, splice site mutation or frameshift deletion/insertion in tumour-suppressor genes. **b**, Frequent alterations observed in human liver cancer (The Cancer Genome Atlas Program HCC dataset) are 'geno-copied' in a C-G2P mouse model (oncoprint based on https://www.cbioportal.org/study/summary?id=lihc_tcga). ORF, open reading frame; RNAi, RNA interference. **c**, RUBIX mouse model generated in this study. Schematic overview of sleeping

beauty transposon perturbation plasmids to ectopically overexpress genes of interest (oncogenic-driver perturbations) or shRNA to enable gene knockdown (tumour-suppressor perturbations). Functional elements are highlighted. BC, barcode in which three redeployed RTL-probe capture sequences (as indicated) are embedded; EF1, polymerase II promoter; IR, inverted/direct repeats of sleeping beauty transposon; pA: polyadenylation signal; sh, shRNA embedded in miRE context. Note that we used Visium mouse transcriptome probe set v1 to derive barcodes. Each 50-nt barcode is separated and flanked by spacer sequences of approximately 20 nt to avoid potential steric hindrance during hybridization. Further information in Methods.

Reassuringly, with few exceptions (for example, *Olfr1033* and *Olfr1358*), chemosensory receptor transcripts that provided the repertoire for barcode redeployment (*n* = 1,216) were generally not detected by 10X Visium in murine livers (Extended Data Fig. 3a,b). Overall, 5/38 redeployed barcodes had insufficient signal across all samples investigated (log(1*p*)-transformed average expression, <0.05 counts per 1 × 10⁴; Extended Data Fig. 3). Notably, redeployed barcodes expressed using a Pol III promoter (hU6; Extended Data Fig. 3c) were detectable but we noticed a trend where detection became weaker as the barcode was positioned farther from the 5′ end (Extended Data Fig. 3c). Although detection strength of individual barcodes varied, barcode-triplets enabled us to spatially identify all eight perturbations (Fig. 3b and Extended Data Figs. 3,4).

**Spatial perturbation mapping across tissue**

Given the observed uncertainties related to individual barcode read-outs, we used a variational Bayesian model, which accounts for multiple sources of variability (including, for example, local 10X Visium spot sensitivity) to assign perturbations to each nodule (Fig. 3c, Extended Data Fig. 5 and Methods). Notably, nodule-level predictions correlated

across serial tissue sections and between 10X Visium and 10X CytAssist replicate experiments (Pearson's correlation coefficient (*r*) = 0.63–0.78 depending on perturbation; Extended Data Fig. 6), demonstrating the quantitative reproducibility of the approach.

Investigation of the expression levels of the genes targeted by each perturbation provides an orthogonal readout of inferred perturbation-plasmid integration. Accordingly, across lesions, a generalized linear model (GLM; Methods) confirmed the expected trends of overexpression or silencing based on the corresponding perturbation, including increased expression of *Notch1* and reduced expression of *Pten* (Fig. 3d) as well as *Kmt2c* and *Trp53* (Extended Data Fig. 7a,b). Further validation of mtCtnnb1 expression was achieved by a similar analysis of glutamine synthetase (GS) immunohistochemistry (IHC), which indicates hepatic WNT–Ctnnb1 signalling activity[25] (Fig. 3d). Last, the plasmids targeting *Trp53* and *Kmt2c* contained green and red fluorescent protein barcodes, respectively, providing additional IHC-based validation (Extended Data Fig. 7c–e).

Thus, the probe-based barcode capture of PERTURB-CAST spatially maps combinatorial perturbations within hundreds of coexisting tumours generated by RUBIX and provides a foundation

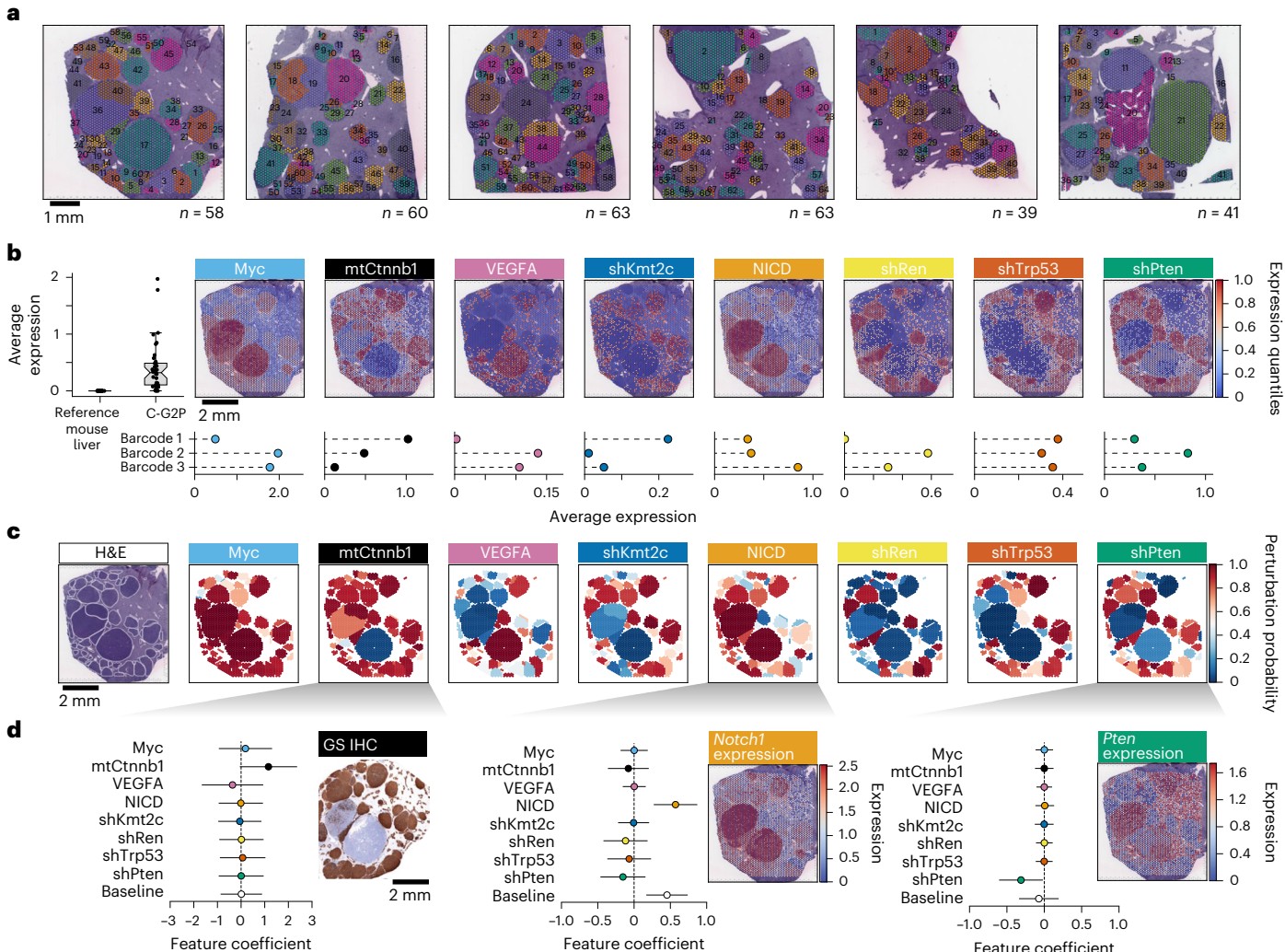

**Fig. 3 | PERTURB-CAST spatially resolves multiplexed genetic perturbations in hundreds of coexisting cancer clones. a**, RUBIX establishes hundreds of coexisting tumours in the context of native tissue. Respective H&E-stained tissue samples for six topographically separated regions (approximately 6 × 6 mm) that were used for 10X Visium for FFPE-spatial transcriptomics analysis. A total of 324 nodules (colour-coded and numbered) were annotated. Colours were chosen arbitrarily. **b**, PERTUB-CAST allows perturbation-specific barcode identification. Average log(1$p$)-transformed expression of all 38 barcode-associated transcripts used (left). Combined data for the reference control liver datasets from ref. 21 and the six main spatial transcriptomics samples (C-G2P). Spatially resolved expression of triplet barcodes (as indicated in Fig. 2c) for each of the eight perturbations (top right). Aggregated log(1$p$)-transformed and quantile-rescaled expression per 10X Visium spot. A representative sample is shown. Average log(1$p$)-transformed expression of individual barcodes in each triplet array for each perturbation averaged across the six spatial transcriptomics samples (bottom right). **c**, Conversion of PERTURB-CAST barcode signals to perturbation maps. Spatially resolved visualization of the inferred probabilities

indicating the presence or absence of each of the eight perturbations associated with annotated tumour nodules (Methods). A representative sample is displayed. **b,c**, Both the quantitative barcode expression (**b**) and probabilities (**c**) of all samples can be explored through the interactive web browser https://chocolat-g2p.dkfz.de/. **d**, Validation of inferred perturbation integration. A GLM model predicts the phenotype expression signals based on the estimated probabilities of perturbation presence using Bayesian modelling (Methods). Phenotypes are defined as direct target transcripts associated with perturbations such as shPten–*Pten* and NICD–*Notch1*. Expression data were log(1$p$)-transformed. GS, a well-established marker for active WNT signalling in murine livers, was used to infer mtCtnnb1-GS-positive phenotype via IHC on a corresponding serial section. Baseline depicts background phenotype marker expression. Data are presented as feature coefficients shown as mean and error bars depict 3σ confidence intervals (CIs). Data are derived from 324 nodules across six topographically separated regions used for 10X Visium from a single RUBIX experiment with two animals. Mapping GS IHC data are derived from three corresponding sections from a single RUBIX experiment with two animals. A representative sample section is displayed.

to comprehensively chart tumour genotypes (interactive maps at https://chocolat-g2p.dkfz.de/).

**Comparative analyses in hundreds of coexisting cancer clones**
Across the entirety of the 324 identified nodules, the Bayesian model calculates the probabilities for all $2^8$ = 256 possible genotypes defined by the combinations of eight perturbations, thereby converting spatial barcode signals into genotypically defined clonal maps (Fig. 4a where individual perturbation probabilities for one nodule are highlighted as an example and Extended Data Fig. 8).

We observed that tumour clones established in our initial C-G2P experiments typically exhibited combinatorial alterations with quintets being the most prevalent (approximately 30%; Fig. 4b,c). The absence of nodules with low integration numbers and the overall tendency towards multiple perturbations corroborates expected and previously described genetic interactions of oncogenes and tumour suppressors, defined simply by pairwise cooperation inferred from individual experiments[19,20]. Furthermore, alterations of well-recognized oncogenes, for example, Myc (82% of all nodules) and mtCtnnb1 (80%) occurred most frequently, indicating strong clonal selection, whereas

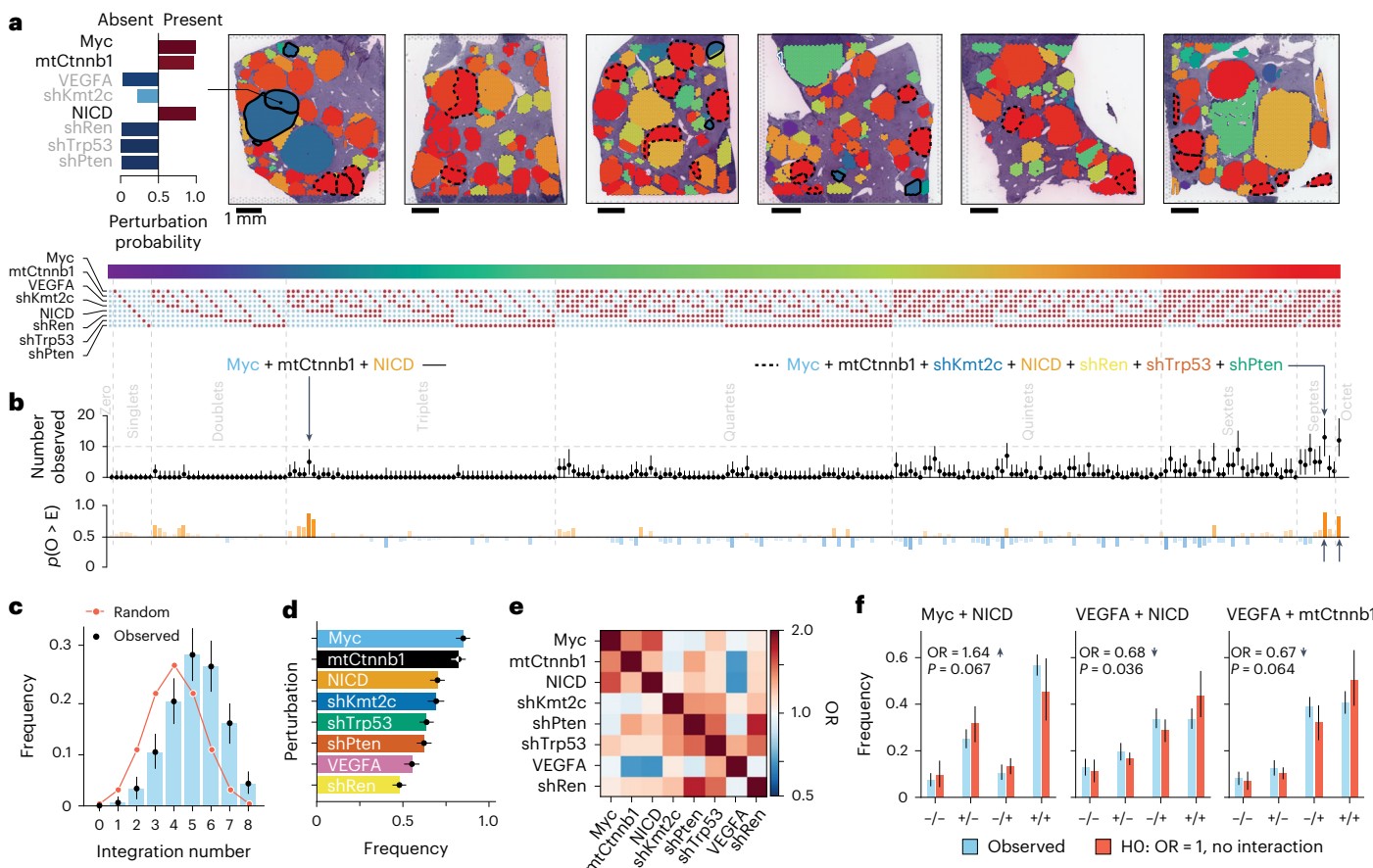

**Fig. 4 | C-G2P enables comparative genotype analyses and disentangles context-dependent genetic interactions. a**, Genotype maps (right); $2^8$ powerset embedding of spatially mapped perturbations encompassing 324 nodules across six topographically separated regions. Each of the 256 combinations is colour-coded. Perturbation probabilities for a representative nodule are depicted (left; black text highlights present perturbations, while grey text highlights absent perturbations). Nodules sharing representative similar genotypes are encircled and indicated in **b** (solid lines, Myc + mtCtnnb1 + NICD; dashed lines, Myc + mtCtnnb1 + shKmt2c + NICD + shRen + shTrp53 + shPten). **b**, Clonal selection. Observed occurrences of genotypically defined tumour clones (median and 95% CI) across $2^8$ powerset embedding (top). Grey text indicates combinatorial complexity. The highlighted genotypes are encircled in **a**. Probability $p(O > E)$ that observed occurrences (O) deviate from the expected baseline distribution (E) (Methods; bottom). Deviations of >0.5 indicate increased tumorigenic potential (orange), whereas values of <0.5 suggest potentially disadvantageous combinations (blue). **c**, Combinatorial order distribution. Observed distribution of the perturbation integration order (mean and 95% CI). A binomial distribution with $p = 0.5$ is included as a reference of a random unbiased integration rate (red line). **d**, Ranking of cancer-driving perturbations. Marginal frequencies of

individual perturbations in descending order (mean and 95% CI). **e**, Pairwise co-occurrence and mutual exclusivity patterns. An OR > 1 suggests co-occurrence, whereas OR < 1 indicates mutual exclusivity (Methods). Perturbations are ordered according to **d**. **f**, Identification of pairwise genetic interactions. Comparison of observed versus expected frequencies (median and 95% CI) for selected gene pairs, calculated using multiplicative models of gene interaction. We simulated the expected probabilities for the pairwise groups under the assumption of no interaction OR, which indicates the direction of the gene interaction effect (arrows), are reported along with the corresponding $P$ values. OR values were estimated from 5,000 posterior samples. A softmax GLM with interaction fixed at one defined the null. $P$ values reflect two-tailed deviations of observed double-positive proportions from the null based on 5,000 draws (Methods). Data are derived from 324 nodules across six topographically separated regions used for 10X Visium from a single RUBIX experiment with two animals. Bayesian modelling of perturbation probabilities was used to infer the occurrence of individual perturbation combinations per nodule. From the inferred Bayesian posterior, we sampled 5,000 points and computed the median and CI for the frequencies of individual perturbations as well as individual genotypes and calculated the OR (Methods). H0, null hypothesis.

VEGFA (53%) and shRen (46%) were less frequent, possibly reflecting low tumorigenic potential (Fig. 4d).

The frequency at which perturbations are observed across nodules is dependent on the rate of successful integrations and the neoplastic potential of the combinatorial perturbation. Assuming a fixed integration rate for each perturbation allows modelling of an expected distribution of combinatorial events and assessment of whether specific observed combinations are enriched, suggesting higher tumorigenic potential (orange), or depleted and thus indicating disadvantageous effects (blue) independent of technical influences (Fig. 4b(bottom) and Methods). Notably, among genotypes with fewer combinations, the triplet comprising Myc, mtCtnnb1 and NICD emerged as frequent ($n = 5$, $p(O > E) = 0.87$; solid line and arrow in Fig. 4a and b(top), respectively),

which suggests a strong association of this specific compound genotype with tumorigenesis. Interestingly, although septets seemed to be generally prevalent, the specific septet devoid of VEGFA (dashed line and arrow in Fig. 4a and b(top), respectively) demonstrated enrichment ($n = 13$, $p(O > E) = 0.90$) comparable to the complete octet ($n = 12$, $p(O > E) = 0.81$), whereas septets lacking mtCtnnb1 or Myc, for example, were less enriched. This in turn suggested a diminished cancer-driving effect of VEGFA in the setting of the combinatorial alterations tested.

## Cancer-driving co-dependencies and potential context dependencies

To pinpoint which perturbations contributed to the observed patterns of enriched and depleted genotypes, we conducted co-occurrence

odds ratio (OR) analysis. This analysis measures second-order epistatic interactions, which quantifies the deviations from purely additive effects in a commonly used multiplicative model[6,26] (Fig. 4e,f and Methods). Our results revealed co-dependency patterns for Myc, mtCtnnb1 and NICD as well as shTrp53 and shPten, aligning with previous observations[18–20]. The combination of Myc and NICD exhibited the most pronounced effect (OR = 1.64; $P$ = 0.067). In contrast, we observed a tendency towards mutual exclusivity between VEGFA and NICD (OR = 0.68; $P$ = 0.036) and between VEGFA and mtCtnnb1 (OR = 0.67; $P$ = 0.064; Fig. 4e,f).

Together, these observations indicate a context-dependent oncogenic effect of VEGFA.

### Phenotypic landscapes of engineered tumour heterogeneity

To elucidate spatial phenotypes and enable subsequent genotype-to-phenotype analyses (Fig. 5), we leveraged tissue-wide transcriptional signatures and defined sets of transcripts that characterized prevalent cell states (Supplementary Figs. 4 and 5 and Methods). To finally map the complexity of tumour ecosystems, we visualized phenotype-associated transcriptional signatures within their spatial context (Fig. 5a and Supplementary Figs. 6–18). Furthermore, to highlight associations for nodule-intrinsic phenotypes as well as those related to the tumour microenvironment (TME), we used transcript-resolved heatmap presentations (Fig. 5b–d). Thereby, C-G2P allowed us to chart the heterogeneous phenotypic landscape of hundreds of coexisting genotypically defined tumours and their surrounding tissue environment (interactive maps at: https://chocolat-g2p.dkfz.de/).

### Stratification of coexisting liver tumour subtypes

Our approach readily distinguished prominent subtypes of liver tumours (Fig. 5a(top)). First, we observed nodules with cholangiocyte-like transcriptional signatures (for example, $Krt19^+Cldn7^+$) indicative of cholangiocarcinoma (CCA)[27,28] (Fig. 5a,b and Supplementary Fig. 6). Microscopy inspection indeed classified these tumour nodules as CCAs exhibiting a glandular growth pattern and stroma deposition as well as CK19-protein expression by IHC (Supplementary Fig. 7). Cholangiocarcinoma is the second-most common type of liver cancer following hepatocellular carcinoma (HCC) and both tumour types can develop from hepatocytes in the HDTV injection-based mouse model[27,28]. Interestingly, C-G2P pinpointed, among others, expression of solute carrier family 15 member 2 (Slc15a2) for which genomic variants have been linked to sorafenib-therapy response[29] as well as pancreatic glycoprotein 2 (Gp2) as being associated with CCA (Fig. 5b and Supplementary Fig. 6). The latter observation harmonizes with earlier research suggesting that anti-GP2 IgA autoantibodies enable early CCA detection in subsets of human patients[30], hence indicating that our C-G2P approach captures key elements of CCA biology that have so far not been observed in animal models.

Moreover, given the spatial resolution of C-G2P, we could immediately relate the prominent second and third cluster of tumour nodules to metabolic liver zonation. Spatial division of metabolic functions is not only central to liver-tissue organization under physiological conditions[31,32] but has been proposed to enable molecular classification of human HCC[33–35]. In alignment with this zonation-based molecular classification, C-G2P enabled us to stratify nodules either as portal-like (for example, $Sds^+Sdsl^+$) or central-like (for example, $Cyp2e1^+Oat^+$)[31,32], the latter being the most abundant tumour class observed (Fig. 5a,b and Supplementary Figs. 8 and 9). Interestingly, recent findings from zonation fate-mapping animal models suggest liver cancer prevention strategies that leverage central-zonation-dependent mechanisms, particularly targeting Gstm3, which we also identified as a central-like tumour marker (Fig. 5b)[36].

Last, focusing on the tumours that could not readily be assigned to the aforementioned subtypes, we identified a fraction of nodules that revealed enrichment of hepcidin antimicrobial peptide (Hamp), Hamp2 and uridine phosphorylase 2 (Upp2) expression ($Hamp2^+Upp2^+$; Fig. 5a and Supplementary Fig. 10). Hamp and its paralogue Hamp2 have both been associated with midlobular zonation[32], a feature of liver structure important for regeneration[37]. Upp2, on the other hand, is involved in pyrimidine salvage, which fuels glycolysis and enables growth of cancer cells under nutrient-limited conditions[38,39].

We further identified subgroups of nodules sharing cholangiocytic as well as portal-like features (Fig. 5b), an observation in agreement with a proposed hybrid periportal hepatocyte cell type[40]. Similarly, subsets of nodules from the major classes, with the exception of central-like nodules, shared striking enrichment of numerous histone-associated transcripts (Fig. 5b and Supplementary Fig. 11). Upregulation of genes encoding histone proteins is described as the most prominent gene regulatory programme at the G1–S phase transition in human pluripotent cells[41].

### Tumour–stroma and tumour–immune cell connections

Next, by focusing on cellular ecosystems of the liver TME (Fig. 5a(bottom) and Fig. 5d), we identified prominent fibroblast-associated transcriptional signatures (for example, $Col1a1^+Col3a1^+$)[42,43] at the tumour–stroma border. We further observed regionally segregated expression patterns associated with haematopoietic/immune cell clusters (Fig. 5a). These included signatures likely to be associated with erythroblasts (for example, $Hbb-bt^+Slc4a1^+$)[44,45], platelets (for example, $Pf4^+Itga2^+$)[46], mast cells (for example, $Cpa3^+Cma^+$)[47,48], B cells ($Jchain^+Igkc^+$)[49] and neutrophil subpopulations (for example, $Elane^+Mpo^+$ and $Ngp^+Camp^+$)[45,50,51]. Signatures associated with Kupffer cells/macrophages (for example, $Marco^+Clec4f^+$ and $Csf1r^+C1qa^+$)[3,24] were primarily detected within the non-tumour compartment (Supplementary Figs. 12–18).

Our approach immediately revealed connections between tumour-intrinsic cell states and the microenvironment, such as a notable link between CCA and fibroblast-like signatures (Fig. 5c). This observation aligns with human data indicating that cancer-associated fibroblasts are the major cellular component of CCA-associated desmoplastic stroma[43]. Our approach indeed grouped fibroblast-like signatures alongside growth arrest-specific 6 (Gas6) and thrombospondin 1 (Thbs1), both of which were previously identified as marker transcripts for a mechanoresponsive cancer-associated fibroblast subpopulation[42] (Fig. 5d and Supplementary Fig. 12). Our results further pointed towards additional associations such as a link between CCA and macrophages (for example, $Csf1^+C1q^+$) as well as a connection between enriched erythroblast ($Hbb-b^+Slc4a1^+$) occurrence and the histone-associated subgroup of nodules (Fig. 5c).

### Complex genotype–phenotype relationships

Using spatial maps that combine phenotypic and genotypic data from the same tissue sections enables detailed investigation of phenotype–genotype relationships (Figs. 4a and 5a–d). We therefore assigned binary phenotype labels to nodules (Extended Data Fig. 9 and Methods) and calculated the OR values to assess the connection of each perturbation to specific tumour-intrinsic and microenvironmental phenotypic groups (Fig. 6a). Remarkably, in the setting of the combinatorial perturbations tested, our findings indicated that CCA reveal a strong positive association with VEGFA, exceeding any other observed linkage, as well as a strong negative association with mtCtnnb1 (Fig. 6a). Furthermore, consistent with the central role of WNT signalling in liver zonation[52], portal-like tumours revealed negative associations, whereas central-like nodules revealed positive associations with mtCtnnb1 (Fig. 6a). Notably, these genotype–phenotype observations align well with the aforementioned zonation-based classification of human HCCs and single-cell RNA-sequencing (scRNA-seq) data from human liver cancer, which revealed that the central-like HCC subtype is associated with Ctnnb1 mutations[33–35], hence indicating that our C-G2P approach mirrors features of human HCC biology.

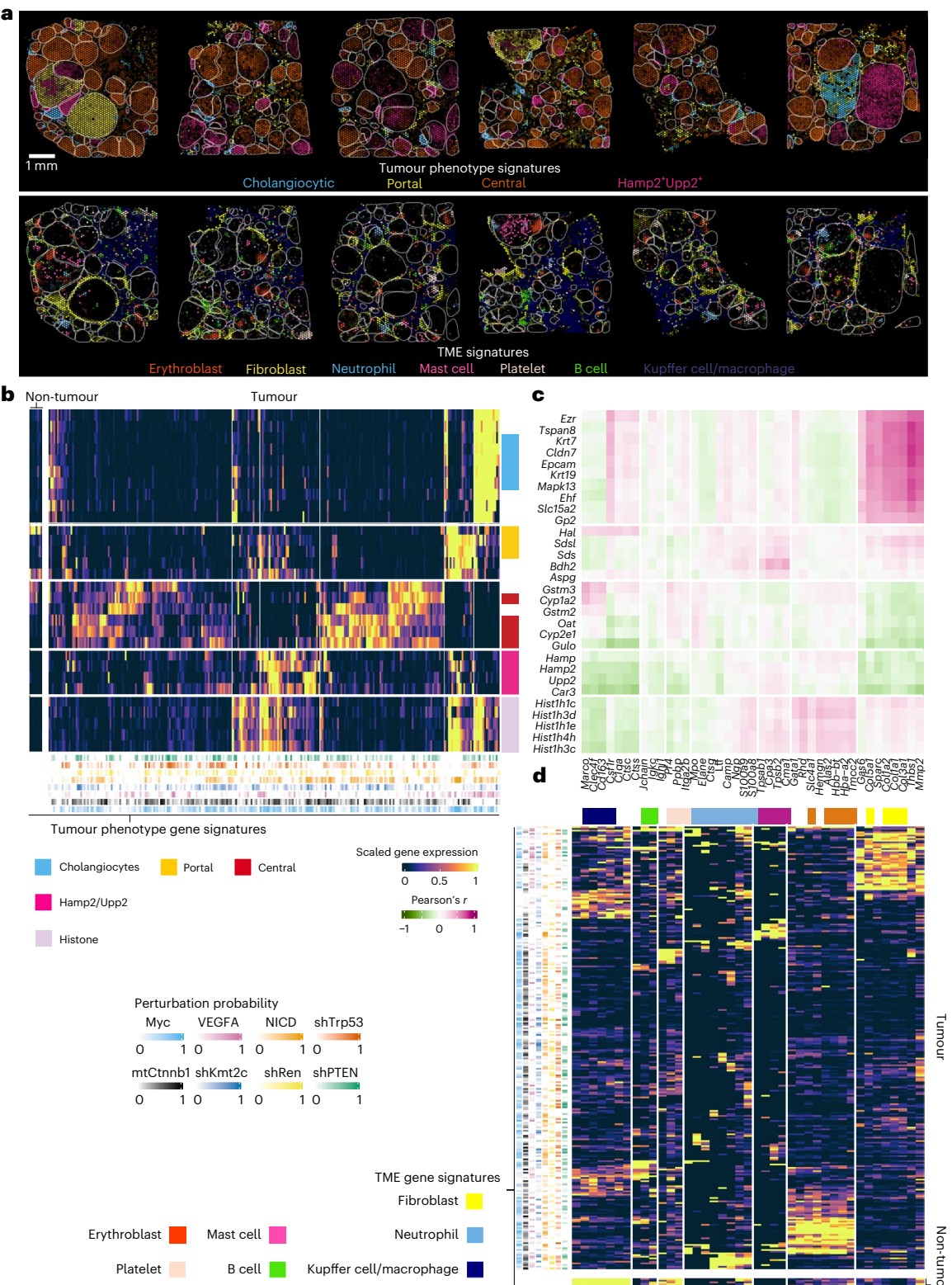

**Fig. 5 | C-G2P maps tumour ecosystems comprising hundreds of coexisting cancer clones. a**, The tumour ecosystem. Spatial maps of tumour-intrinsic phenotypes (top) and TME phenotypes (bottom) across six topographically separated regions. Colour shade depicts aggregated log(1*p*)-transformed expression of phenotype-associated transcripts (colour-code as in **b** and **d**). Nodule borders are highlighted (grey). The aggregated values for all samples and underlying quantitative data of individual transcript expression can be explored through the interactive web browser interface (https://chocolat-g2p.dkfz.de/). **b**, Co-clustering of tumour-intrinsic phenotypes by associated transcripts.

Tumour phenotypes are colour-coded. **c**, Associations between tumour-intrinsic and TME phenotypes. Pearson's correlation coefficient for each pair of tumour-intrinsic and TME phenotype-associated transcripts across all nodules. **d**, Co-clustering of TME phenotypes by associated transcripts. TME phenotypes are colour-coded. **b**,**d**, Clustering based on Spearman correlations. Phenotypes are subdivided using hierarchical clustering. Scaled ($p^{10}$) estimated plasmid probabilities per nodule are indicated (**b**(bottom) and **d**(left) (Methods). Data are derived from 324 nodules across six topographically separated regions used for 10X Visium from a single RUBIX experiment with two animals.

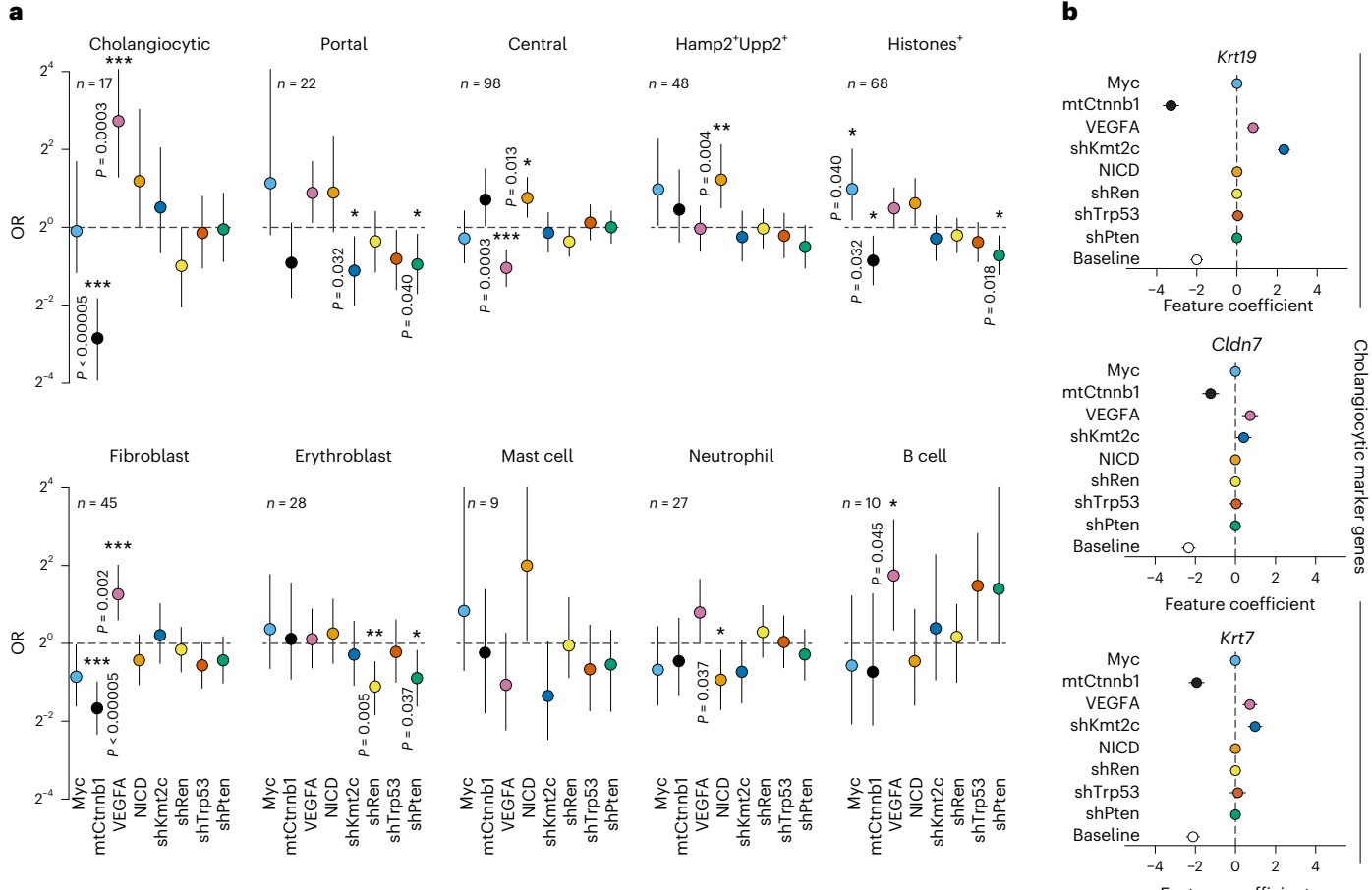

**Fig. 6 | C-G2P decodes relationships between complex genotypes and tumour-intrinsic and microenvironmental phenotypes. a**, Identification of genotype–phenotype relationships. Comparison of the prevalence of perturbations between phenotypic groups and the remainder of the nodules (total $n$ = 324) for tumour-intrinsic phenotypes (top) and TME (bottom) using ORs. OR > 1 indicates enrichment of perturbations within the phenotypic group; OR < 1 indicates depletion (Methods). The number of nodules with a given phenotype ($n$) is indicated. Note that groups are not mutually exclusive. The median and 90% CI are reported. Significant relationships are indicated (exact $P$ values are provided); two-tailed deviations from one, computed with 20,000 samples from the posterior (Methods); ***$P$ < 0.001, **$P$ < 0.01, *$P$ < 0.05. **b**, Identification of genotype–phenotype relationships for genes associated with cholangiocytes. A GLM model predicts gene expression signals at each 10X Visium spot using estimated probabilities of perturbation presence (Methods). Feature coefficients, shown as the mean and 3σ CIs, indicate associations between gene expression and perturbations for representative transcripts. Bayesian modelling of perturbation probabilities was used to infer the occurrence of individual perturbations per nodule (Methods). Data are derived from 324 nodules across six topographically separated regions used for 10X Visium from a single RUBIX experiment with two animals.

Focussing on relationships between tumour genotypes and TME phenotypes, we observed strong positive associations for VEGFA with fibroblast signatures, alongside a negative association with mtCtnnb1 (Fig. 6a), largely resembling the patterns observed for CCA and being in line with their prominent spatial association (Fig. 5c). A similarly positive association for VEGFA was observed for B cell-like signatures alongside negative association for Myc and mtCtnnb1, and positive associations for shTrp53 and shPten (Fig. 6a). Reflective of our findings, immune cell infiltration, as evaluated by CD45-positive IHC, was reported in a compound Trp53 and Pten-knockout HDTV injection model of liver cancer[20], whereas immune cell exclusion has been observed in a corresponding Myc–mtCtnnb1 model[19].

To broaden our analysis beyond binary phenotypes, we leveraged spot-level continuous expression of phenotype-associated marker transcripts. We therefore calculated associations for each perturbation using the aforementioned GLM analyses (Fig. 6b and Extended Data Fig. 10a,b). Despite its limited sensitivity to identify associations for transcripts that reveal sparse spatial expression (Methods), this analysis supported the identified associations for VEGFA and mtCtnnb1 for cholangiocyte-associated transcripts such as *Krt19* and *Cldn7* (Fig. 6b).

Similarly, GLM analyses substantiated the observed inverse mtCtnnb1 associations for portal-like markers (that is, negative association for *Sds* and *Sdsl*) versus central-like markers (that is, positive associations for *Oat* and *Gulo*; Extended Data Fig. 10a). Notably, GLM analyses uncovered additional transcript-specific contributions of perturbations not readily apparent using the nodule phenotype-binarization approach (Fig. 6a). For example, we observed a marked relation of the cholangiocyte-associated transcripts *Krt19* and *Gp2* with shKmt2c perturbation (Fig. 6b and Extended Data Fig. 10a,b).

Given that multiple transcripts associated with predefined phenotype signatures shared similar 'GLM-patterns' (Extended Data Fig. 10a,b), we finally interrogated perturbation–phenotype associations on a transcriptome-wide scale (Methods). Focusing on 1,283 genes that showed associations with perturbations, we observed clusters of transcripts that correspond to specific cell states (Extended Data Fig. 10c). For example, global-GLM analysis grouped together fibroblast-associated transcripts such as *Col1a1*, *Col1a2*, *Gas6* and *Thbs1* or transcripts related to the aforementioned histone-enriched phenotype. Similarly, GLM analysis aggregated *Krt19*, *Epcam*, *Gp2* and *Krt7*, all of which defined the cholangiocytic phenotype that was

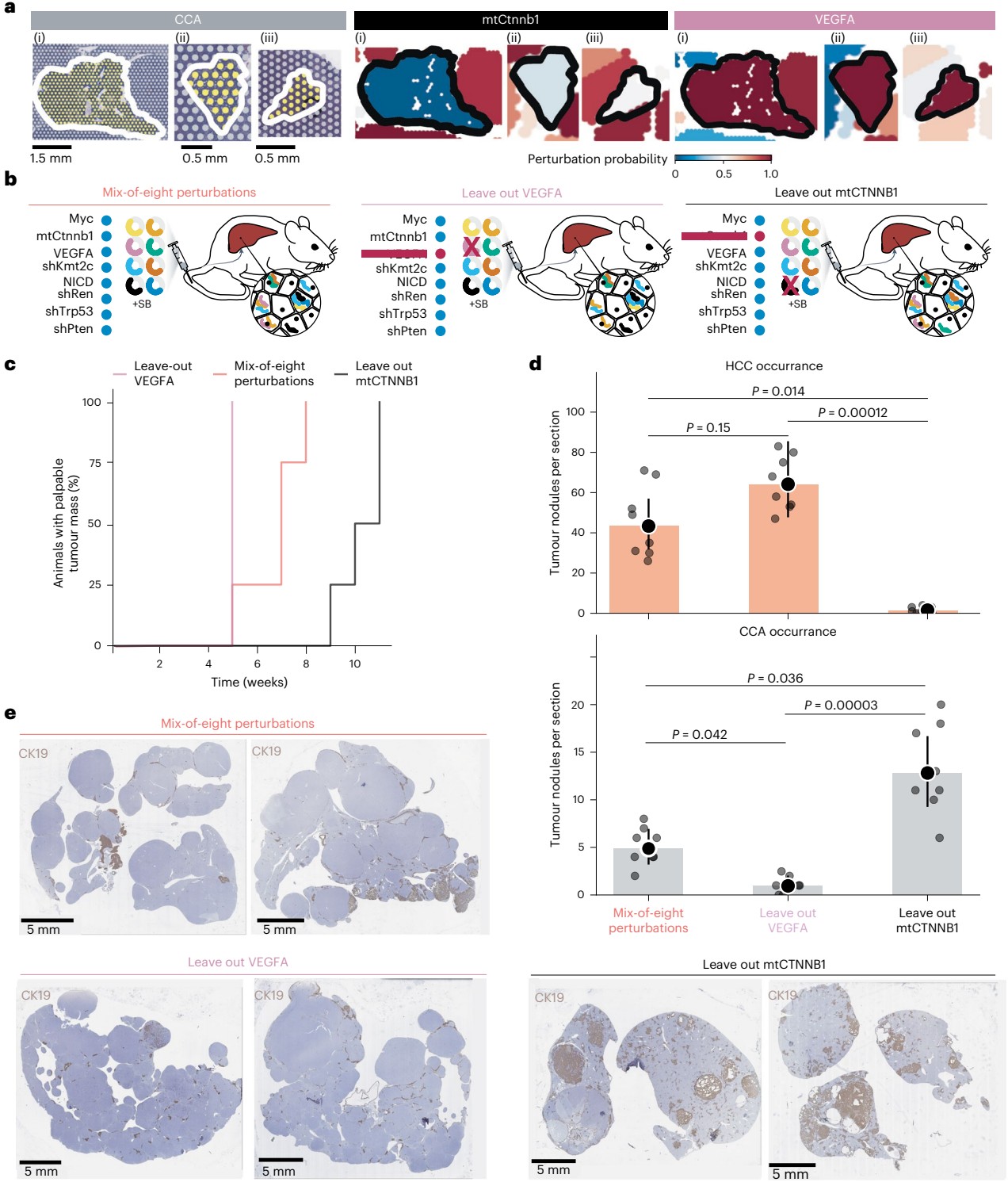

**Fig. 7 | VEGFA and mtCTNNB1 confer epistasis control of CCA development within heterogeneous tumour ecosystems. a**, Spatially resolved co-occurence of VEGFA and mutual exclusivity of mtCtnnb1 for the CCA tumour subtype as revealed by C-G2P. Magnified views of three representative nodules ((i)–(iii)) identified as CCA. Nodules identified as CCA (left; the area covered by the tumour nodule is indicated as 10X Visium spots in yellow) as well as the mtCtnnb1 (middle; as in Fig. 3c) and VEGFA (as in Fig. 3c) perturbation probabilities are shown. Bayesian modelling of perturbation probabilities is used to infer the occurrence of individual perturbations per nodule (Methods). Data are derived from 324 nodules across six topographically separated regions used for 10X Visium from a single RUBIX experiment with two animals. Perturbation probabilities for all samples can be explored through the interactive web browser

https://chocolat-g2p.dkfz.de/. **b**, Experimental design. Parallel RUBIX mouse models were performed using the leave-one-out experimental design. **c**, Time to tumour occurrence. Animals were palpated twice weekly to monitor tumour development. **d**, Histological quantification of liver tumour subtypes. H&E images were analysed, and tumour nodules were counted and classified as either HCC (top) or CCA (bottom); two independent liver-tissue sections per animal. The median ± s.d. alongside individual tumour counts are indicated. Group comparisons used a two-sided Kruskal–Wallis test with Dunn's post-hoc test (Holm–Bonferroni correction). Exact adjusted *P* values are shown. **e**, Abundance of CCA. CK19 IHC was used as a cholangiocyte marker. Representative samples from a total of two separate sections per animal are depicted. **b**–**e**, n = 4 animals per group.

linked to positive associations with VEGFA and negative associations with mtCtnnb1 (Extended Data Fig. 10c and Fig. 6a,b).

## Epistatic regulation of CCA development

Our C-G2P experiments suggested that VEGFA and mtCTNNB1 mediate opposing epistasis effects relevant to CCA development within the genetically heterogenous tumour ecosystems analysed, as nodules of this tumour class were consistently negative for mtCTNNB1 and positive for VEGFA (Fig. 7a). To substantiate this finding, we took advantage of the flexibility of our RUBIX mouse model and adjusted combinatorial complexity of genetic perturbations to directly assess the individual contributions to tumour development and phenotype. Specifically, we employed a 'leave-one-out' strategy, where we compared animal cohorts that received a full mix of all eight perturbation plasmids to cohorts where either VEGFA or mtCTNNB1 was deliberately omitted (Fig. 7b). In mice that received all eight perturbation plasmids, all animals developed tumours within eight weeks and histopathology revealed multiple tumour nodules, a fraction of which identified as CCA (HCC, 42 ± 17.7 versus CCA, 5 ± 1.96 per section). Immunohistochemistry for CK19 confirmed the CCA classification (Fig. 7c–e), matching our earlier C-G2P results (Fig. 5). In the cohort missing VEGFA, tumours developed faster and all animals presented tumours within five weeks. The majority of nodules identified as HCC (HCC, 63 ± 13.6 versus CCA, 1 ± 0.756 per section). Using IHC, we observed that tumour nodules were generally negative for CK19, confirming the absence of CCA, although CK19 was still detectable in normal bile ducts (Fig. 7c–e). In contrast, the absence of mtCTNNB1 delayed tumour development, extending the time for all animals to present tumours to 11 weeks. Observed tumours were predominantly CK19-positive CCAs (HCC, 1 ± 1.51 versus CCA, 12 ± 4.71 per section; Fig. 7c–e). Together, these results confirm predictions of C-G2P-based findings revealing that whereas VEGFA expression essentially contributes to CCA development in the setting of combinatorial alterations investigated, mtCTNNB1 elicits a dominant epistasis-masking effect on this particular liver tumour subclass.

## Discussion

Here we introduced PERTURB-CAST, an approach that seamlessly integrates perturbation mapping for in vivo functional genomics with spatial transcriptomics interrogation by redeploying RTL probes from commercial technology for molecular barcode identification. Combined with RUBIX, which allows induction of hundreds of tumours, each with distinct combinations of alterations, in a single tissue, our CHOCOLAT-G2P framework offers the capability to characterize tumour gene expression and cellular microenvironments, and helps address the long-standing question of how multiple genetic changes interact to shape disease phenotypes within cellular ecosystems.

By applying C-G2P in an autochthonous mouse model of liver cancer, we explored a wide range of cancer-driving combinatorial genotypes sampled from nearly all of the $2^8$ combinations of perturbations present in hundreds of coexisting tumours. The integration of PERTURB-CAST for spatial transcriptomics enabled simultaneous mapping of the genotype of each nodule alongside tumour-intrinsic and microenvironment-related phenotypes on the same tissue sample. PERTURB-CAST eliminated the need for complementary readouts as well as the requirement for analyses on serial tissue samples, thereby preserving spatial relationships and providing the basis for detailed genotype–phenotype analyses.

Interrogating 324 liver tumour nodules from a single C-G2P experiment revealed mutual exclusivity between mtCtnnb1 and VEGFA, indicating epistatic fitness effects of these two alterations. C-G2P revealed that their exclusivity was further underscored by phenotypic divergence. Specifically, VEGFA induced a cholangiocytic histology and gene expression profile. VEGFA-perturbed nodules also exhibited a greater abundance of cancer-associated fibroblasts compared with

nodules with mtCtnnb1, indicating that genetic alterations also shape, and possibly co-opt, their TME. In contrast, mtCtnnb1, which we identified as a crucial contributor to overall liver tumour occurrence, masked the emergence of the CCA subtype, thus exemplifying Bateson's classical definition of epistasis[26].

C-G2P can be applied and extended in a number of ways. First, it is straightforward to adjust combinatorial complexity and exchange the alterations to additional cancer drivers[1] and other perturbations, potentially within a compressed screening framework[16]. C-G2P may also be conducted in different mouse strains or growth conditions to model interactions and selective forces between tumour genomes and host genetics, immunocompetence, environmental exposures and therapeutic interventions[7,53]. Second, we envision the applicability of PERTURB-CAST and C-G2P beyond liver cancer. Currently available autochthonous animal disease models that similarly build on stable integration of perturbations include diverse tumour types such as lung, pancreas, stomach and soft tissue cancers[7,54,55]. Moreover, the perturbation plasmids we employed here (Extended Data Fig. 1 and Methods) are expected to be compatible with scRNA-seq readouts and imaging-based screening platforms[14,16,17,56–58] that could in the future enable complementary insights.

In summary, C-G2P provides a multiplexed approach for higher-order combinatorial cancer screens in an individual mouse. Its design—built on PERTURB-CAST, which uses off-the-shelf spatial transcriptomics protocols—facilitates comprehensive readouts of tumour genotypes and spatial phenotypes from the same tissue sample.

As C-G2P may be extended to other disease models, perturbations and (spatial) omics technologies, we envisage a broad range of applications to decode the relationships between complex genotypes and phenotypes[59] within the holistic context of tissue and the entire organism.

There are some limitations of the study. First, RUBIX currently generates tumour heterogeneity by establishing random combinatorial alterations simultaneously. Although this scalable approach could aid to initially explore a vastly unknown epistatic interaction space, this contrasts tumour development in humans where, in most cases, cells sequentially acquire alterations[1]. To address this, experimental strategies that enable stepwise introduction of genetic alterations require further exploration. Inducible perturbation systems, previously used in the liver cancer mouse model employed here, may provide a potential solution. Alternatively, we envision that serial injections of differentially barcoded plasmid pools could capture spatially resolved in vivo functional genomics data across temporal alteration trajectories. Although the random simultaneous introduction of alterations and establishment of hundreds of coexisting tumour nodules could provide a valuable opportunity for studying interclonal interactions, the complexity of our current model may as such not fully recapitulate the ancestral lineage, clonal evolution and genetic epistasis seen in human tumours. To more systematically investigate genetic interactions, we speculate that the multiplexing capabilities offered by MultiMir combinatorial RNA interference[60] as well as CRISPR/Cas12a[61] warrant further investigation. For multiplex CRISPR perturbations, it may however, be necessary to employ modifications that avoid double-stranded breaks, such as CRISPR interference, to minimize the risk of unwanted chromosomal rearrangements and other detrimental effects[16].

Second, PERTURB-CAST hinges on the availability of robust RNA-detection probes, a requirement that is not always satisfied, as evidenced by the failure to detect 5/38 barcodes tested in this study. We anticipate that prospective massively parallel assays[61] based on C-G2P could be leveraged to select reliable probes.

Third, the moderate resolution of the 10X Visium platform we employed does not allow for single-cell analyses. Additional computational approaches and next-generation spatial transcriptomics platforms could address this shortcoming[62–65]. PERTURB-CAST could

indeed be integrated with complementary probe-based single-cell spatial omics technologies such as the recently introduced CosMx WTx assay, which uses two in situ hybridization probes per transcript with 35–50 nt RNA-targeting domains[66]. Finally, PERTURB-CAST may offer immediate implementation into high-resolution 10X Visium HD, which operates on the same RTL-probe-based transcript capture technology[67].

## Methods

### Animal experiments

Group size was determined on the basis of our experience in previous experiments[68,69]. For HDTV injections, eight-week-old female C57Bl/6 animals were purchased from Envigo. The mice were injected (into the lateral tail vein in 5–7 s) with 5 µg DNA of each of a total of eight perturbation plasmids (40 µg total DNA) mixed with 20 µg CMV-SB13 transposase (1:2 ratio) prepared in sterile 0.9% sodium chloride solution in a total volume corresponding to 10% of the body weight, as described before. Two animals were used. We labelled this approach RUBIX as the perturbation plasmids are equipped with molecular barcodes (Extended Data Fig. 1) and the plasmid mixtures injected allow for all possible combinations to become integrated in the genome of hepatocytes (Fig. 1a). All animals were monitored twice weekly and animal experiments were performed in compliance with all relevant ethical regulations determined in the animal permit. After tumours were palpable (10 weeks), the animals were euthanized and their livers were harvested (Extended Data Fig. 2). As a control group, two animals were injected with 40 µg pT3-EF1-shRen and 20 µg CMV-SB13 transposase (1:2 ratio) prepared in sterile 0.9% sodium chloride. For fixation, livers were incubated in 4% paraformaldehyde for 48 h. The sample processing procedure is illustrated in Extended Data Fig. 4. For the leave-one-out experiments (Fig. 7), HDTV injections were performed essentially as described before. For omission of either VEGFA or mtCTNNB1 perturbation from the plasmid pools, the respective plasmids were replaced with a control plasmid. Housing conditions for the mice were: 12 h light–12 h dark cycle, an ambient temperature of 20–24 °C and relative humidity of 45–65%. All animal experiments were approved by the regional board Karlsruhe, Germany.

### 10X Visium for FFPE spatial transcriptomics

Spatial transcriptomics were performed using the manual 10X Visium workflow for samples embedded in paraffin blocks or the 10X Visium CytAssist workflow for samples already placed on glass slides and stained with H&E (Extended Data Fig. 2). Both workflows were carried out according to the manufacturer's protocol (CytAssist, CG000495, RevC; manual Visium, CG000407, RevD). Briefly, slices of approximately 5 µm were cut from FFPE blocks using a microtome and floated onto a water bath at 42 °C until all wrinkles were resolved. For the manual Visium workflow the slice was then placed inside the capture frame of the spatial transcriptomics slide (M.R.). Slices used for the CytAssist workflow were placed on frosted glass slides. Deparaffinization and staining of the slides was similar between both workflows. After drying the slide, paraffin was removed through incubation at 60 °C for 2 h and a subsequent incubation in xylol. Rehydration was done by sequential washes with decreasing ethanol concentrations. After rehydration, the tissue was stained with H&E and imaged using a Leica Aperio AT2 microscope at ×40 magnification. Following imaging, the slides were destained by incubation in 0.1 N HCl and formalin crosslinks were removed by incubation with TE buffer pH 9.0 for 1 h at 70 °C (manual workflow) or decrosslinking buffer for 1 h at 95 °C (CytAssist workflow.) The tissue was then permeabilized and incubated with RTL probes at 50 °C for approximately 20 h. Free probes were washed away and the bound probes were ligated, followed by washing steps to remove unligated probes. For the manual workflow, the probes were released by treating the slices with RNase and a permeabilization enzyme. For the CytAssist workflow, the slices were stained with a diluted eosin solution and placed in the CytAssist together with the Visium spatial transcriptomics slides and incubated for 30 min at 37 °C with RNase and permeabilization enzyme. For both protocols, the spatial barcode was added to the probes by extending them and the probes were released using a 0.08 M potassium hydroxide solution. For the CytAssist workflow, a pre-amplification PCR consisting of eight cycles was performed. After clean-up with 1.2× SPRIselect beads, 25% of the product was used as input for the index PCR. For both protocols, a quantitative PCR was used to select the number of cycles for the index PCR. To reduce PCR duplicates and avoid overamplification, cycle number at a $C_q$ value of 10% was used for the index PCR. The PCR product was purified using 0.85× SPRIselect beads.

For samples already stained and mounted on a slide, the slides were first imaged and then incubated in xylol to remove the coverslip. Sample rehydration was done by sequential washes with decreasing ethanol concentrations. The slides were destained and decrosslinking was performed by incubating with a decrosslinking buffer at 95 °C for 1 h. After decrosslinking, the samples were incubated with probes for 20 h at 50 °C. Excess probes were washed away and the probes were ligated. Thereafter, the unligated probes were washed away. The samples were stained again with eosin and placed in the 10X CytAssist together with the spatial transcriptomics slides. A mixture of RNase and permeabilization enzyme was added to the spatial transcriptomics slides and the 10X CytAssist was started. After incubation, the spatial transcriptomics slides were removed and the enzymes were washed off. The spatial barcodes were attached to the probes with an extension enzyme. Probes were released using 0.08 M potassium hydroxide solution. The probes were then amplified through eight PCR cycles; 25% of the purified PCR products were used as input for the index PCR. The cycle number of the index PCR was determined using the cycle number at a $C_q$ value of 10%.

For all samples, the final sample concentration was determined using Agilent Tapestation 4150 with D1000 HS tapes. Sequencing for both protocols was performed on an Illumina NovaSeq6000 system. Four samples were pooled on one SP flow cell with 100 cycles to aim for a read count of $250 × 10^6$ reads per sample. The FASTQ files and the alignment were done using spaceranger 2.0.1. A total of 12 spatial transcriptomics datasets were generated (Extended Data Fig. 2). Note that the utility of sample ML-II_B_2Cyt is constrained by tissue detachment of the sample during the processing for 10X Visium CytAssist.

The 10X Visium for FFPE protocol engages RTL probes that capture a 50-nt sequence specific to endogenous transcripts (note that we used Visium mouse transcriptome probe set v1). We leveraged this strategy to likewise capture molecular barcodes via RTL probes (Fig. 1). We hence labelled this approach PERTURB-CAST.

### Economized molecular barcode selection

To enable PERTURB-CAST, we aimed to avoid additional expenses and protocol modifications by redeploying RTL probes from commercially available 10X Visium reagents (originally designed to detect endogenous transcripts) as barcode identification reagents, provided that the selected transcripts are not expressed in mouse liver. We named this strategy redeploy probes for barcode capture (REDPRO-BC; Fig. 1 and Supplementary Fig. 1). To this end, we initially analysed a publicly available bulk RNA-seq dataset including a total of 128 murine liver samples (GSE137385)[70] to identify transcripts that were generally not detected (fragments per kilobase of transcript per million mapped reads = 0 over all samples). Note that this approach can be error-prone due to the initial source data. Consequently, we went on to validate non-expression of selected transcripts (olfactory, vomeronasal and taste receptors) in additional datasets (including bulk RNA-seq from GSE148379)[19], information provided in MGI GXD[71] as well as 10X Visium data[24]. The endogenous transcripts associated with the REDPRO-BCs used in this study are illustrated in Extended Data Fig. 3 and the respective nucleotide sequences for 10X Visium RTL-probe capture barcodes (reverse complement to RTL-probe sequence provided by 10X Genomics) are

listed under the section 'Molecular cloning'. Note that we used Visium mouse transcriptome probe set v1. Visium mouse transcriptome probe set v2 is not compatible with the barcodes employed in this study.

## Molecular cloning

The transposon plasmids used in this study including overexpression constructs for NICD, mutant human *CTNNB1* (T41A) and human *MYC* and potent shRNA constructs targeting *Trp53*, *Pten*, *Kmt2c* and *Renilla luciferase* were previously described and validated in animal experiments[19,68,72]. For spatial transcriptomics, note that NICD overexpression can be investigated via *Notch1* expression given that the 10X Visium RTL probe identifies the exogenous transcript introduced. However, endogenous *Notch1* is similarly identified. VEGFA overexpression plasmid was cloned by insertion of a codon-optimized gene fragment (gBlock, IDT) based on *Vegfa* NCBI reference sequence NP_033531.3 by replacing h*MYC* from a previously validated expression plasmid[68] using NEBuilder HiFi-DNA assembly according to the manufacturer's protocol (NEB). All plasmids were individually modified to express molecular barcodes. Specifically, fluorescent protein-based peptide barcodes (mKate2, mOrange2 and mWasabi) were ordered as codon-optimized gene fragments (gBlock, IDT) and cloned into previously validated shRNA expression plasmids to replace GFP[68] using NEBuilder HiFi-DNA assembly according to the manufacturer's protocol. Small-peptide barcodes (for example, AU1, AU5 and so on), as described previously[14], were ordered as oligonucleotides (Sigma), annealed and cloned using NEBuilder HiFi-DNA assembly according to the manufacturer's protocol. Long RNA barcodes (stretches of at least 650 nt derived from the combination of multiple oligonucleotide–miner probe sequences designed against *Arabidopsis thaliana* Chr1 (ref. [73]) were ordered as gene fragments (gBlock, IDT) and cloned using NEBuilder HiFi-DNA assembly according to the manufacturer's protocol. REDPRO-BC triplet arrays were ordered as gene fragments (gBlock, IDT) and cloned using NEBuilder HiFi-DNA assembly according to the manufacturer's protocol. Briefly, each REDPRO-BC triplet array incorporates one 50 nt sequence derived from olfactory receptors, one 50 nt sequence derived from taste receptors and one 50 nt sequence derived from vomeronasal receptors (based on 10X Genomics RTL-probe sequences against murine transcripts; note that we used Visium mouse transcriptome probe set v1; sequences below), separated and flanked by spacer sequences of approximately 20 nt to avoid potential steric hindrance during hybridization. The spacer sequences used were derived from T7 and T3 promoters (described in ref. [56]) and/or AsCas12a-DR sequences (described in ref. [61]), and/or the 10X Capture sequences cs1 and cs2 (described in ref. [57]), and as such provide additional functionality, which was not tested in this study. Single 50-nt REDPRO-BCs were ordered as oligonucleotides (Sigma), annealed and cloned using NEBuilder HiFi-DNA assembly according to the manufacturer's protocol. Note that the REDPRO-BC length should enable straightforward en masse cloning engaging commercially available oligonucleotide pools (such as in ref. [61]), which was not tested in this study. Peptide barcodes were integrated in frame with the respective coding regions. RNA barcodes were integrated in the 3′ untranslated region of coding regions expressed under the control of a polymerase II promoter (EF1), unless otherwise specified. Subsets of perturbation plasmids were equipped with REDPRO-BC arrays either 5′ and 3′ of the shRNA expression cassette (mir-E-based) to account for mir-E processing[74], or with additional REDPRO-BC arrays driven by a polymerase III promoter (hU6) in reverse orientation to the EF1-driven transcript. Extended Data Figs. 1 and 3 provide simplified illustrations of the plasmid design and barcode position. Respective FASTA sequences of plasmids are available on request. Plasmids were validated by restriction digest and Sanger sequencing (Microsynth).

Selected REDPRO-BC barcode sequences based on Visium mouse transcriptome probe set v1 were as follows.

**Myc.** Olfr103, TGGGAGTGAGAGACATACAAGAACCACAGCCCTTTCTCTTTGCTATTTTC; Tas2r102, AACACAAGTGTGAATACCATGAGCAATGACCTTGCAATGTGGACCGAGCT; Vmn1r1, TAAAAGGCAGTGTCAGTACCTTCACAACACCAGCATTTCCCGCAAAGCAT; Olfr1018, CAGTTCCATGGTTATCAATGTTCTCACCTTGAGTTTGCCCTACTGTGGAC; Tas2r118, TTATTGGCACTGTGTTTGATAAGAAATCTTGGTTCTGGGTCTGCGAAGCT; Vmn1r174, ACTTCAACCAGAGGCCAGAGCAGCAAACACAATTCTCATGCTGATGATCA; Olfr1, TGGCCAGCATCTTTCTTGTCCTTCCATTTGCACTCATTACCATGTCCTAT.

**mtCtnnb1.** Olfr1000, GGCACAGTAGGTATGTTCACTGGTCTGATAATTCTGGGGTCCTATGTATG; Tas2r103, TGTCACTAATCACAGGGTTCTTGGTATCATTATTGGACCCAGCTTTATTG; Vmn1r178, GTCTCTTCATGAGTCATTTCAGTAAAGTTTTTGCTGCAGGATTCCCCACT; Olfr1019, TGCTTGGTCCTAATGCTGGGCTCTTACTTCGCTGGCCTAGTGAGTTTAGT; Tas2r119, GATATCCAGGTTGGTGCCATGGCTGATCCTGGCATCGTGGTCTATGTAA; Vmn1r175, AGTACAAACATGTGCTCCACCTGCTTTCTGAGCACTTATCAGCTTGTCAC.

**NICD.** Olfr1006, AGGGAACATGTTGCTGGTTGTTTTAATCCGAATTGATTCTAGACTGCATA; Tas2r105, GACCTCGGAGATGTACTGGGAGAAAAGGCAATTCACTATTAACTACGTTT; Vmn1r139, AAGCATTGGCAAGTCACAGGCAAAGAGTGACACAGAGACGTTCCTCAATT.

**VEGFA.** Olfr1002, AGGCCTTATAAGCACTGTGGTCCATACTACTTCTGCATTTATTCTTCCAT; Tas2r104, TAACGTGGCTAGCTTCCTTTCCGCTAGCTGTGAAGGTCATTAAAGATGTT; Vmn1r12, ACTACATTGTCAGGAGCTTGATTTTAACTGTGACAACTTCCAGGGATATG.

**shPTEN.** Olfr1013, GTACACATTGACTTTGATGGGAAATAGCTCCCTCATTATGTTAATCTGCA; Tas2r110, ACTAGTGAATATCATGGACTGGACCAAGAGAAGAAGCATTTCATCAGCGG; Vmn1r170, TGATTCTCCTGAACAGACACCACCACAGACTGCAGCATATTCAATCCACA; Olfr1015, TGTCTATGTGAAAATCCTTTCCAGTATGGTGGGCTTCACTGTCCTCTCAA; Tas2r114, TGTAATTTGTCTGTTAATCCCAGAAAGCAACTTGTTATTCATGTTTGGTT; Vmn1r172, GGAAGTAAATGCCCAGAGAGTCTTCAAAGGAAGACAGTCATAGCTGTTTT.

**shREN.** Olfr1008, CCAGGCTCTGCTATTCACCAGTAAAATTTTCACATTAACTTTCTGTGGCT; Tas2r106, AAGGCACTGAAGCAATTAAAATGCCATAAGAAAGACAAGGACGTCAGAGT; Vmn1r157, CAGATCCTCTTGCTTTGCCATTTTGAGGTTGGGACCGTGGCCAATGTCTT.

**shTRP53.** Olfr1009, CCAGAGACTCTGCATACAGCTGGTGATCGGACCCTATGCTGTTGGCTTTT; Tas2r107, GCTCTCTAAGATCGGTTTCATTCTCATTGGCTTGGCGATTTCCAGAATTG; Vmn1r167, GTTTCAGTATAGGCATGCGGCATCTTATCATTTGCCCATGATGGAGTGTTC; Olfr1014, TTGCTGTGTATGCATTAACTGTGTTAGGAAACAGCACCCTCATTGTGTTG; Tas2r113, GATCAATCATTGTAACTTTTGGCTTACTGCAAACTTGAGCATCCTTTATT; Vmn1r171, AACAGCACTGCCCTCATGATCACTATTCCGTTGACCAATGAAGTTGTCTC; Olfr107, TTACTGCTTTCTTGCTCAGACACTCACCTCAGTGAGGGCCTGATGATGGC.

**shKMT2C.** Olfr1012, ATCTACTCTCGGCCAAGTTCCAGTTATTCCTTGGAAAGGGATAAAATGGT; Tas2r109, TTCTAGAATTTTCCTGCTCTGGTTCATGCTAGTAGGTTTTCCAATTAGCT; Vmn1r169, GGTACCTGGGGTAGGGTGATGCTCCATGGAAGAGCCCCCAAATTTGTGAG.

## Histopathology

After fixation, representative specimens of the liver were routinely dehydrated, embedded in paraffin and cut into 4-μm-thick sections. The tissue sections were stained with H&E according to standard protocols. Slides were scanned using a SCN400 slide scanner (Leica Biosystems) at ×20 magnification.

**Table 1 | Details of antibodies used in IHC**

| Antibody | Host | Company | Catalogue number | Antigen retrieval | Dilution | Detection reagent | Chromogen | Blocking |
|---|---|---|---|---|---|---|---|---|
| CK19 | Rabbit | Abcam | ab133496 | Dako target retrieval solution 10× concentrate, pH9 (catalogue number S2367) | 1:100 | PolyviewPlus AP anti-rabbit | Permanent AP red | / |
| HNF4α | Rabbit | Abcam | ab181604 | Dako target retrieval solution 10× concentrate, citrate pH6, (catalogue number S2369) | 1:400 | PolyviewPlus AP anti-rabbit | Permanent AP red | / |
| GS | Mouse | BioScience | BD610517 | Dako target retrieval solution 10× concentrate, citrate pH6 (catalogue number S2369) | 1:1,000 | PolyviewPlus HRP anti-mouse | DAB | $H_2O_2$ |
| tRFP | Rabbit | Evrogen | AB 233 | Dako target retrieval solution 10× concentrate, citrate pH6 (catalogue number S2369) | 1:500 | PolyviewPlus AP anti-rabbit | Permanent AP red | / |
| GFP | Rabbit | Cell Signalling | 2956 | Dako target retrieval solution 10× concentrate, pH6 (catalogue number S1699) | 1:100 | PolyviewPlus HRP anti-rabbit | DAB | $H_2O_2$ |

Nodule annotation was initially performed by experienced pathologists (D.F.T. and H.W.) based on H&E-stained sections using the quPath software and the 10X Loupe Browser software. Nodule annotation was further refined based on specific transcript expression (example provided in Supplementary Fig. 3) using the 10X Loupe Browser software (H.W. and M.B.).

### IHC
After heat-induced antigen retrieval at pH 6 or pH 9, FFPE tissue sections were incubated overnight with the primary antibody and blocked with hydrogen peroxide if necessary. Depending on the primary antibody, an anti-mouse or anti-rabbit secondary antibody conjugated to horseradish peroxidase (HRP) and alkaline phosphatase (AP), respectively, was applied (PolyviewPlus, ENZO Life Sciences GmbH). The signal was visualized using either 3,3'-diaminobenzidine (Dako liquid DAB+ substrate, Agilent Technologies, Inc.) or AP (Permanent AP red, Zytomed Systems) as a chromogen. Details are given in Table 1.

Slides were scanned using a SCN400 slide scanner (Leica Biosystems) at ×20 magnification. The individual histochemical GFP, RFP and GS staining was evaluated using the quPath software as: high, very intense uniform staining; moderate, moderate intensity or intense non-uniform staining; and low, low intensity and non-uniform staining. Mapping of barcode signals to respective IHC data was performed by manual assessment of marker staining on IHC images using the quPath software (H.W. and M.B.). Next, corresponding tumour nodules were selected, categorized and stratified using the 10X Loupe Browser software (H.W. and M.B.). Note that this approach can be error-prone due to shifts in the z-plane based on serial sectioning for each IHC sample and samples used for 10X Visium.

### Computational data analysis
We used Python (v3.9.12) and the packages anndata (v0.11), scanpy (v1.9.8), squidpy (v1.4.1), sagenet (v1.1.0), Cell2module (GitHub version, retrieved in February 2024), pandas (v2.0.3), Torch (v2.1.1), Numpy (v1.24.3), Matplotlib (v3.7.2), Pyro (v1.8.6), SciPy (v1.11.3) and alpha_shape (GitHub clone c171a7d). We used R (v4.3.0) and the packages SingleCellExperiment (v1.24.0), ZellKonverter (v1.12.1), scater (v1.30.1), ComplexHeatmap (v2.16.0), glasso (v1.11), FSA (0.9.6), dplyr(1.1.4), ggplot2 (v3.5.1), igraph (v2.0.1.1) and scran (v1.28.2).

### Genotyping
**Barcode expression pre-preprocessing.** Before analysing the 10X Visium data, we applied a filtering criterion of unique molecular identifier counts of >5,000. For the CytAssist platform, we excluded the outermost layer of spots due to unexpectedly high unique molecular identifier counts. In addition to manually identifying cancerous nodule

regions, we annotated 'normal tissue' regions to acquire representation of areas without any cancerous cells. The selection of normal regions was based on a minimum distance of 250–700 μm from the nearest tumour, depending on the tissue section, to minimize contamination from adjacent tumour regions. Tumour nodules were defined as described in the 'Histopathology' section.

**Bayesian modelling of perturbation probabilities from barcode counts.** In our model, the observed expression count matrix $D_{s,b}$ (spots $s$ by barcode genes $b$) is assumed to follow a negative binomial distribution. This matrix has a mean $\lambda_{s,b}$ and overdispersion $\phi_b$. The overdispersion parameter $\phi$ is sampled from a Gamma distribution (shape = 1,000; rate = 0.03) skewed towards higher values to encourage the likelihood to approximate a Poisson distribution in the absence of overdispersion evidence.

The mean expression for each spot is calculated as:

$$\lambda_{s,g} = \mu_s \sum_r A_{s,r} \sum_g G_{r,g} B_{g,b} \kappa_b + \xi_b$$

where $\mu_s$ represents the sensitivity of each spot, $A_{s,r}$ maps spots to clonal nodules $r$, $G_{r,g}$ estimates the expected number of integrated copies of plasmid $g$, $B_{g,b}$ links plasmids to their corresponding barcodes and $\kappa_b$ is the barcode expression rate; $\xi_b$ is a barcode-specific additive noise term. The per-nodule plasmid integration number ($G_{r,g}$) is modelled as an expected count of integration events, described by $F_{r,g,o}$. Here $F_{r,g,o}$ captures the probability of no integration and higher indices reflect the integration of increasing numbers of copies. This is modelled using a Dirichlet distribution with a uniform concentration parameter and an order $o = 6$. This assumes the maximum of six copies of the same plasmid per clone, balancing the need to capture dosage-dependent variation with the practicality of limiting the number of parameters to be learnt. For normal tissue regions, the probability of perturbation presence was fixed to $1 \times 10^{-3}$ to indicate near absence, but not zero, to prevent numerical instabilities.

Both $\kappa_g$ and $\xi_g$ are sampled from a weakly regularized exponential distribution with a rate of one. Spot sensitivity $\mu_s$ is modelled by a gamma distribution (shape = 3; rate = 0.3) that is weakly centred around one. This parameter accounts for both the sensitivity variability across 10X Visium spots and the dilution effects on the barcode signal due to varying tumour purity.

**Perturbation probability model inference.** We infer our Bayesian model using a variational posterior approximation. Specifically, we employ a log-normal guide distribution to approximate the parameters that have exponential and gamma distributed priors. In addition, we use a Dirichlet approximation for the posterior of $F_{r,g,o}$. The model and its

inference framework are implemented in Pyro (v1.8.6)[75]. The variational approximation is conducted via the stochastic variational inference method[76], employing the Adam optimizer set at a learning rate of 0.01 and using three samples for Kullback–Leibler divergence estimation. We perform inference over 10,000 gradient steps, monitoring the evidence lower bound to assess convergence.

**Occurrence of individual perturbation combinations.** Considering the probabilistic nature of our estimates for $F_{r,g,o}$, we utilized samples from the estimated posterior to analyse tumour populations. Due to uncertain integration copy number estimates, we focused on presence/absence categories. These probabilities were computed as $1 - F_{r,g,o}$, and representative genotypes were sampled with Bernoulli distribution for each region and perturbation. We aggregated the data across 324 nodules into 256 possible genotype states, which allows us to compute medians and CIs for marginal integration numbers (indicating the count of different plasmids integrated) and individual perturbation for frequencies as well as frequencies of individual genotypes.

Although it may be tempting to interpret high frequencies of genotype occurrence as advantageous for tumour proliferation, such raw frequencies could be confounded by technical factors such as initial plasmid concentration and integration rate. To address this, we constructed a null hypothesis (H0) over the 256 individual genotype numbers. This hypothesis holds the marginal expected number of integrations and perturbation frequencies constant across the population, attributing variations solely to technical effects, and assumes that the genotypes are independently distributed.

In practice, we adjust the observed perturbation frequencies to account for technical biases by normalizing these frequencies—dividing the average observed perturbation frequency for each perturbation by the sum of all plasmid frequencies and multiplying by the expected number of integrations. We then simulate the distribution of genotypes by drawing Bernoulli samples using these rescaled probabilities for each perturbation. This process is repeated for the number of nodules (324) and the results are aggregated back into the 256 genotype states to create a sampling strategy that reflects the desired properties. By comparing deviations between 5,000 samples drawn from both the inferred posterior (observed) and the simulated H0 (expected), we can identify genotypes with significant tumorigenic effects (observed > expected) or disadvantageous effects (observed < expected).

**Co-occurrence ORs and model of second-order interaction effect.** With posterior estimates of the genotypes within the tumour population, we can test for interaction effects between individual perturbations. Wrange of variables accountinge categorize the frequencies of perturbations A and B into four groups: $p_{00}$ (A−B−), $p_{01}$ (A− and B+), $p_{10}$ (A+B−) and $p_{11}$ (A+B+). The system can be described using a softmax linear model expressed as:

$$p_{i,j} = \exp(\theta_{00} + i\theta_{10} + j\theta_{01} + ij\theta_{11})/\sum_{ij} \exp(\theta_{00} + i\theta_{10} + j\theta_{01} + ij\theta_{11})$$

Here, $\theta_{00}$ and $\theta_{01}$ represent the effects of individual perturbations and $\theta_{11}$ is the interaction effect. By setting $\theta_{00}$ to zero to eliminate softmax non-identifiability and using $Z$ as the normalization constant $\Sigma_{i,j} e^{\theta_{i,j}}$, we derive the following relationships:

$$p_{10}/p_{00} = [e^{\theta_{10}}/Z]/[1/Z] = e^{\theta_{10}}$$

similarly,

$$p_{01}/p_{00} = e^{\theta_{01}}$$

and

$$p_{11}/p_{00} = e^{\theta_{10} + \theta_{01} + \theta_{11}}$$

Thus, computing the pairwise odds ratios (OR) effectively determines the interaction effect $\theta_{11}$:

$$OR = p_{11}p_{00}/p_{10}p_{01} = p_{11}p_{00}/[p_{10}/p_{01}][p_{01}/p_{00}]$$
$$= e^{\theta_{10} + \theta_{01} + \theta_{11}}/e^{\theta_{10}} e^{\theta_{01}} = e^{\theta_{11}}$$

For each gene pair, we estimated the ORs and assessed their significance by drawing 2,000 samples from the posterior probability of perturbation presence for each nodule. By fitting a softmax linear model directly to the data and setting the interaction effect $\theta_{11}$ to zero (OR = 1), we simulated the expected probabilities for the pairwise groups under the assumption of no interaction.

## Genotype-to-phenotype GLM

To explore the relationships between inferred perturbation probabilities and phenotypic features—specifically, IHC staining status and gene expression—we employed a GLM model. Here the inferred perturbation probabilities serve as the explanatory variables $X$.

For the IHC staining analysis conducted at the nodule level, we used binary annotations (positive/negative) and modelled the outcomes with a Bernoulli distribution. The staining status for each nodule, $Y_{r,m,k}$ ($r$, region; $m$, gene; $k$, sample) is modelled as:

$$Y_{r,m,k} \sim \text{Bernoulli}(\sigma(\sum_g X_{r,g} w_{g,m}) + z_k)$$

where $\sigma(x) = 1/1 + e^{-x}$ is the sigmoid link function. The weight matrix $w_{g,m}$, akin to L1 regularization, is sampled from a Laplace distribution centred at zero with a scale parameter $b$. We set the scale to one for the intercept and 0.1 for the perturbation weights to impose stronger regularization on the perturbations. The batch effect $z_k$ for each sample $k$ is also sampled from a Laplace distribution (0, 1). The explanatory variable $X_{rg}$ is directly sampled from $1 - F_{r,g,0}$, estimated by the perturbation probability model (non-learnable in the GLM).

Gene expression is modelled similarly, with few key differences. As gene expression is recorded as a non-zero integer at the spot level $s$, we use a Poisson distribution:

$$Y_{s,m,k} \sim \text{Poisson}(\mu_s \exp(\sum_g X_{s,g} w_{g,m} + z_k))$$

In addition to the parameters used in the IHC model, spot sensitivity $\mu_s$ is factored in, sampled from the posterior of the perturbation probability model. $X_{s,g}$ is calculated as $\sum_r A_{s,r}(1 - F_{r,g,0})$. The weight matrix $w_{g,m}$ for perturbation-related weights is sampled from a Laplace distribution centred at zero with a strongly regularizing scale $b = 1 \times 10^{-3}$.

## GLM inference

We infer our Bayesian model using a mean field variational posterior approximation. The model and its inference framework are implemented in Pyro (v1.8.6)[75]. The variational approximation is conducted via the stochastic variational inferederive the following relationships:nce method[76,77], employing the Adam optimizer set at a learning rate of 0.01 and using three samples for Kullback–Leibler divergence estimation. At each gradient descent step, the parameters $X$ and $\mu$ are sampled from their respective posterior distributions, as estimated by the perturbation probability model. This approach integrates the uncertainties associated with their estimation directly into the GLM framework. We perform inference over 2,000 gradient steps, monitoring the evidence lower bound to assess convergence.

## Statistical analysis of the leave-one-out experiment

The leave-one-out experiment (Fig. 7) evaluates whether the removal of the previously identified epistatically interacting perturbation, VEGFA or mtCTNNB1, from the eight-plasmid mix affects the formation of HCC or CCA. Three setups were tested: (1) all eight perturbations,

(2) all perturbations minus VEGFA and (3) all perturbations minus mtCTNNB1. For each setup, four mice were used. Tumours were identified and counted on two independent liver sections per mouse, with H&E histopathology and CK19 staining employed to differentiate between HCC and CCA subtypes. To assess group differences using a commonly applied approach, we performed a two-sided Kruskal–Wallis test, followed by Dunn's post-hoc test with Holm–Bonferroni correction for multiple testing. We further used an alternative approach to model tumour burden across experimental groups while accounting for biological variability and applied a Bayesian hierarchical Poisson regression.

This statistical model aims to estimate the occurrence rate of HCC or CCA types across different experimental conditions, accounting for group-level and animal-specific variability. Tumour counts, $y_i$, observed for each slide $i$ are modelled as Poisson-distributed outcomes. The rate parameter $\mu_i$ for each observation depends on the experimental condition and animal-specific effects. Group-level experiment effects ($\beta_g$) are drawn from a normal distribution $\beta_g \sim N(0, \sigma_\beta)$, where variability controlled by the hyperprior $\sigma_\beta \sim \text{HalfCauchy}(1.0)$. Animal-specific random effects ($\alpha_a$) account for individual variability and are sampled from $N(0, \sigma_a)$, with $\sigma_a \sim \text{HalfCauchy}(0.05)$. The latter is assumed to be less strong than the group-level effect, which is reflected in a more regularised hyperprior. The model specifies the expected log-counts as $\log(\mu_i) = \beta_g + \alpha_a$, where $g$ and $a$ index the group and animal associated with observation i. The tumour counts are then sampled from a Poisson distribution, $y_i \sim \text{Poisson}(\mu_i)$.

The posterior parameter distribution was estimated using 3,000 Markov chain Monte Carlo samples. The significance of differences between experimental effects was assessed by comparing the group-level parameters ($\beta_g$) across groups. P values were computed as the proportion of posterior samples where the differences in $\beta_g$ exceeded or fell below zero, depending on the direction of the expected effect. Model-derived estimates corroborated the findings of the non-parametric Kruskal–Wallis analysis.

**Reading Visium space ranger output into data objects.** We utilized the anndata package (v0.11) in Python and the SingleCellExperiment package (v1.24.0) in R to generate and manage 10X Visium data objects. To facilitate communication between R and Python, we employed ZellKonverter (v1.12.1). For data processing, we employed Scanpy (v1.9.8), squidpy (v1.4.1) and scater (v1.30.1) in Python and R[76,78,79]. We refined our analysis by subsetting all objects to include only features shared across all 11 slides, resulting in a total of 19,464 genes.

**Publicly available databases of cell-type markers.** We used scLiverDB, PanglaoDB and MSigDB to collect an initial set of marker genes for prevalent cell types and gene sets in normal and tumour liver tissues of mouse and human[80–82]. This yielded a list of a total 2,323 genes.

**Data normalization and preprocessing.** After applying filtering criteria to exclude genes with raw counts of <10 or >$1 \times 10^6$ for any individual slide, as well as barcode genes, the count matrices were normalized to each spot to ensure a total count of $1 \times 10^4$. Subsequently, the normalized values were log-transformed ($\log(x + 1)$). This preprocessing was executed in Python using Scanpy (v1.9.8). Utilizing Scanpy (v1.9.8) with the Seurat flavour, we identified 15,000 highly variable genes for each of the 11 Visium and Visium CytAssist slides. The intersection of these sets resulted in 9,205 genes. Subsequently, in the final refinement step, we narrowed the gene set down to 80 core markers, resulting in 7,361 genes. This final gene set was employed for all subsequent phenotype analyses and visualizations.

**Gene coexpression networks.** Using the spot level-normalized expression values after filtering out uninformative genes, we conducted Gaussian graphical modelling[83] to infer a sparse gene coexpression network. We utilized the R package glasso (v1.11) for this purpose. The regularization parameter was optimized through a grid-search approach and set to 0.3. Following the construction of the initial Gaussian graphical modelling, we refined the network by filtering edges to retain only those with a Pearson's pairwise correlation coefficient of at least 0.25. The isolated genes were, in turn, dropped from the graph. We used the R package igraph (v2.0.1.1) to visualize the graphs.

**Nodule-level expression aggregation.** We computed two types of aggregates for normalized expression values within each nodule: mean-based and quantile-based. For the mean-based aggregates, we calculated the average normalized expression of each gene across all spots within each nodule. For the quantile-based aggregates, we determined $q_{95}$ of expression values across all spots per nodule. We used these aggregates in the subsequent nodule-level analyses.

**Binarization of nodule phenotypes.** After quantile normalization, we computed the average of the scaled expression values for the core markers per phenotype, we then thresholded the values by 0.5, where all nodules with an aggregate value of >0.5 are considered to have the corresponding phenotype signature and otherwise not. The mean-based aggregates for each gene are quantile-normalized further by

$$x - q_{25}/q_{99} - q_{25}$$

where $q_{25}$ and $q_{99}$ represent the 25th and 99th quantile values of the aggregate gene expression for the corresponding gene across all nodules and all slides. We then binarized the values >0.5 as one (on) and otherwise zero (off).

**Binarization of nodule TME signatures.** We binarized the TME signatures following the same procedure as for nodule phenotypes but using the initial quantile-based aggregates instead.

**Phenotype and TME heatmaps.** We generated heatmaps of scaled gene expression using the processed expression values at the nodule level, employing ComplexHeatmap (v2.16.0)[84]. Clustering of both rows and columns was performed based on Spearman's correlations. In addition, hierarchical clustering was applied to subdivide genes into clusters. The colour bar associated with genes indicates their corresponding phenotypes, with emphasis on the core markers. An attached annotation heatmap illustrates scaled ($p^{10}$) estimated plasmid probabilities per nodule.

**Spatial integration and nodule unification.** To integrate all slides into a unified embedding space and classify spots based on their phenotypic signatures in an unbiased manner, we employed an ensemble spatially-aware classifier implemented in SageNet (v1.1.0)[85]. Data from all slides were trained and fed into this classifier. Subsequently, we clustered the spots within the embedded space using Scanpy's wrapper of Leiden clustering (with a resolution of one)[86]. We then performed voting classification to classify nodules to the most-dominant class across the spots belonging to the corresponding nodules. We call these classes the 'unified nodule annotations'. Finally, we extracted spatially informative genes from the SageNet model.

**Inter-nodule differential gene expression.** To delve deeper into inter-nodule transcriptional differences, we conducted differential gene expression analysis using the FindMarkers method from the R package scran (v1.28.2). This method allowed us to perform a light-weight differential gene expression analysis on the unified nodule annotations.

**Cell type-informed factor analysis.** We concatenated all raw anndata objects and subsetted them to the set of 'core markers' and associated genes (as listed in Fig. 5) as well as 500 highly variable genes across slides.

We then used cell2module (github.com/vitkl/cell2module) to perform non-negative matrix factorization. The cell2module model treats raw RNA count data $D$ as negative-binomial (NB) distributed, given transcription rate $\mu_{c,g}$ and a range of variables accounting for technical effects:

$$D_{c,g} \sim \text{NB}(\mu = \mu_{c,g}, \alpha_{a,g})$$

and

$$\mu_{c,g} = (\sum_f w_{c,f} g_{f,g} + s_{e,g}) y_c$$

where $\mu_{c,g}$ denotes expected RNA count $g$ in each cell $c$, $\alpha_{a,g}$ denotes the per gene $g$ stochastic/unexplained overdispersion for each covariate $\alpha$, $w_{c,f}$ denotes cell loadings of each factor $f$ for each cell $c$, $g_{f,g}$ denotes gene loadings of each factor $f$ for each cell $c$, $s_{e,g}$ denotes additive background for each gene $g$ and for each experiment $c$ to account for contaminating RNA and $y_c$ denotes normalization for each spot $c$ to account for RNA-detection sensitivity, sequencing depth. We recovered 40 factors representing groups of coexpressing cell-type signatures using the default cell2module parameters. After training, we inferred the posterior of the gene loadings per factor. Subsequently, we extracted genes with the top five posterior median values and compared them with predefined marker gene lists per cell type. Finally, we mapped each factor to the cell type with the highest number of overlapping genes.

### Reporting summary

Further information on research design is available in the Nature Portfolio Reporting Summary linked to this article.

### Data availability

We used publicly available datasets from scLiverDB (https://guolab. wchscu.cn/liverdb#!/), PanglaoDB (https://panglaodb.se/), MSigDB (https://www.gsea-msigdb.org/gsea/msigdb), GEO (https://www.ncbi. nlm.nih.gov/geo/), MGI (https://www.informatics.jax.org/) and the LiverCellAtlas (https://www.livercellatlas.org/). We deposited data related to this manuscript to https://zenodo.org/records/10986436 (ref. 87). In addition, we have launched a web browser for interactive data analyses (https://chocolat-g2p.dkfz.de/). Source data are provided with this paper.

### Code availability

Scripts and custom code for data analysis related to this manuscript are available at https://github.com/gerstung-lab/CHOCOLAT-G2P/ (ref. 88).

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

## Acknowledgements

We thank all of the members of the Tschaharganeh, Sahm, Gerstung and Stegle laboratories. We thank the German Cancer Research Center Central Animal Laboratory, the Center for Model Systems and Comparative Pathology of the Institute of Pathology Heidelberg and the Tissue Bank of the National Center for Tumor Diseases for excellent technical support. D.F.T. was supported by the European Research Council (ERC Starting Grant 'CrispSCNAs', grant number 948172) and the Deutsche Forschungsgemeinschaft (DFG; grant number TS 293/3-1).

## Author contributions

M.B. conceptualized the project, and designed the methodology and experiments. L. Böse, L. Butthof and L.W.-L. cloned plasmids, and performed animal experiments and IHC, with help from M.B. M.R. performed 10X Visium spatial transcriptomics and next-generation sequencing with help from M.B. P.S., D.F.T. and F.S. provided reagents. T.P. supervised the technical processing of FFPE liver samples and preparation of IHC stainings. H.W. and D.F.T. performed the histopathological analyses. H.W. and M.B. performed the spatial transcriptomics-guided nodule annotation. M.B. performed the initial bulk RNA-seq data analysis for probe redeployment. E.H. and A.L. performed 10X Visium data preprocessing. A.L. implemented spatial transcriptomics-based genotyping, with input from E.H., M.B., O.S. and M.G. E.H. carried out spatial transcriptomics-based phenotyping, with input from A.L., M.B O.S. and M.G. A.L. and E.H. performed statistical analyses, with input from O.S. and M.G. M.B., A.L. and E.H. created figures, with input from D.F.T., O.S. and M.G. A.L. and G.R. developed the dedicated webtool interface, with input from E.H. and M.B. M.B., A.L. and E.H. wrote the manuscript, with input from D.F.T. and M.G. All authors read and approved the final paper.

## Funding

## Competing interests

All authors declare no competing interests.

## Additional information

**Extended data** is available for this paper at https://doi.org/10.1038/s41551-025-01437-1.

**Correspondence and requests for materials** should be addressed to Marco Breinig, Moritz Gerstung or Darjus F. Tschaharganeh.

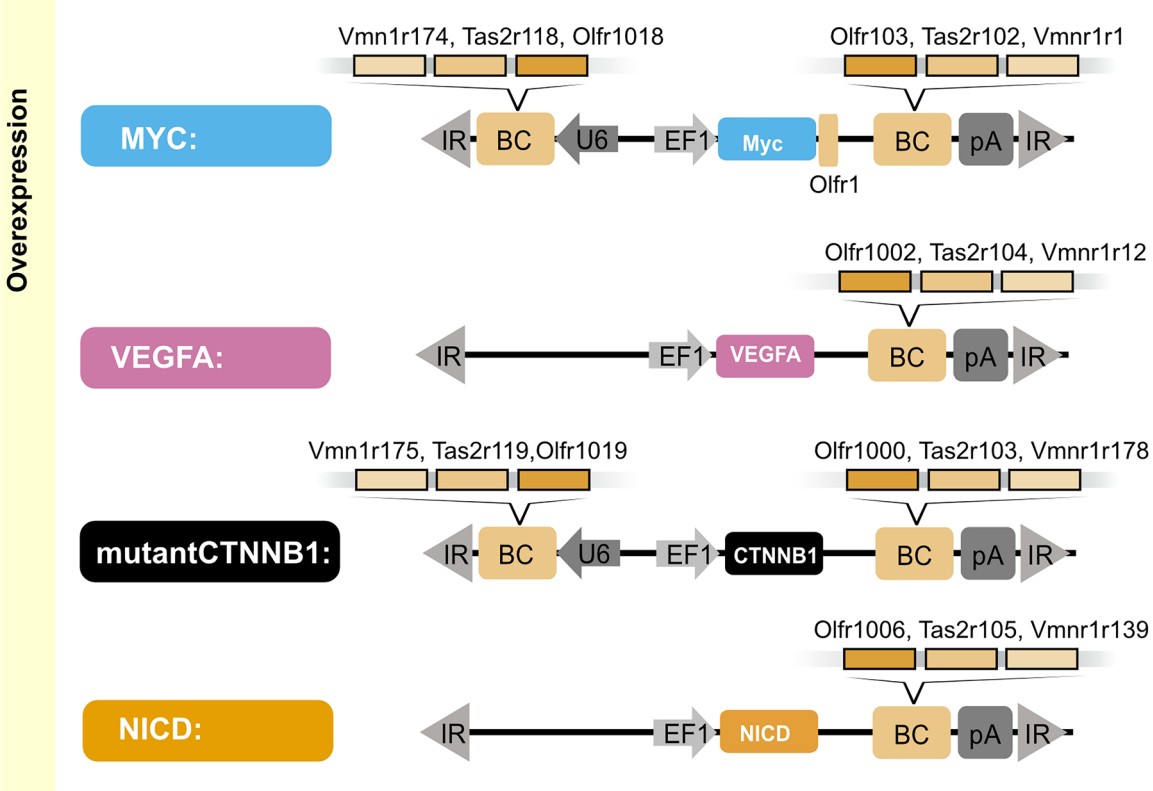

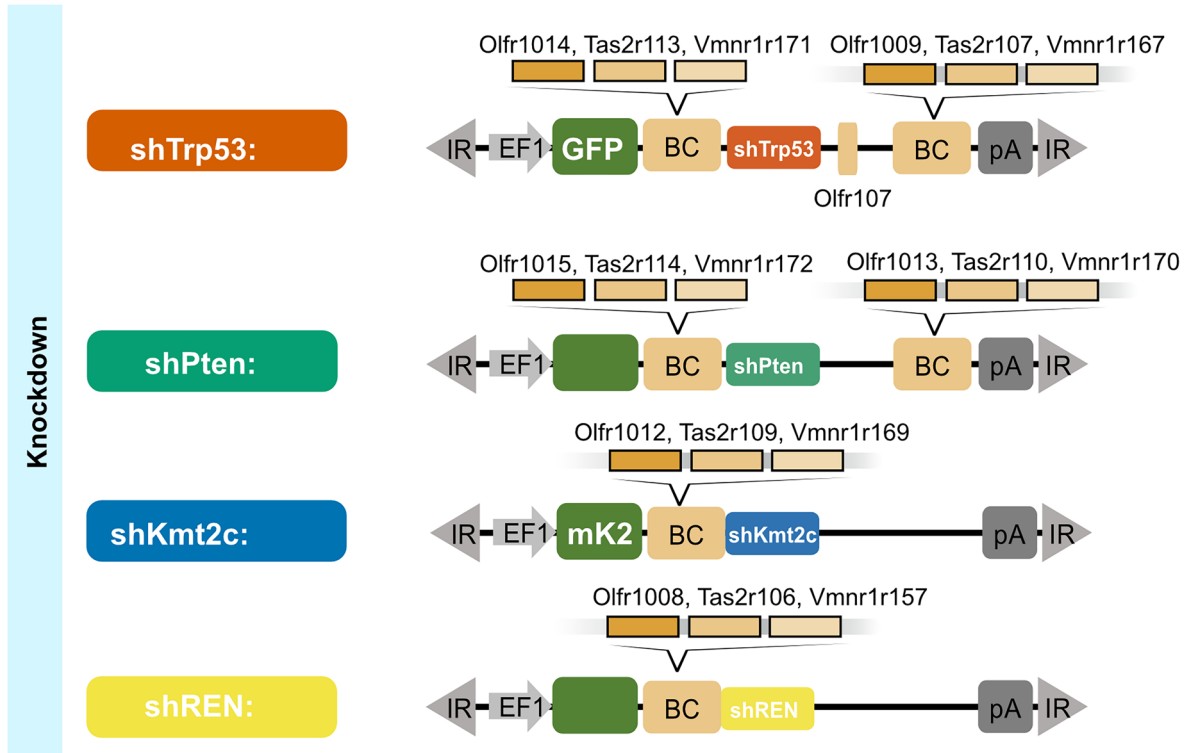

**Extended Data Fig. 1 | See next page for caption.**

**Extended Data Fig. 1 | Overview of perturbation plasmid design and molecular barcoding.** Schematic overview of Sleeping Beauty transposon perturbation plasmids to ectopically overexpress genes-of-interest (oncogenic-driver perturbations) or shRNA to enable gene knockdown (tumor-suppressor perturbations). Functional elements are highlighted. IR: Inverted/direct repeats of sleeping beauty transposon; GFP: green fluorescent protein, mK2: mKate2 red fluorescent protein, EF1: Polymerase II promoter; U6: Polymerase III promoter; pA: polyadenylation signal; sh: short hairpin RNA embedded in mir-E context; BC: a barcode in which 3 redeployed RTL-probe capture sequences (as indicated) are embedded. Note that we used Visium Mouse Transcriptome Probe Set v1 to derive barcodes. Each 50 nt barcode is separated and flanked by ca. 20 nt spacer sequences to avoid potential steric hindrance during hybridization. Spacer sequences used were derived from T7 and T3 promoters and/or AsCas12a-DR sequences and/or 10X Capture sequences cs1 and cs2 (not shown). Functionality of spacer sequences was not tested in this study. Note that plasmids were equipped with multiple orthogonal barcodes at varying positions. See Methods for further information.

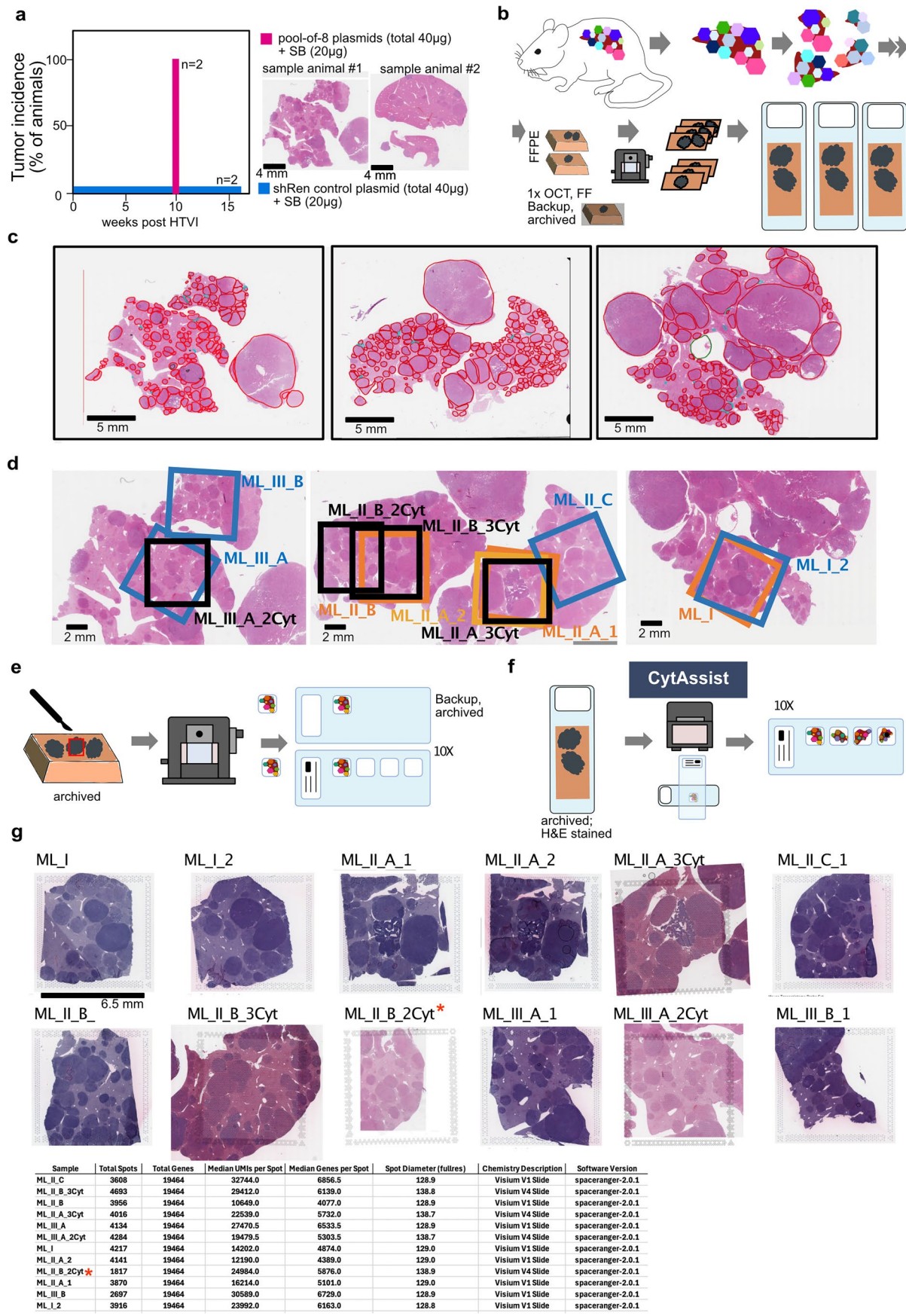

**Extended Data Fig. 2 | See next page for caption.**

**Extended Data Fig. 2 | Sample processing and ROI selection for spatial transcriptomics. a**, RUBIX with a pool-of-8 plasmid mix results in rapid liver tumour development. Injection of a shRNA targeting *Renilla* (shRen; matching total plasmid concentration for pool-of-8 mix) served as control, *n* = 2 each group. Representative H&E-stained samples revealing multiple tumour nodules from the two individual animals are shown. Absence of tumours in the control group (shRen only) indicates that random integration of transposon plasmids itself is unlikely to contribute to tumorigenesis. **b**, Tissue preprocessing. Following liver tumour development, livers were extracted, divided and processed to FFPE as well as fresh frozen specimens. FFPE samples were initially sectioned to enable sample selection. **c**, C-G2P liver samples. Overview of three representative FFPE samples used in this study. A total of 513 tumour nodules (red outline) were identified based on histopathological examination (based on H&E). **d**, Overview of ROIs selected for 10X Visium. 6 segregated regions were

selected across 3 FFPE samples. Squares indicate approximate position of ROIs selected for ST. Orange: first 10X Visium run, light orange: first 10X Visium run, replicate ROI; blue: second 10X Visium run; black: 10X Visium CytAssist run. Note overlap between ROIs, where serial sections are used for 10X Visium. **e**, 10X Visium workflow. Samples for ST are derived directly from FFPE blocks and mounted on 10X Visium slides. **f**, 10X Visium CytAssist workflow. Samples for ST are derived from sections already mounted on glass slides and transferred to 10X Visium slides using the 10X CytAssist instrument. **g**, Overview of all samples used for ST. 12 samples from a single RUBIX experiment with two animals were used for 10X Visium in this study. Respective H&E stainings are depicted. Note that the utility of sample ML-II_B_2Cyt is constrained by tissue detachment of the sample during the processing for 10X Visium CytAssist and was not included for further analyses (asterisk). QC summary stats for each sample related to Visium runs performed are provided.

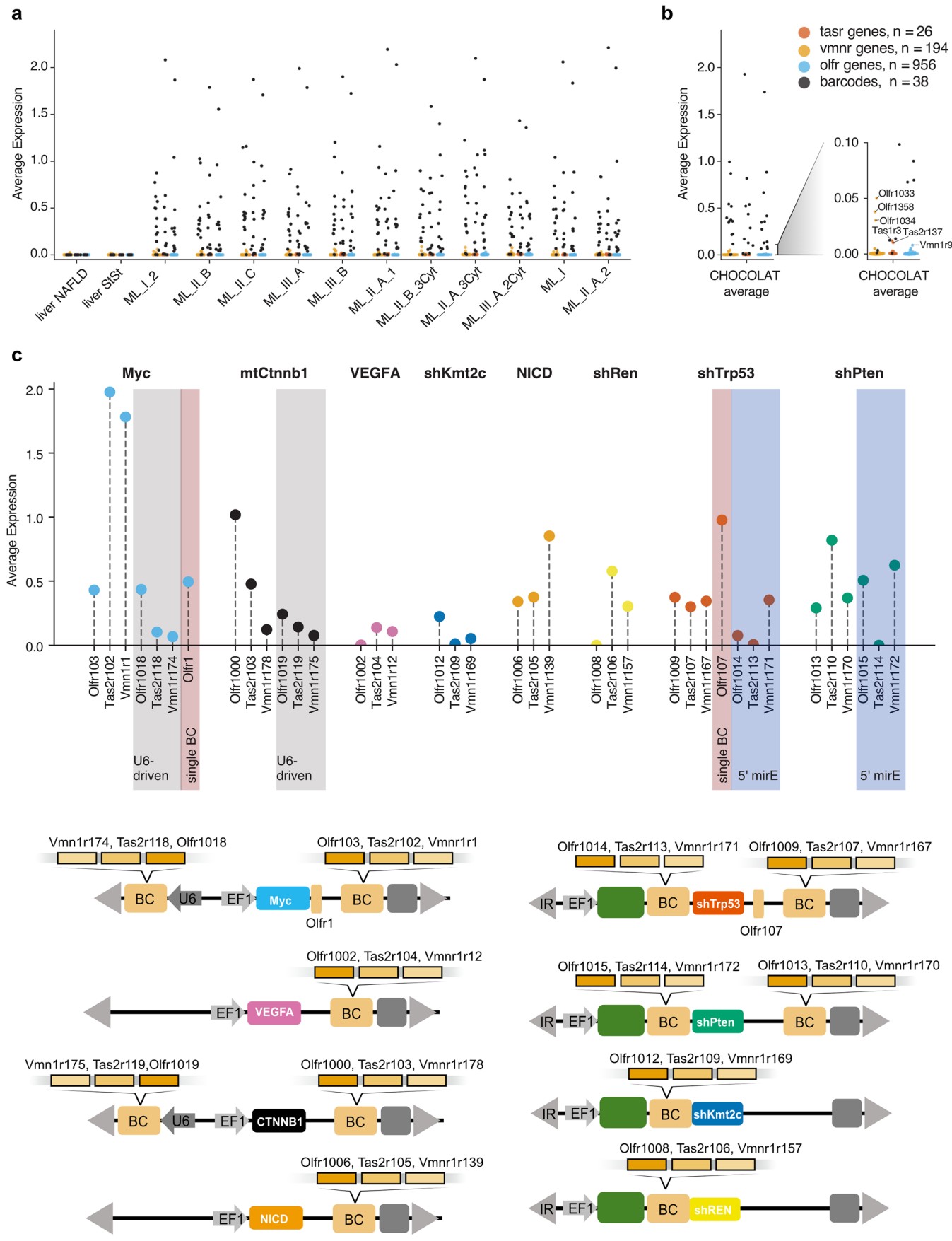

**Extended Data Fig. 3 | See next page for caption.**

**Extended Data Fig. 3 | Specific detection of redeployable barcodes by available 10X RTL probes. a**, Expression of all potentially redeployable barcodes derived from sensory receptor transcripts in murine liver. Transcripts associated with three groups of sensory receptors - Tasr (red; $n = 26$), Vmnr (yellow; $n = 194$), Olfr (blue; $n = 956$). Redeployed barcodes (black) are well separated from endogenous sensory receptor transcripts (total $n = 1,216$). Data for all 11 C-G2P samples generated in this study from a single RUBIX experiment with two animals is depicted. Reference control liver datasets (NAFLD, StSt) are from[17]. **b**, Sensory receptor-associated transcripts expressed in murine liver. Sensory receptor-associated transcripts that reveal average expression between <0.01 and >0.008

in murine liver are depicted. Expression values are aggregated across all 11 C-G2P samples shown in **a**. **c**, Expression of all 38 redeployed barcodes used in this study. Grouped according to associated perturbation (see Extended Data Fig. 1, Methods). Expression is averaged across 6 primary C-G2P samples. Note that 5/38 revealed insufficient signal (average expression < 0.05). In all panels, expression values are log1p-transformed. Barcodes expressed from a Pol III (hU6) promotor are highlighted. Barcodes expressed in 5′ mirE position for shRNA constructs are highlighted. Simplified plasmids maps are depicted to illustrate barcode positioning.

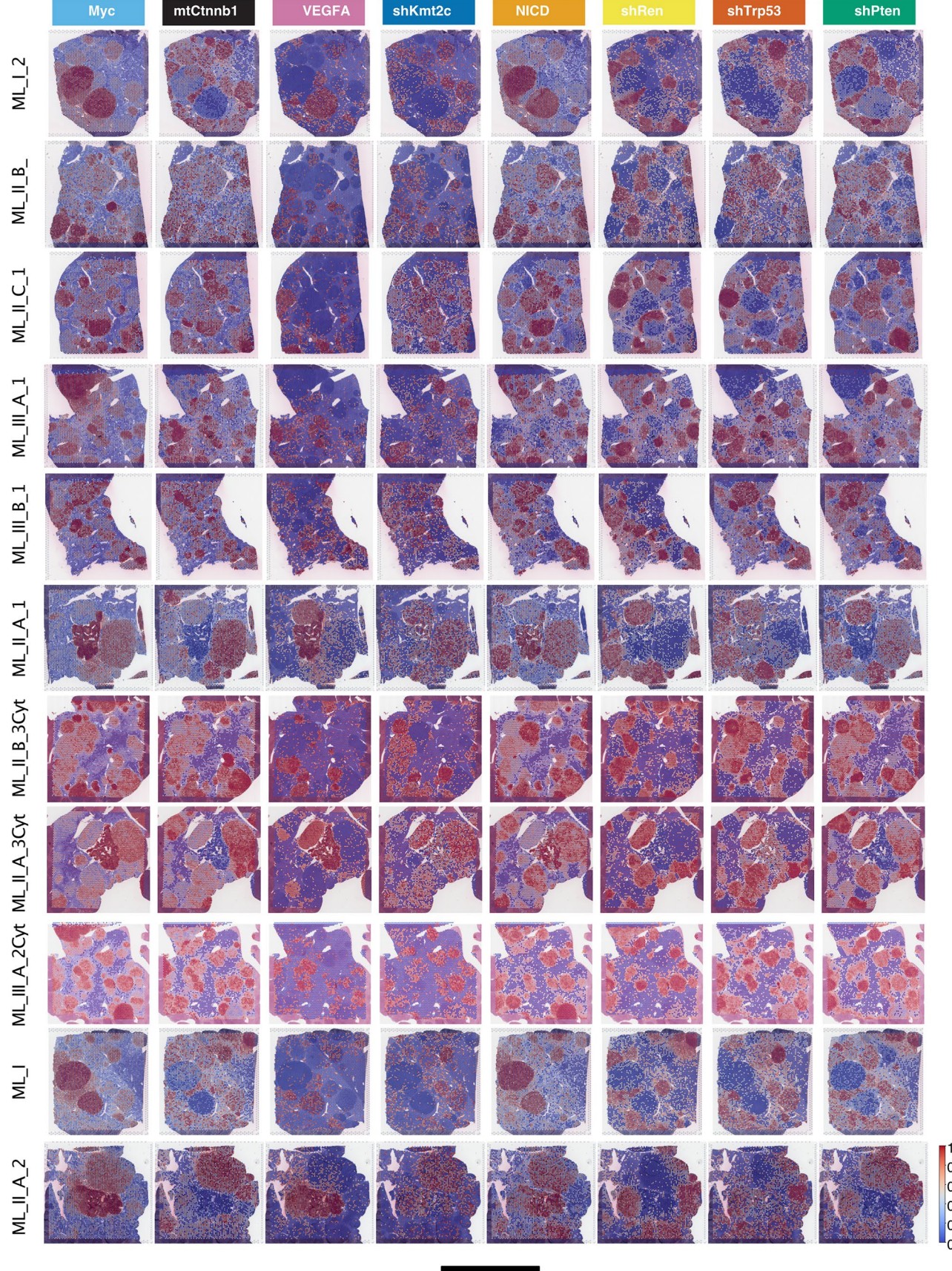

**Extended Data Fig. 4 | Spatially resolved triplet barcode expression enables identification of all 8 perturbations used.** Spatially resolved expression of triplet barcodes for each of the 8 perturbations across all 11 samples in this study from a single RUBIX experiment with two animals. Gene expression is log1p-transformed and quantile-rescaled, as in Fig. 3c. The quantitative barcode expression for all samples can be explored through the interactive web browser (https://chocolat-g2p.dkfz.de/).

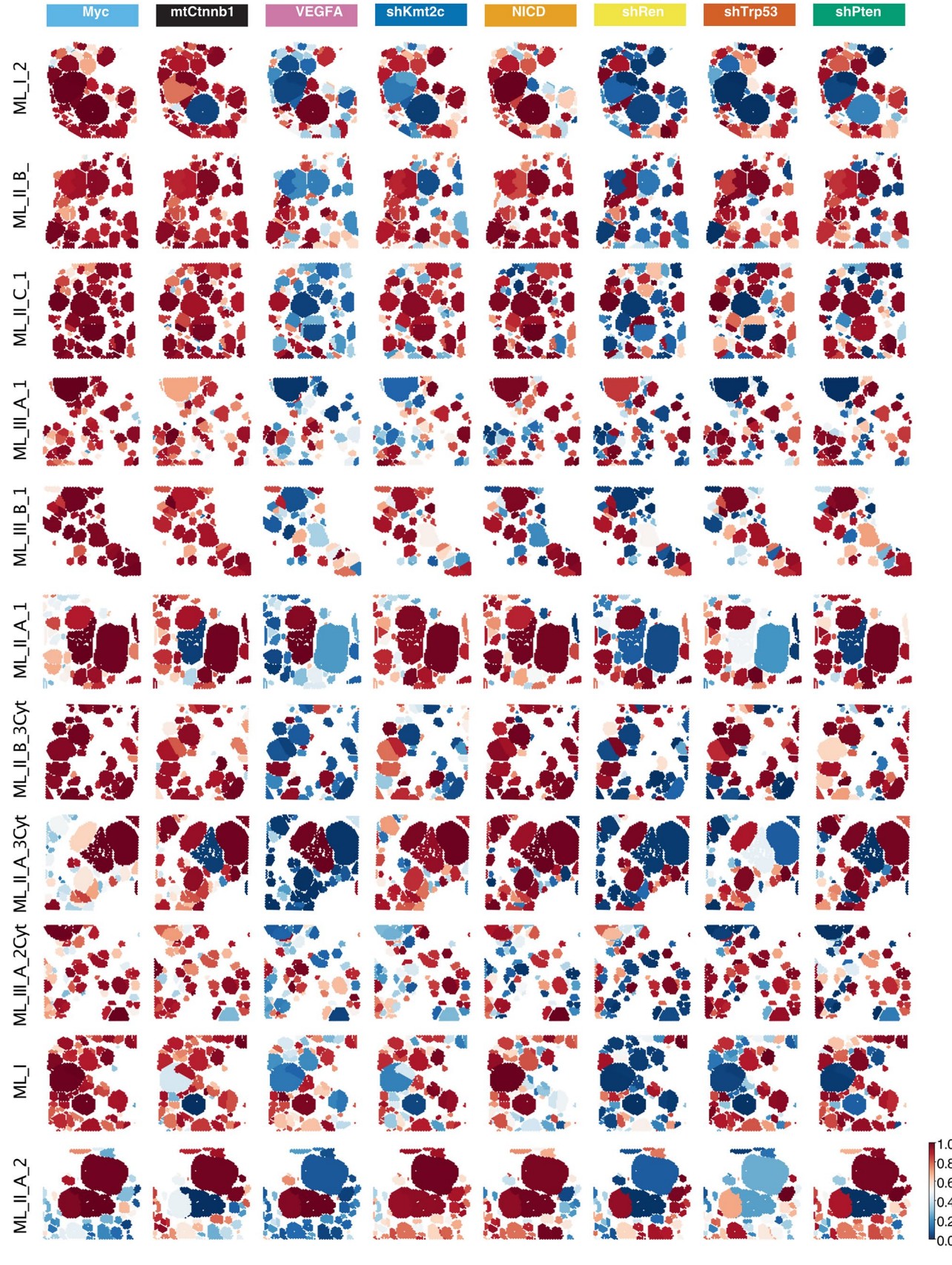

**Extended Data Fig. 5 | Spatially resolved perturbation mapping.** Spatially resolved visualization of the inferred probabilities indicating the presence or absence of each of the 8 perturbations associated with annotated tumour nodules for all 11 samples used in this study from a single RUBIX experiment with two animals (Methods), as in Fig. 3d. Perturbation probabilities for all samples can be explored through the interactive web browser (https://chocolat-g2p.dkfz.de/).

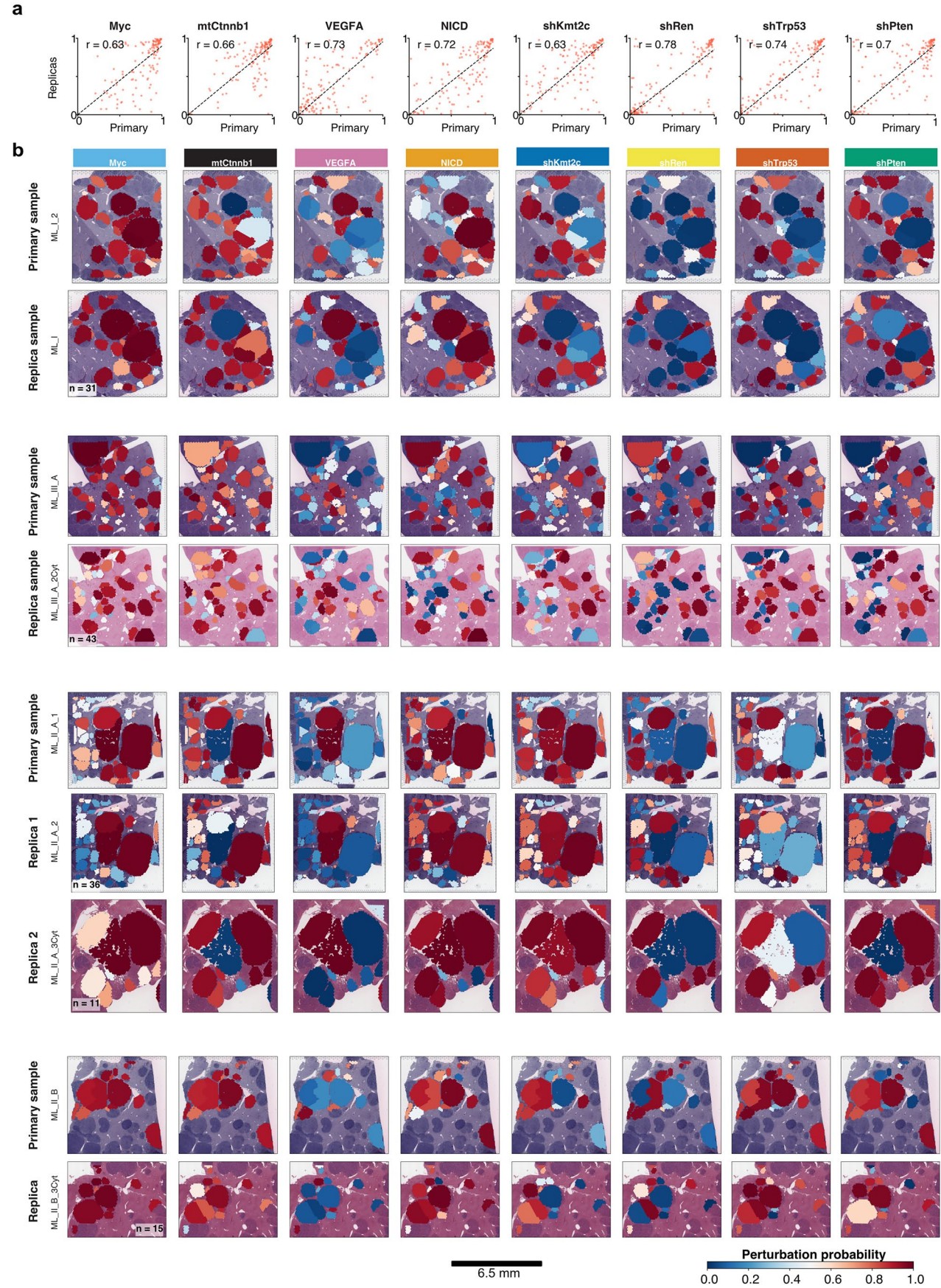

**Extended Data Fig. 6 | See next page for caption.**

**Extended Data Fig. 6 | Quantitative reproducibility of spatial perturbation mapping. a**, Quantitative reproducibility. Scatterplots of inferred perturbation probabilities for nodules on the primary section to those on the corresponding replica sections, with Pearson's correlation values displayed. In total, 136 nodule pairs were analysed. **b**, Matching nodules across samples. Spatial maps of the inferred probabilities (as in Fig. 3c) indicating the presence or absence of each of the 8 perturbations associated with annotated tumour nodules for all samples that have matching ROIs from a single RUBIX experiment with two animals. Matching nodules were manually annotated (Methods). Addition of "_Cyt" in sample name indicates use of 10X CytAssist. Number of matching nodules is indicated.

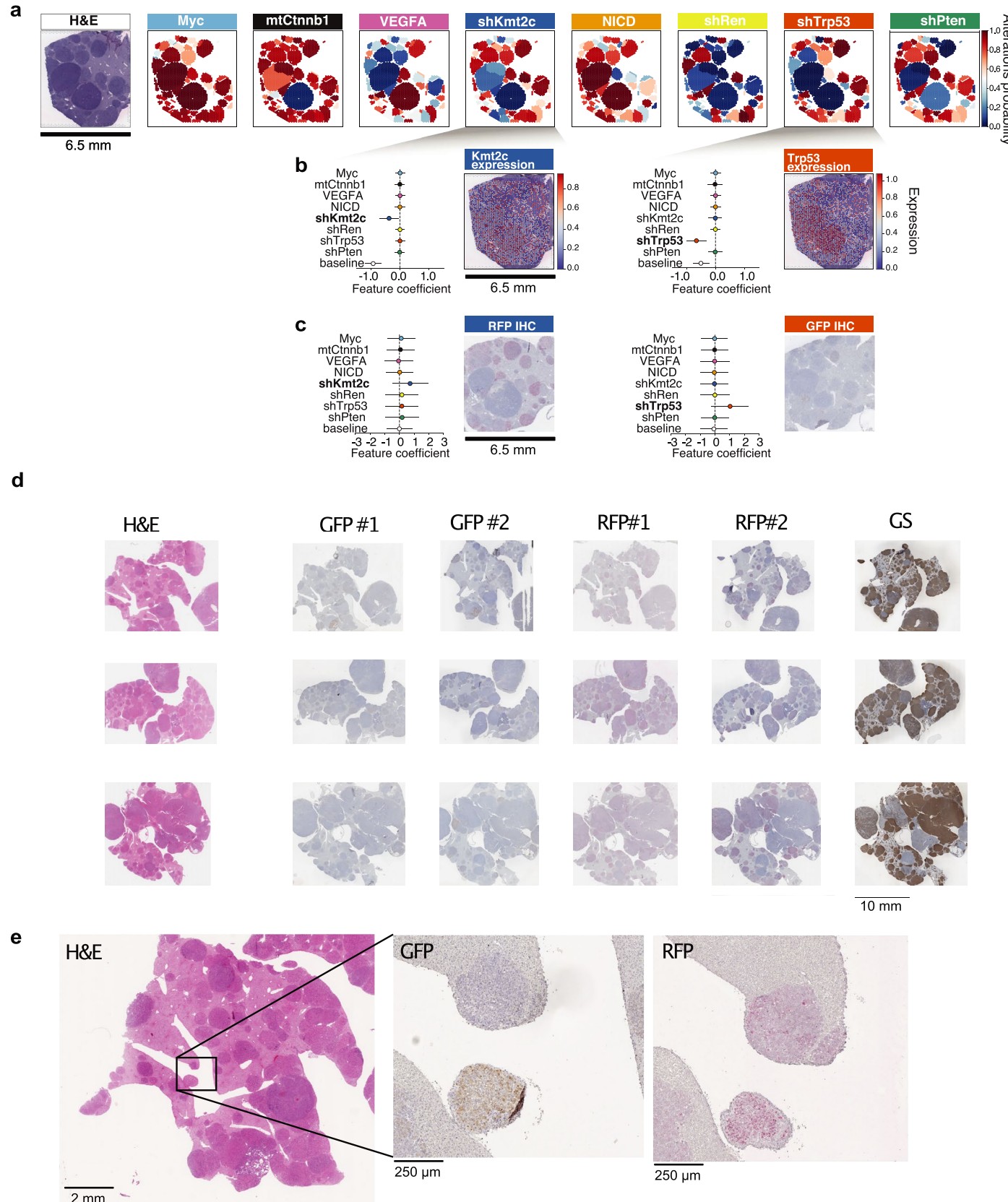

**Extended Data Fig. 7 | See next page for caption.**

**Extended Data Fig. 7 | Validation of inferred perturbation integration.**
**a**–**c**, Perturbation–phenotype association. Spatially resolved visualization of the inferred probabilities indicating the presence or absence of each of the 8 perturbations associated with annotated tumour nodules (Methods). **a**, A representative sample is displayed. A generalized linear model (GLM) predicts phenotype expression signals based on the estimated probabilities of perturbation presence (Methods). **b**, Phenotypes are defined as direct target transcripts associated with perturbations such as shKmt2c-Kmt2c and shTrp53-Trp53. Expression data are log1p-transformed. Note that shTrp53 is linked to a GFP peptide barcode and shKmt2c is linked to a RFP barcode (Extended Data Fig. 1). **c**, Hence we infer shTrp53-GFP-positive phenotype and shKmt2c-RFP-positive phenotype. Representative IHCs for a corresponding ROI on a serial section. Baseline depicts background phenotype marker expression. Data are presented as feature coefficients shown as mean and error-bars depict 3σ confidence intervals. As in Fig. 3c, and **d. d**, H&E and IHC for GFP, RFP, GS. Three

representative FFPE samples from a single RUBIX experiment with two animals were sectioned and stained for H&E (see Extended Data Fig. 2). GFP and RFP IHC staining was performed on two individual serial sections. GS IHC staining was performed on serial sections. GFP and RFP were embedded in perturbation plasmids as orthogonal barcodes (see Methods and Extended Data Fig. 1). GS is a well-known marker for liver WNT/mtCtnnb1-signalling activity (see Fig. 3d). **e**, GFP and RFP detection for the same hepatocellular tumour nodules to spatially map peptide barcode combinations associated with introduced perturbations. Left: H&E. Zoom-in: GFP and RFP respectively. Note that one tumour nodule is positive for RFP alone whereas the other tumour nodule is positive for both GFP as well as RFP peptide barcodes. A representative example is shown. Data is derived from 324 nodules across 6 topographically separated regions used for 10X Visium from a single RUBIX experiment with two animals. Mapping GFP and RFP IHC data is derived from 3 corresponding sections from a single RUBIX experiment with two animals.

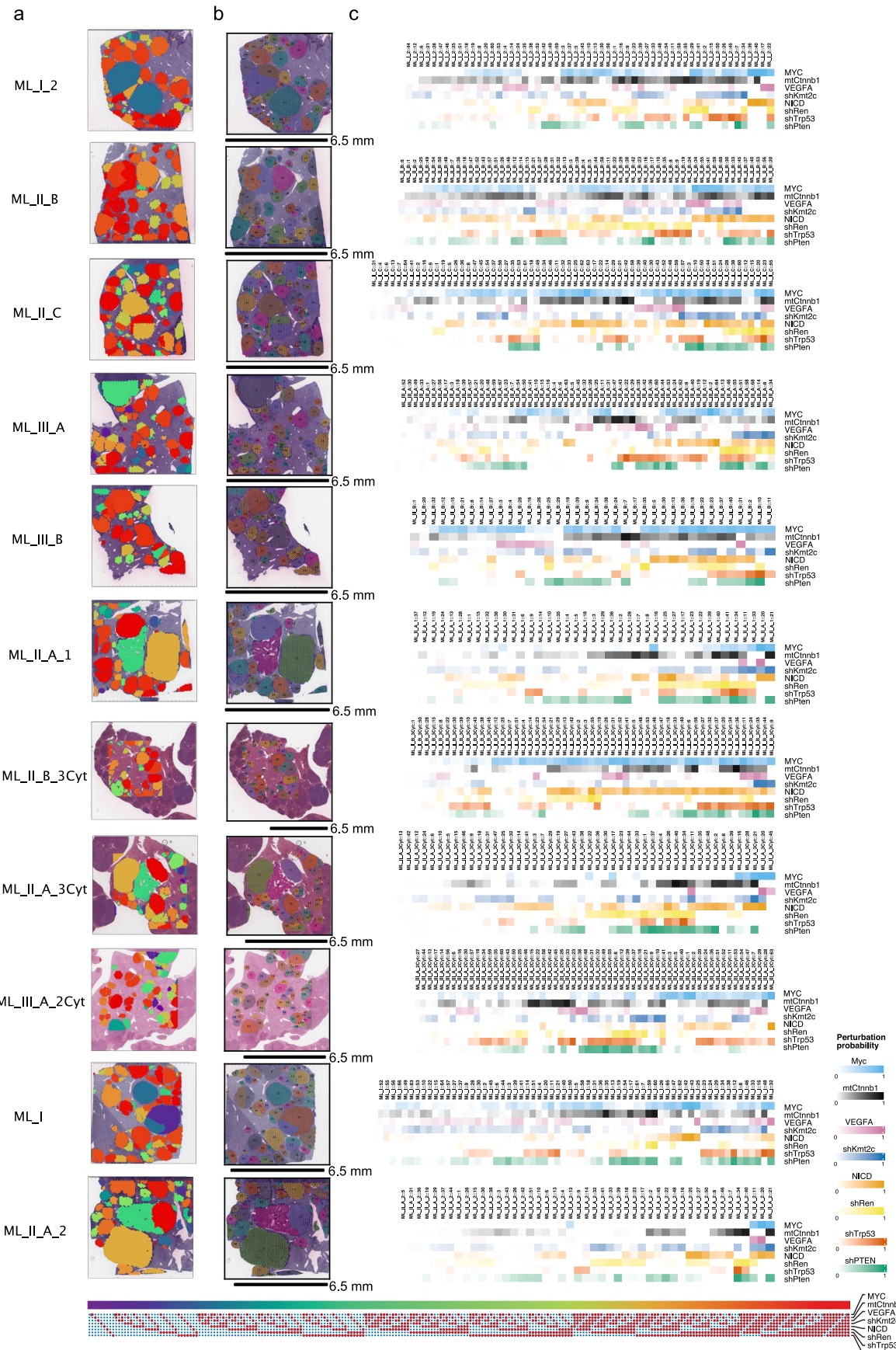

**Extended Data Fig. 8 | C-G2P enables spatial genotype mapping. a**, Converting barcode signals to genotype maps. $2^8$ powerset embedding of spatially mapped perturbations for all 11 samples used in this study from a single RUBIX experiment with two animals, encompassing 622 nodules. Each of the 256 genotypes are colour-coded. As in Fig. 4a. **b**, Spatially resolved nodule annotation. For all 11 samples used in this study. As in Fig. 3a. **c**, Tumour genotypes. Scaled ($p^{10}$) estimated plasmid probabilities per nodule. As in Fig. 5.

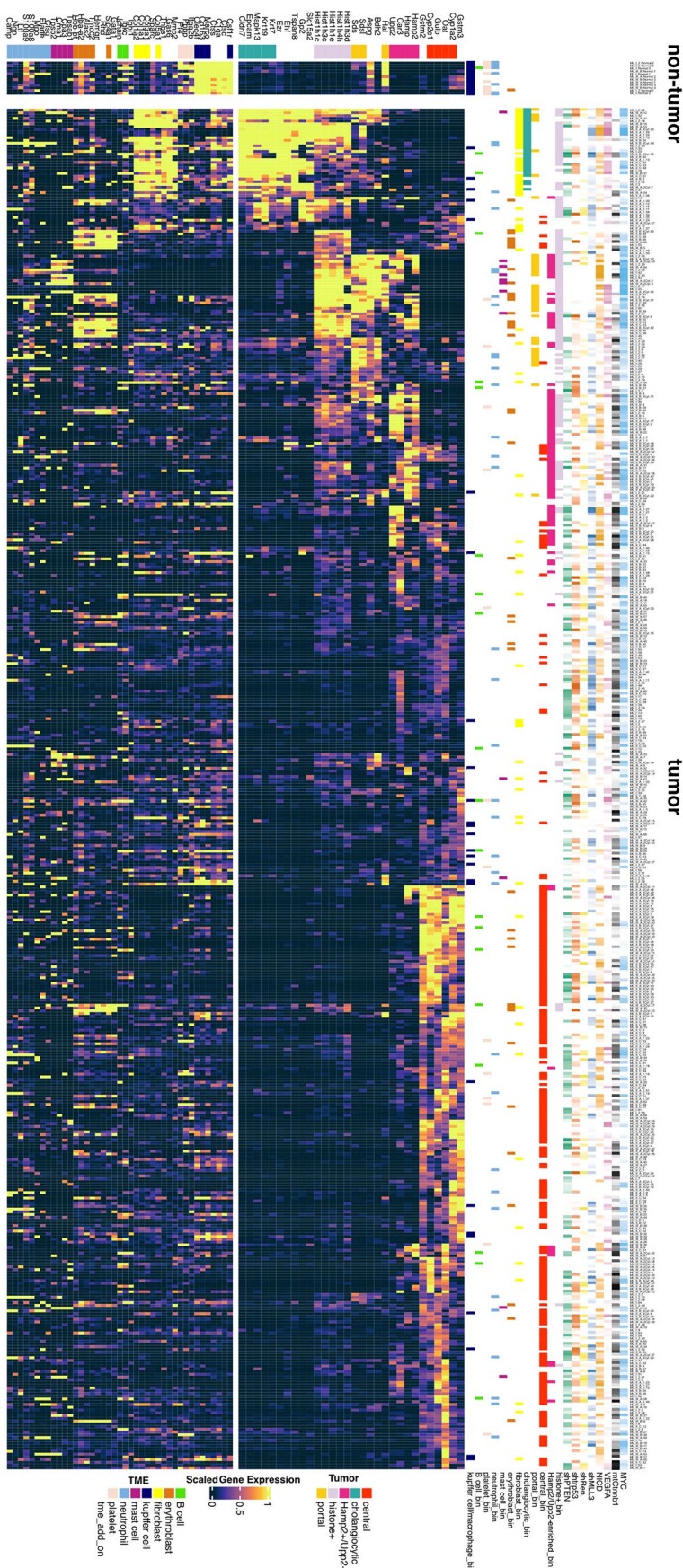

**Extended Data Fig. 9 | Overview of genotypes and phenotype-binarization for all nodules across 11 samples.** As in Fig. 5. Top panel: Scaled plasmid probabilities and per-nodule phenotype binarization as used in Fig. 6 (Methods). *n* = 622 total nodules from 11 samples used in this study from a single RUBIX experiment with two animals.

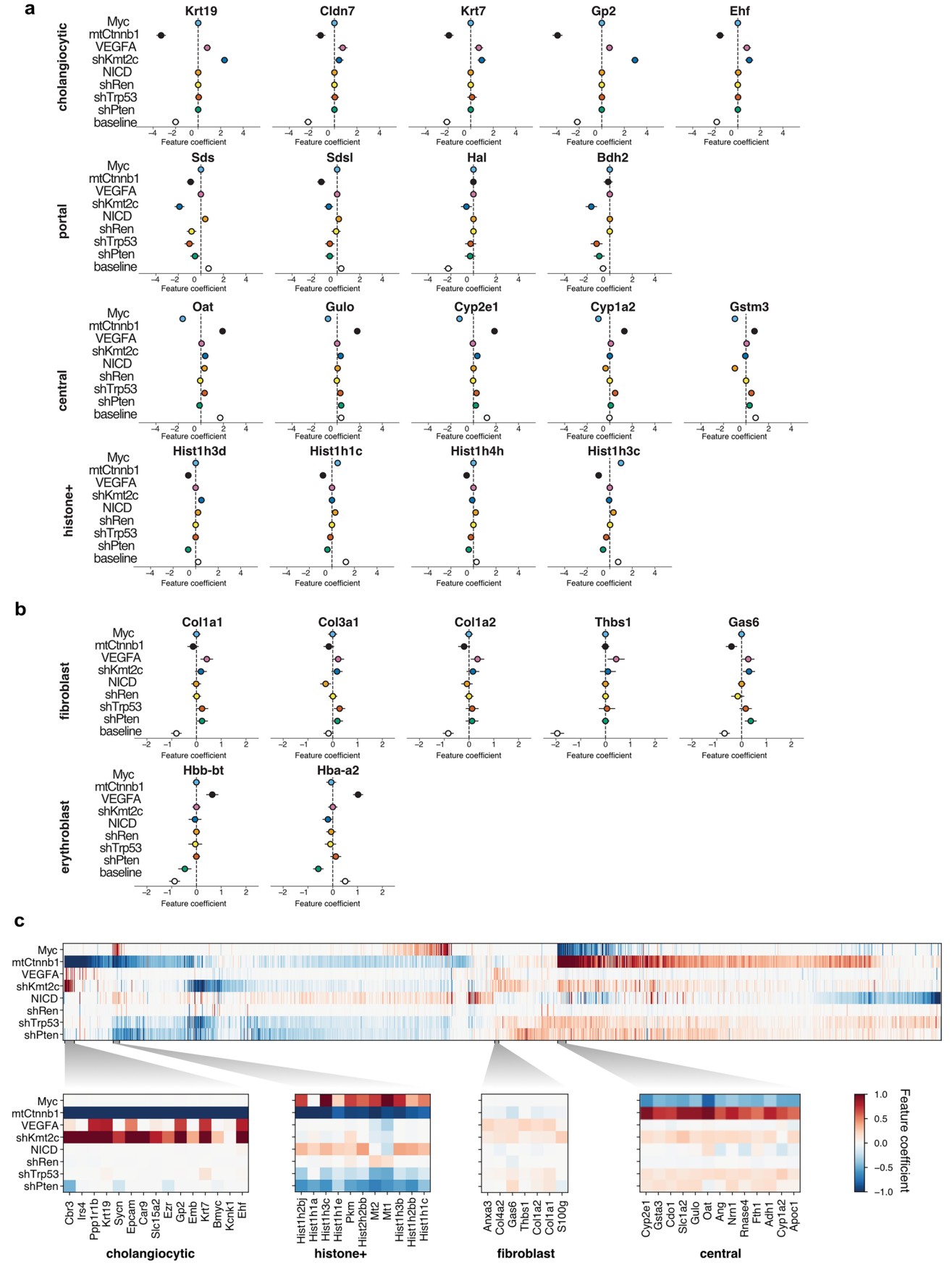

**Extended Data Fig. 10 | See next page for caption.**

**Extended Data Fig. 10 | Genotype–phenotype relations as evidenced by GLM. a**, Tumour-intrinsic genotype–phenotype relations. A generalized linear model (GLM) predicts gene expression signals at each 10X Visium spot, using estimated probabilities of perturbation presence (Methods). Data are presented as feature coefficients shown as mean and error bars depict 3σ confidence intervals. Feature coefficients indicate associations between gene expression and perturbations for representative transcripts of four tumour-intrinsic phenotypes. **b**, TME-related genotype–phenotype relations. As in **a** for representative transcripts of two exemplary TME phenotypes. **c**, GLM-inferred genotype–phenotype associations. Top: heatmap of 1,283 genes with at least one significant (3σ) feature weight, ordered by 1D UMAP embedding. Bottom: Detailed views of four representative clusters linked to marker genes of known phenotypic groups. Data is derived from 324 nodules across 6 topographically separated regions used for 10X Visium from a single RUBIX experiment with two animals.

# Reporting Summary

## Statistics

For all statistical analyses, confirm that the following items are present in the figure legend, table legend, main text, or Methods section.

| n/a | Confirmed | |
|---|---|---|
| ☐ | ☒ | The exact sample size (*n*) for each experimental group/condition, given as a discrete number and unit of measurement |
| ☐ | ☒ | A statement on whether measurements were taken from distinct samples or whether the same sample was measured repeatedly |
| ☐ | ☒ | The statistical test(s) used AND whether they are one- or two-sided *Only common tests should be described solely by name; describe more complex techniques in the Methods section.* |
| ☐ | ☒ | A description of all covariates tested |
| ☐ | ☒ | A description of any assumptions or corrections, such as tests of normality and adjustment for multiple comparisons |
| ☐ | ☒ | A full description of the statistical parameters including central tendency (e.g. means) or other basic estimates (e.g. regression coefficient) AND variation (e.g. standard deviation) or associated estimates of uncertainty (e.g. confidence intervals) |
| ☐ | ☒ | For null hypothesis testing, the test statistic (e.g. $F$, $t$, $r$) with confidence intervals, effect sizes, degrees of freedom and $P$ value noted *Give P values as exact values whenever suitable.* |
| ☐ | ☒ | For Bayesian analysis, information on the choice of priors and Markov chain Monte Carlo settings |
| ☒ | ☐ | For hierarchical and complex designs, identification of the appropriate level for tests and full reporting of outcomes |
| ☐ | ☒ | Estimates of effect sizes (e.g. Cohen's *d*, Pearson's *r*), indicating how they were calculated |

*Our web collection on statistics for biologists contains articles on many of the points above.*

## Software and code

Policy information about availability of computer code

Data collection: version of software that was used to get Visium 10x matrices (spaceranger 2.0.1. )

Data analysis: We used Python (v3.9.12) and the packages, anndata (v0.11), scanpy (v1.9.8), squidpy (v1.4.1), sagenet (v1.1.0), Cell2module (GitHub version, fetched 02.2024), pandas (v2.0.3), Torch (v2.1.1), Numpy (v1.24.3), Matplotlib (v3.7.2), Pyro (v1.8.6), SciPy (v1.11.3), and alpha_shape (GitHub clone c171a7d). We used R (v4.3.0) and the packages SingleCellExperiment (v1.24.0), ZellKonverter (v1.12.1), scater (v1.30.1), ComplexHeatmap (v2.16.0), FSA(0.9.6), dplyr(1.1.4), glasso (v1.11),ggplot2 (v3.5.1), igraph (v2.0.1.1), and scran (v1.28.2).

All scripts and custom code for data analysis are available at https://github.com/gerstung-lab/CHOCOLAT-G2P/.

For manuscripts utilizing custom algorithms or software that are central to the research but not yet described in published literature, software must be made available to editors and reviewers. We strongly encourage code deposition in a community repository (e.g. GitHub). See the Nature Portfolio guidelines for submitting code & software for further information.

## Data

Policy information about availability of data

All manuscripts must include a data availability statement. This statement should provide the following information, where applicable:
- Accession codes, unique identifiers, or web links for publicly available datasets
- A description of any restrictions on data availability
- For clinical datasets or third party data, please ensure that the statement adheres to our policy

We used publicly available datasets from scLiverDB (https://guolab.wchscu.cn/liverdb#!/), PanglaoDB (https://panglaodb.se/), MSigDB (https://www.gsea-msigdb.org/gsea/msigdb), GEO (https://www.ncbi.nlm.nih.gov/geo/), MGI (https://www.informatics.jax.org/), and the LiverCellAtlas (https://www.livercellatlas.org/).
We deposited all data to https://zenodo.org/records/10986436/.
In addition, we have launched a web-browser for interactive data analyses (CHOCOLAT-G2P.dkfz.de).

## Research involving human participants, their data, or biological material

Policy information about studies with human participants or human data. See also policy information about sex, gender (identity/presentation), and sexual orientation and race, ethnicity and racism.

| | |
|---|---|
| Reporting on sex and gender | not applicable |
| Reporting on race, ethnicity, or other socially relevant groupings | not applicable |
| Population characteristics | not applicable |
| Recruitment | not applicable |
| Ethics oversight | not applicable |

Note that full information on the approval of the study protocol must also be provided in the manuscript.

# Field-specific reporting

Please select the one below that is the best fit for your research. If you are not sure, read the appropriate sections before making your selection.

☒ Life sciences      ☐ Behavioural & social sciences      ☐ Ecological, evolutionary & environmental sciences

For a reference copy of the document with all sections, see nature.com/documents/nr-reporting-summary-flat.pdf

# Life sciences study design

All studies must disclose on these points even when the disclosure is negative.

| | |
|---|---|
| Sample size | sample size for animal experiments was determined based on prior work: PMID: 29969439, 34509979 |
| Data exclusions | We excluded one animal from the analysis due to failure of hydrodynamic tail vein injection for the initial experiment depicted in ED Fig.2a. |
| Replication | Corresponding hydrodynamic tail vein injection experiments were performed in n=2 animals per group for ST-based readouts.<br>Hydrodynamic tail vein injection experiments shown in Fig.7 were performed in n=4 animals per group.<br>For 10X Visium, we covered regions of interest via serial sections to provide replicas.<br>2 standard 10X Visium experiments were performed, one additional 10X CytAssist experiment was performed in which serial sections were included. All attempts at replication were successful. |
| Randomization | Animals were randomly assigned to the hydrodynamic tail vein injection experiments, ensuring unbiased allocation to experimental conditions. However, regions of interest (ROIs) for 10x Visium were not randomly selected; instead, they were chosen based on histopathological evaluation (H&E staining) to ensure the presence of at least 30 tumor nodules within a defined area (6.5 × 6.5 mm). This approach was necessary to ensure sufficient n of tumor nodules for analysis. Since this selection was based on objective histological criteria rather than experimental conditions, covariate bias was minimized. |
| Blinding | Blinding was not performed in this study because the selection of regions of interest (ROIs) for 10x Visium was based on objective histopathological evaluation (H&E staining). The requirement to observe at least 30 tumor nodules within a predefined area (6.5 × 6.5 mm) necessitated informed selection. |

April 2023

# Reporting for specific materials, systems and methods

We require information from authors about some types of materials, experimental systems and methods used in many studies. Here, indicate whether each material, system or method listed is relevant to your study. If you are not sure if a list item applies to your research, read the appropriate section before selecting a response.

## Materials & experimental systems

| n/a | Involved in the study |
|---|---|
| ☐ | ☒ Antibodies |
| ☒ | ☐ Eukaryotic cell lines |
| ☒ | ☐ Palaeontology and archaeology |
| ☐ | ☒ Animals and other organisms |
| ☒ | ☐ Clinical data |
| ☒ | ☐ Dual use research of concern |
| ☒ | ☐ Plants |

## Methods

| n/a | Involved in the study |
|---|---|
| ☒ | ☐ ChIP-seq |
| ☒ | ☐ Flow cytometry |
| ☒ | ☐ MRI-based neuroimaging |

## Antibodies

| | |
|---|---|
| Antibodies used | CK19, 1:100, Abcam, Catalogue number: Ab133496<br>Hnf4alpha, 1:400, Abcam, Catalogue number: Ab181604<br>GS, 1:1000, BioScience, Catalogue number: BD610517<br>tRFP, 1:500, Evrogen, Catalogue number: AB233<br>GFP, 1:100, CellSignaling, Catalogue number: 2956 |
| Validation | HNF4alpha, 1:400, Abcam, ab181604; Validation was performed by supplier.<br>https://www.abcam.com/en-us/products/primary-antibodies/hnf-4-alpha-antibody-epr16885-chip-grade-ab181604#<br>Specificity and sensitivity confirmed by provider in IHC with multi-tissue microarray (TMA) validation.<br><br>GS, 1:1000, BioScience BD610517; Validation was performed by supplier.<br>https://www.bdbiosciences.com/en-us/products/reagents/microscopy-imaging-reagents/immunofluorescence-reagents/purified-mouse-anti-glutamine-synthetase.610517?tab=product_details<br>Specificity and sensitivity confirmed by provider via Western blot analysis of glutamine synthetase on a rat cerebrum lysate and IHC of Glutamine synthetase staining on a rat cerebrum section.<br><br>tRFP. 1:500 Evrogen, AB233; Validation was performed by supplier.<br>https://evrogen.com/products/antibodies/AB-tRFP.shtml<br>Specificity and sensitivity confirmed by provider in multiple citations via expression of tRFP fusion constructs. https://evrogen.com/products/antibodies/AB-tRFP_Citations.shtml<br><br>GFP, 1:100, Cell Signaling 2956; Validation was performed by supplier.<br>https://www.cellsignal.com/products/primary-antibodies/gfp-d5-1-rabbit-mab/2956?srsltid=AfmBOorjtEQ-RhQgUyR-S9ekMndKszsrTErFlFyfqODMYaauPQUVgcce<br>Specificity and sensitivity confirmed by provider via Western Blot and IHC of PFA-embedded HCC827 cells transfected with GFP-plasmid.<br><br>CK19, 1:100 Abcam, ab133496;Validation was performed by supplier.<br>https://www.abcam.com/en-us/products/primary-antibodies/cytokeratin-19-antibody-epncir127b-ab133496?srsltid=AfmBOoqwF58K57WZlYYG0TaR6uEqXU3jJOMRTKhBS5F3i-0PWvL8dZab<br>Specificity and sensitivity confirmed by provider via IHC analysis of mouse colon tissue sections.<br><br>CK19, 1:100 Abcam, ab133496; Validation using our own data: Fig.7: CK19 staining is specifically observed in histophatologically well-defined cholangiocarcinoma nodules and normal bile duct cells as shown in Fig 7d. |

## Animals and other research organisms

Policy information about studies involving animals; ARRIVE guidelines recommended for reporting animal research, and Sex and Gender in Research

| | |
|---|---|
| Laboratory animals | 8-10 week old female C57Bl/6 animals were purchased from Envigo and used in this study. Housing conditions for the mice included a 12-hour light/12-hour dark cycle, an ambient temperature of 20–24°C, and relative humidity of 45–65%. All animal experiments were conducted in compliance with the regional regulations and in approval of the regional board Karlsruhe, Germany (G-81/20). |

| | |
|---|---|
| Wild animals | No wild animals were used in the study. |
| Reporting on sex | All mice in this study were female |
| Field-collected samples | No field collected samples were used in the study. |
| Ethics oversight | All animal experiments were approved by the regional board Karlsruhe, Germany (G-81/20) |

Note that full information on the approval of the study protocol must also be provided in the manuscript.

## Plants

| | |
|---|---|
| Seed stocks | not applicable |
| Novel plant genotypes | not applicable |
| Authentication | not applicable |

