## [Peer Review File · Nature Biomedical Engineering]

Integrated in vivo combinatorial functional genomics and spatial transcriptomics of tumors to decode genotype to phenotype relationships

Corresponding Author: Dr Marco Breinig

Version 0:

Decision Letter:

Dear Dr Breinig,

Thank you for your revised manuscript, "Integrated combinatorial functional genomics and spatial transcriptomics of tumors decodes genotype to phenotype relationships", which has been seen by the original reviewers one more expert. In their reports, which you will find at the end of this message, you will see that the reviewers #1-3 acknowledge the improvements to the work and have no further questions. Reviewer #4 raises a few additional criticisms that we hope you will be able to address.

As before, when you are ready to resubmit your manuscript, please upload the revised files, a point-by-point rebuttal to the comments from all reviewers, the [reporting summary](https://www.nature.com/authors/policies/ReportingSummary.pdf), and a cover letter that explains the main improvements included in the revision and responds to any points highlighted in this decision.

As a reminder, please follow the following recommendations:

- * Clearly highlight any amendments to the text and figures to help the reviewers and editors find and understand the changes (yet keep in mind that excessive marking can hinder readability).
- * If you and your co-authors disagree with a criticism, provide the arguments to the reviewer (optionally, indicate the relevant points in the cover letter).
- * If a criticism or suggestion is not addressed, please indicate so in the rebuttal to the reviewer comments and explain the reason(s).
- * Consider including responses to any criticisms raised by more than one reviewer at the beginning of the rebuttal, in a section addressed to all reviewers.
- * The rebuttal should include the reviewer comments in point-by-point format (please note that we provide all reviewers will the reports as they appear at the end of this message).
- * Provide the rebuttal to the reviewer comments and the cover letter as separate files.

We expect that you will be able to resubmit the manuscript within 12 weeks of receiving this message. If this is the case, you will be protected against potential scooping. Otherwise, we will be happy to consider a revised manuscript as long as the significance of the work is not compromised by work published elsewhere or accepted for publication at *Nature Biomedical Engineering*.

We look forward to receive a further revised version of the work. Please do not hesitate to contact me should you have any questions.

Best wishes,

Filipe

Reviewer #2 (Report for the authors (Required)):

In this revised version on the m/s, the authors have addressed most of the points I raised during the first review although, from their answers, I get the impression that I was not able to clearly communicate my global impression of the article: It is a novel and cool strategy, with good potential to advance multiplexed in vivo studies. However, I was not convinced of the biological relevance of the findings in cancer research. In their rebuttal, the authors now explain the novelty of their findings, but this was never an issue. In any case, they have addressed some of my concerns and have clarified others and, more importantly, the article is now in a journal more suited for the methodological advance that PERTURB-CAST represents. Overall, I am happy to support its publication in nBME.

Reviewer #3 (Report for the authors (Required)):

The authors have thoughtfully addressed my questions and incorporated additional results and discussion. The manuscript has been substantially enhanced in this revised version.

Reviewer #4 (Report for the authors (Required)):

In this manuscript Breinig, Lomakin, Heidari et al describe a new method to link cancer genotype in mice with gene expression state and the tumor microenvironment. Determining the impact of diverse and complex cancer genotypes in tumors that develop entirely within the natural in vivo context is of great importance. The use of triplets of RTL probes for non-expressed gene as barcodes is clever, the triplets end up being critical. Validation of the BC detection is multifold and suggest that this overall approach is accurate. As Review 4, I have tried to consider both the updated manuscript as well as the previous Reviewers' comments (and the authors' very wordy response).

Thoughts:

1. I actually do not find Figure 1 very clear at all. The text is too small and the figures are too dense. This does the paper (and the reader) a disservice. They should consider splitting Figure 1 into at least 2 figures and incorporating aspects of Supplementary Figure 2 and 3. For example, from Figure 1 it's not even clear what genes that are perturbing. Also, there is quite a lot going on in the vectors in Supplementary Figure 3 (protein tags that seem not to have mostly worked and BCs driven by the pol2 and pol3 promoters). This would be more educational if they made small technical conclusions in the Results about all of these components. Were the Pol2 or Pol3 BCs detected better? It seems that the BC worked better than protein tags, so that seems like a win for their approach, otherwise this really could have been down with peptide tags. In my opinion, not need to do it on adjacent section is not a big advantage of their approach.
2. Perhaps there are other things of importance in the vectors that I missed because I didn't have my magnifying glass. My PhD advisor told me that papers can be reviewed by middle-aged people who might need reading glasses, and you don't want to be the one to remind them of that. Also Figure 3, so small, perhaps split into two if there are two main points in there.
3. While the totally random combination of genetic alterations is presented as a strength in their current study this will not always be the case. Even in their study while there are theoretically 256 combinations, it seems that most initial cells get multiple integrations thus the number of cells that gets only 1 or 2 genetic alterations is likely very low. If one wanted to study many defined three gene combinations they would need a different approach to create those manipulations. Presumably the Authors can imagine ways to do that which could be added to the Discussion.
4. I acknowledge their logic of using shRNAs over Cas9/sgRNAs. However most loss-of-function somatic cancer modeling is with Cas9 so it seems bizarre to not have that as a future direction or mention at all in the Discussion.
5. In retrospect it seems that they should have include other "inert" vectors. Their shRen show up in almost half the tumors and they don't have an inert cDNA, but it also would have shown up in a lot of tumors. So there are lots of vectors that just get carried along in the tumors. Not sure if the Authors would want to add some discussion of how to do this better in the future (perhaps they disagree with me, which is fine)
6. I am not sure that they get across as clearly as they should that many others have generated tumors of defined genotypes or random combination of genotypes (Sidi Chen, Shramek, Roland Rab, Winslow Labs and others) but their approach is way better because of the transcription read out of the cancer cells and the TME. Roland Rad did something extremely similar in pancreas many many years ago and essentially had the same problem in that many tumors had most if not all of the genetic alterations.
7. Pro-codes and other methods that tag cell lines prior to injection are not at all like this approach. Conflating genetically engineered cancer models in which tumors grow from single cells within the correct environment with cell line transplants will ultimately be acknowledge as a major contributor to confusion in cancer biology.
8. They only look at the number of tumors with each perturbation combination. Are there combination that lead to larger tumors?
9. The new leave-one-out data is nice and solidifies their ~new biology.

Minor points.

1. Fig. 2b, I am not sure why the top like is a rainbow. I would seem to be more useful if it indicated singles, doubles, triples, etc
2. Fig 4b, seem backwards from how most people would plot this.
3. Fig S1C. They should show the oncoprint for the genomic alterations that their NICD models (would that be all amplification in Notch genes?)
4. Sup Figures, fonts are pretty small.
5. From the Reviewers comments, while it is unarguable that human tumors acquire mutations sequentially, it is arguable whether that is actually important. There are very few examples to suggest that mutations order is actually important (some modeling of genomic alterations sequentially has been done in GEMM (using Cre/Flp system)) but this hasn't been done in high throughput (as the Author note). The don't agree that the inability to do this sequentially is a major problem for this manuscript.
6. I also don't entirely agree with the review that call this "standard state-of-the art (barcoded genetics..." I am not sure of many other papers of "barcoded genetics" with ST readouts etc. Yes this isn't all new stuff but it innovative and new.

Version 1:

Decision Letter:

Dear Dr Breinig,

Thank you for your revised manuscript, "Integrated combinatorial functional genomics and spatial transcriptomics of tumors decodes genotype to phenotype relationships". Having consulted with Reviewer #4 and Reviewer #5 (who delivered a report as Reviewer #1 in previous round of revisions at Nature Cancer), I am pleased to write that we shall be happy to publish the manuscript in *Nature Biomedical Engineering*. Please note that, as per Reviewer #5's comments, acceptance is conditional on ensuring the research community has access to the data, code, and methods.

We will be performing detailed checks on your manuscript, and in due course will send you a checklist detailing our editorial and formatting requirements. You will need to follow these instructions before you upload the final manuscript files.

Best wishes,
Filipe

Dr Filipe Almeida
Senior Editor, <http://www.nature.com/nbme>>*Nature Biomedical Engineering*

Reviewer #4:

The authors have satisfactorily addressed my concerns and comments.

Reviewer #5:

I find that the Authors have done an outstanding job addressing my previous comments, supported by clarifications, additional analyses, and better placement of their own work in the context of the field. I am strongly supportive of publication of this work and would ask that data, code and methods be made easily accessible to the research community.

Version 2:

Decision Letter:

Dear Dr Breinig,

I am happy to inform you that your manuscript, "Integrated in vivo combinatorial functional genomics and spatial

transcriptomics of tumors to decode genotype to phenotype relationships", has now been accepted for publication in *Nature Biomedical Engineering*.

Over the next few weeks, the figures will be checked for production quality, the text edited to ensure that it conforms to house style, and the manuscript typeset.

Our Articles are published about 40 days after the acceptance date (we recommend that you inform your institutional press office of this timeframe), and you will be notified of the actual publication date a few days in advance. Articles can be published any working day of the week, and are pushed live shortly after 10 am London time.

Publishing agreement. You will be asked to digitally sign a publishing agreement (grant of rights). After the signed publishing agreement has been received, the proofs of the article will be sent to you for review. If you have any queries during the production process, or you cannot meet the requested deadline for returning the proofs, please contact rjsproduction@springernature.com.

Nature Biomedical Engineering is a Transformative Journal. Authors may publish their research with us through the traditional subscription access route, or make their paper immediately open access through payment of an article-processing charge. More [information about publication options](https://www.springernature.com/gp/open-research/transformative-journals) is available.

You may need to take specific actions to [comply](https://www.springernature.com/gp/open-research/funding/policy-compliance-faqs) with funder and institutional open-access mandates. If the work described in the accepted manuscript is supported by a funder that requires immediate open access (as outlined, for example, by [Plan S](https://www.springernature.com/gp/open-research/plan-s-compliance)) and your manuscript was originally submitted on or after January 1st 2021, then you should select the gold OA route. Authors selecting subscription publication will need to accept our standard licensing terms (including our [self-archiving policies](https://www.springernature.com/gp/open-research/policies/journal-policies)), and these will supersede any other terms that the author or any third party may assert apply to any version of the manuscript.

Acceptance of your manuscript is conditional on agreement, by all authors, with both our [media embargo](http://www.nature.com/authors/policies/embargo.html) and [confidentiality and pre-publicity](http://www.nature.com/authors/policies/confidentiality.html) policies. In particular, you may arrange your own publicity of the Article (for instance, through your institutional press office), as long as you ensure that journalists strictly adhere to the media embargo.

To assist you in disseminating the work, as soon as the Article is published you will be able to take advantage of the Springer Nature [SharedIt](https://www.springernature.com/gp/researchers/sharedit) initiative to [generate a unique shareable link to the Article](http://authors.springernature.com/share) that will allow anyone (with or without a subscription) to read it. Recipients of the link who are subscribers will also be able to download and print the PDF.

Thank you for having submitted this work to *Nature Biomedical Engineering*.

Best wishes,

Barbara Cheifet
Editor
Nature Biomedical Engineering

Point-by-Point reply

Reviewers' comments:

Reviewer #1 (Remarks to the Author):

In this manuscript, Breinig, Lomakin and Hedari et al present a novel approach of multiplexing chemosensory receptor transcripts as molecular barcodes. This combinatorial approach enables a detailed mapping of phenotypic and genotypic landscape of tumors. With this approach they were able to characterize the tumor ecosystems comprising of multitudes of co-existing clones. Overall, this pilot study is important and may facilitate the development of new strategies that can leverage the power of existing off the shelf assays to study the tissue ecosystems in greater detail. The computational analysis is nicely done and included appropriate spatial-cell analysis pipelines. The manuscript is well-written and easy to follow.

We thank Reviewer #1 for recognizing the importance of our study and its potential to leverage existing commercialized ST platforms to study tumor ecosystems by means of in vivo functional genomics in greater detail.

Major comment:

1. Validation with Orthogonal Methods: While the manuscript offers substantial insights into the genotypic and phenotypic across the tumor zonation, only a small portion of the reported findings are supported by any orthogonal approaches (for instance, GFP, RFP and GS in Extended Figure 10). However, some of the other findings seem to be very speculative. The authors should therefore employ orthogonal methods such as multiplex immunofluorescence staining to validate the findings to further support their conclusions. I find this to be of particular importance given that if true, this approach could be used broadly. Hence, rigor and confidence in findings from this technical development are necessary. For example, in Figure 2, the authors describe the context dependent genetic interactions, and multiple integration events. At least some of the overexpression/knockdown and multiple integrational events should be corroborated using mIF.

We thank Reviewer #1 for the valuable suggestions related to validation with orthogonal methods.

Regarding the observed context-dependent genetic interactions relevant to tumor biology that are driven by multiple integration events and explicitly mentioned by Reviewer #1 (illustrated in Fig.2 and Fig.3 of our manuscript), we would like to highlight that our approach has since enabled focused follow-up in vivo experiments using alternative measurements to extend our findings beyond liver-tumor zonation. These orthogonal readouts corroborate our finding that VEGFA and mutant Ctnnb1 overexpression within nodules epistatically control cholangiocarcinoma development despite genetic complexity.

Specifically, based on our observations (Fig.2, Fig.3 and Extended Data Fig. 18), we hypothesized the following:

- A) Omission of VEGFA overexpression from our combinatorial search space of eight alterations should accelerate tumor occurrence and reduce the fraction of cholangiocarcinoma-like nodules.
- B) Omission of mutant Ctnnb1 overexpression should delay tumor occurrence and increase the fraction of cholangiocarcinoma-like nodules.

The flexibility of our mouse model allowed us to test these hypotheses using traditional orthogonal methods, such as monitoring time to tumor occurrence in comparative cohorts and performing H&E + Krt19/CK19 IHC to estimate the fraction of cholangiocarcinoma-like nodules. Specifically, we conducted new hydrodynamic tail vein (HDTV) injections in three mouse cohorts, each with four animals (Point-by-point Figure 1).

For animals that received the mixture of all 8 perturbations plasmids all mice developed tumors within a time frame of 8 weeks. Histopathological investigation based on H&E staining revealed the presence of multifocal tumor nodules, a fraction of which were assigned as cholangiocarcinoma and IHC for CK19 confirmed this observation, hence corroborating results from our initial experiments shown in Figure 1, 2, and 3.

For animals that received a mixture of plasmids leaving out VEGFA all mice developed tumors within 5 weeks. Histopathological investigation based on H&E staining revealed the presence of multifocal tumor nodules almost all of which were assigned as hepatocellular carcinoma and IHC for CK19 confirmed this observation due to the absence of CK19 positivity in tumor nodules. Note that CK19 positivity can nonetheless be observed in normal bile duct cells hence providing an internal control for antibody-specificity.

For animals that received a mixture of plasmids leaving out mtCTNNB1 all mice developed tumors within 12 weeks. Histopathological investigation based on H&E staining revealed the presence of tumor nodules almost all of which were assigned as cholangiocarcinoma and IHC for CK19 confirmed this observation.

Taken together, these results (Point-by-point Figure 1) validate predictions of C-G2P-based findings related to genetic interactions between multiple integrated perturbations through in vivo follow-up experiments: mtCTNNB1 epistatically masks and counteracts the occurrence of cholangiocarcinoma phenotypes whereas VEGFA dictates the development of cholangiocarcinoma in the context of genetically heterogenous tumor ecosystems in our mouse model.

Along this line we would again like to point out that standard animal experiments designed to test phenotypic effects of all possible combinations for 8 perturbations in individual cohorts with 4 animals for each combination would mean to sacrifice over 1000 animals, with a substantial fraction of combinations likely not inducing tumors at all (particularly single and pairwise combinations, see doi:10.1053/j.gastro.2020.08.015. and manuscript Fig.2). Our approach allowed us to test all possible combinations in a single experiment, with the direct capability to investigate multiple disease relevant phenotypes other than solely focusing on tumor development.

Overall, these results give us great confidence in the potential of our approach to robustly map higher-order combinatorial perturbations within tumor nodules, infer complex epistatic relationships and to thereby enable new discoveries.

We display these findings as an additional main Figure 4 in a revised version of the manuscript.

Point-by-point Figure 1; New main Fig 4.: **VEGFA and mutant Ctnnb1 confer epistasis control of cholangiocarcinoma development within heterogeneous tumor ecosystems.**

a, Experimental design. In vivo multiplex perturbation HDTV_i injections were performed using parallel in three different animal cohorts using leave-one-out experimental design.

b, Time to tumor occurrence. Animals were palpated twice weekly to monitor tumor development. N = 4 animals per group.

c, Histological quantification of liver tumor subtypes. H&E images were analyzed and tumor nodules were counted and classified into either hepatocellular carcinoma (HCC) or cholangiocarcinoma (CCA). Median +/- SD, N = 4 animals per group, 2 independent liver tissue sections per animal.

d, Abundance of cholangiocarcinoma. CK19 IHC was used as a cholangiocyte marker. Representative samples are depicted. Scale bar = 5 mm.

Besides these additional in vivo experiments that support our key finding related to tumor biology, we fully agree that rigor and confidence in our findings are essential, especially in the context of our technical development that focuses on complex perturbation and phenotype mapping.

To proactively address this concern and to better illustrate our orthogonal validation methods that aimed to demonstrate combinatorial perturbations, we now provide a higher-resolution zoom-in for GFP- and RFP-detection of the same hepatocellular tumor nodules (Point-by-point Figure 2). We hope to thereby better demonstrate that we generated tumors that are indeed positive for both peptide-barcodes and consequently both associated perturbations (GFP and RFP, lower nodule), whereas neighboring tumors exist that solely express a single peptide barcode (RFP, upper nodule). We have included this Figure in the revised manuscript (ED Fig. 10e).

Point-by-point Figure 2: GFP and RFP detection for the same hepatocellular tumor nodules to spatially map peptide barcodes associated with introduced perturbations. Zoom-in: Note that one tumor nodule is positive for RFP alone (upper nodule) whereas the other tumor nodule is positive for both GFP as well as RFP peptide barcodes (lower nodule).

We also provide an additional higher-resolution zoom-in for GFP- and RFP of a cholangiocarcinoma nodule (Point-by-point Figure 3) to demonstrate that we generated cholangiocarcinoma tumor nodules that are likewise positive for both peptide-barcodes, i.e. both perturbations that were introduced.

Point-by-point Figure 3: GFP and RFP detection for the same cholangiocarcinoma tumor nodule to spatially map peptide barcodes associated with introduced perturbations.

Further, to lend support to observations reported in our manuscript with orthogonal readouts (aside from the validation of cholangiocarcinoma nodules by means of CK19 IHC that we already presented in our manuscript; ED Fig.15), we have conducted additional IHC for immune-associated biomarkers, such as Ly6G (as a neutrophil marker) for which we observed spatially confined abundance within some tumor nodules (Point-by-point Figure 4A). However, it is important to note that these experiments were carried out on subsequently prepared tissue sections from the already heavily used FFPE blocks with which we performed Visium, which means they only partially overlap with the samples used for Visium analysis given a pronounced shift in z-plane. Nonetheless, we observed spatially confined abundance of e.g. Ly6G positive cells within tumor nodules, which overall matches ST-based results shown in the manuscript (Fig. 3b).

Point-by-point Figure 4A: Neutrophil mapping by means of Visium and orthogonal IHC for the same region-of-interest. Left: Ly6G IHC to map neutrophils. Arrows indicate intra-tumor abundance of Ly6G positivity. Representative zoom-in. Ly6G positive cells abundant within the tumor mass. scale bar: 200μm. Right. neutrophil signature based on Visium for the indicated region-of-interest (log2, average expression). Note that experiments were carried out on subsequently prepared tissue sections.

To map additional immune-associated biomarkers, we have now also performed IHC for B220/CD45R as a marker of B-cells which potentially aligned with results from our Visium transcriptional profiling strategy to detect B-cell signatures (Point-by-point Figure 4B). Yet again, these experiments were carried out on subsequently prepared tissue sections from the already used FFPE blocks and are hence associated with a shift in z-plane.

Point-by-point Figure 4B: B-cell mapping by means of Visium and orthogonal IHC for the same region-of-interest. Top: B220 IHC to map B-cells. Representative zoom-in: Arrows indicate abundance of B220 positive cells. Bottom: B-cell like signature based on Visium for the indicated region-of-interest (log₂, average expression). Note that experiments were carried out on subsequently prepared tissue sections.

We have further performed additional IHC for alpha smooth muscle actin (α SMA) as a marker of myofibroblasts which aligned with results from our Visium transcriptional profiling strategy to detect fibroblast signatures that we found to be predominantly abundant at tumor-stroma borders (Point-by-point Figure 5). Yet again, these experiments were carried out on subsequently prepared tissue sections from the already used FFPE blocks.

Point-by-point Figure 5: Fibroblast mapping by means of Visium and orthogonal IHC for the same region-of-interest. Left: aSMA IHC as a marker for myofibroblasts. Right: fibroblast signature based on Visium (log₂, average expression).

Further, we would like to emphasize that we have implemented several additional strategies to ensure the robustness of our novel spatial perturbation mapping approach:

1. We utilized multiple barcodes amenable to spatial transcriptomics (ST) readout via RTL-probes, ensuring rigor in identifying individual perturbations at the nodule level.
2. We evaluated barcode detection performance using both standard Visium-FFPE as well as CytAssist Visium workflows.
3. We conducted independent Visium experiments on largely matching regions of interest (ROIs) taken from serial sections to investigate barcode detection and validate reproducibility to assure confidence.
4. We directly assessed perturbations by measuring transcript knockdown or overexpression of respective endogenous transcripts to validate RNAi- and gene overexpression-based perturbations as an orthogonal method independent of barcode mapping.
5. We analyzed publicly available Visium samples from mouse livers to confirm that chemosensory receptor transcripts that we used for barcode redeployment are not detected in murine liver samples and can be used as barcodes to spatially map perturbations.

Finally, we equipped all our perturbation constructs with additional barcodes (Extended Data Fig. 3), including peptide barcoding, following previously established strategies using readouts based on multiplex immunohistochemistry consecutive staining on a single slide (as described for PerturbMap and MICSSS; see <https://doi.org/10.1016/j.cell.2022.02.015>).

Unfortunately, our efforts to establish robust antibody-based multiplexed readouts on FFPE for peptide barcodes such as AU1, AU5, and VSVG etc. were unsuccessful. Consequently, we were not able to map all 8 perturbations introduced using orthogonal readouts.

Given this challenge, we focused on in-house validated immunohistochemistry (IHC) protocols on consecutive sections, utilizing antibody-based detection of GFP (indicative of shTrp53), RFP (indicative of shKmt2c), and GS (indicative of mtCTNNB1) (Figures 1f and Extended Figure 10) as orthogonal readout methods for 3 out of the 8 perturbations used.

NICD overexpression was validated by means of monitoring Notch1 expression and shPTEN by means of Pten expression (Fig.1f). Consequently, we were able to cover 5 of the 8 perturbations, leaving only MYC and VEGFA overexpression, along with the control shREN perturbation, as the remaining alterations for which we cannot provide direct data from orthogonal methods. Along this line, we note that Reviewer #3 found the results presented in Figures 1f (in which we used orthogonal readouts) to be robust and compelling.

In summary, we believe that this combination of validation strategies alongside our results obtained from additional in vivo experiments demonstrate the feasibility of our novel approach to generate combinatorial perturbations and map them at the level of tumor nodules by means of spatial transcriptomics. We show that our approach allows us to decode epistatic relationships between alterations that effect phenotypes that would have been missed by focusing on tumor development as a single readout.

Further, we would like to note that we were able to demonstrate that our approach achieves this goal while simultaneously obtaining tissue-level transcriptional information using the widely-used commercially available Visium-FFPE spatial transcriptomics platform. We are actually tempted to emphasize, that in contrast to our difficulties with establishing robust multiplex peptide barcode readouts that depend on reliable antibodies and staining protocols, Visium-FFPE-based barcode detection worked „out-of-the-box“, given easy-to-scale barcoding for individual perturbations (at least 3 independent molecular barcodes per plasmid) in order to achieve robust spatial tissue-context nodule detection covering all 8 perturbations.

2. Resolution Enhancement and Feasibility Study: The authors performed the spatial experiments using traditional visium with a resolution is 55um with gaps between the capture spots. The Visium analysis falls on a continuum between deconvolution and mapping approaches. Deconvolution methods aim to identify the cell types and their relative proportions contributing to a spot, while mapping methods seek to assign the most likely dominant cell type to a spot. The manuscript would benefit greatly if authors can repeat ST assay with Visium HD (2 um resolution) on a few interesting tissue (preferably with a higher integration #) since the probes are the same – the tissues should still exist.

It would not only inform the feasibility of this approach while using higher resolution spatial assays (in regards to barcode capture, median UMI/genes and barcode counts/cells) but will also help in establishing the minimum capture region resolution required to perform such studies. Additionally, it would enable proper delineation of the effects of each of the single perturbation as well as their combination and would greatly enhance our understanding of the disease.

We fully agree with Reviewer #1 that our approach could benefit from high-resolution spatial transcriptomics (ST), such as VisiumHD and also regard the moderate resolution of Visium-FFPE assay a general limitation of this particular ST platform. On the other hand, 10X Visium is widely used, conveniently standardized, and CytAssist-compatible, i.e. allows ST analysis from pre-sectioned H&E stained tissue samples, making a compelling case for this platform over other higher-resolution ST assays that are ideally performed with fresh frozen samples, e.g. StereoSeq or Open-ST.

Regarding VisiumHD, we likewise believe that RTL-probe-based detection of combinatorial perturbations is most likely compatible with this assay, as both are built on the same transcript capture technology. As such, we are eager to explore VisiumHD, which has only recently

become commercially available. Nonetheless, while VisiumHD offers enhanced resolution, the significant increase in cost compared to the standard Visium platform presents a potential drawback, as this steep price difference can pose limitations to its broader adoption.

More importantly, VisiumHD uses a new, potentially improved, RTL-probe set that differs from the one we used::

(<https://www.10xgenomics.com/support/cytassist-spatial-gene-expression/documentation/steps/probe-sets/visium-mouse-transcriptome-probe-set-v2-0>).

Unfortunately, none of the 38 "redeployed" barcodes we employed in our experiments are compatible with this new V2-RTL-probes. This means we cannot simply use the existing FFPE tissue sections from the experiments presented in the manuscript. We agree with Reviewer #1 that this strategy would provide an ideal testbed to interrogate our experiments at the single cell tissue level with the opportunity for direct comparison to our current data and we regret that this is not feasible. In theory, the V1-RTL-probe sets could be used with VisiumHD slides, but 10X Genomics has advised against this approach (as communicated to us directly). Additionally, the analytical approach for barcode mapping with VisiumHD would likely be different from what we've used.

Consequently, we feel that developing this approach is currently beyond the scope of our study, which already introduces an entirely novel and robust strategy to spatially map combinatorial perturbations within tumor ecosystems using a widely available standardized spatial transcriptomics assay, even though single cell analysis is not within reach using this platform alone.

In this regard, we would like to point out that – to our knowledge- spatially resolved in vivo perturbation mapping is currently only possible using PerturbMap, an appealing approach involving peptide barcoding (ProCode) and cyclic IHC antibody-based detection (Dhainaut et al. *Cell*, 2022, Volume 185, Issue 7, <https://doi.org/10.1016/j.cell.2022.02.015> as well as a similar approach by Rovira-Clavé et al. *Cancer Cell*, Volume 40, Issue 11). However, PerturbMap has only been employed in scenarios where ex vivo manipulated cells were injected into animals. In contrast, our model enables tumors to develop directly from hepatocytes in the livers of living animals. Additionally, the linear combinatorial design of ProCodes challenges their ability to handle the detection of higher-order combinatorial perturbations, which our technology effectively addresses. Further, to enable comprehensive phenotyping, the published implementation of PerturbMap performed spatial transcriptomics on a serial section whereas our technology achieves this from the very same sample.

We are currently also aware of several preprints that describe technologies for high-resolution spatially resolved perturbation mapping via novel molecular barcoding and detection strategies that we have now included in the revised manuscript:

- doi: <https://doi.org/10.1101/2023.12.26.573143>
- doi: <https://doi.org/10.1101/2023.12.26.572587>
- doi: <https://doi.org/10.1101/2023.11.30.569494>
- doi: <https://doi.org/10.1101/2023.01.31.525983>
- doi: <https://doi.org/10.1101/2024.01.18.576210>

We also note that two of these preprints have now been published during the review process for our manuscript:

- Kudo, T., *et al.* Multiplexed, image-based pooled screens in primary cells and tissues with PerturbView. *Nat Biotechnol* (2024). <https://doi.org/10.1038/s41587-024-02391-0>
- Gu, J., *et al.* Mapping multimodal phenotypes to perturbations in cells and tissue with CRISPRmap. *Nat Biotechnol* (2024). <https://doi.org/10.1038/s41587-024-02386-x>

However, we would like to highlight the following points:

1. None of these approaches involve direct spatial transcriptomics with whole-transcriptome coverage readouts on the same sample.
2. None of them utilize a commercially available assay platform to map perturbations; instead, they rely on custom probes and readout protocols including complex image analysis.
3. None of them have been tested in a setting where tumors develop directly in the organ of an animal, as they use in vitro cell culture experiments or xenograft models where ex vivo manipulated cells are injected into animals.
4. None of them address higher-order combinatorial perturbation experiments. For instance, the manuscript describing the CRISPRmap approach (<https://doi.org/10.1101/2023.12.26.572587>) suggests that combinatorial barcoding could be a compelling future strategy to investigate genetic interactions: “Another opportunity for RNA-based barcoding is to start exploring the effects of combinatorial gene perturbations,...”. In their published manuscript (Gu, J., *et al.*, 2024) they state: “Future studies could leverage this feature to study genetic interactions through combinatorial perturbations...”

Our manuscript reports a technological innovation that complements the aforementioned optical pooled screening approaches, offering the advantage of spatially mapping perturbations and comprehensive transcriptional signatures (building on approximately 19K transcripts that can theoretically be detected by Visium for FFPE) for phenotyping within the same sample using a commercially available ST readout and a mouse model in which tumors develop from normal cells directly in the organ of an animal.

In our revised manuscript, we now clearly emphasize the distinctive features of our approach:

“Previous in vivo approaches to spatially map perturbations relevant to cancer employed ex vivo manipulated cells that were subsequently injected into animals^{14,15,17}.”

And

„Current approaches to spatially map perturbations within tissue engaged custom protocols and orthogonal readouts to also obtain transcriptomics-based phenotypic profiles, such as sequential antibody-based barcode detection and 10X Visium spatial transcriptomics (ST) on an additional tissue sample¹⁴.“

Can the authors also include probes against the barcode (RFP/GFP etc) in their HD run to test for the sensitivity of the technology and the approach?

Related to Visium and the spike-in of customized RTL-probes (e.g., for detecting GFP, RFP, etc.), to our knowledge, this has not been tested with VisiumHD.

Further, as far as we are aware, the use of spike-in customized RTL-probes for standard Visium-FFPE on actual tissue sections has not been extensively reported. Currently, we are only aware of 10X's technical note: CG000621 | Rev C in which they report GFP and RFP detection using mixtures of cells grown and manipulated on plastic dishes and one report for co-detection of viral transcripts (<https://doi.org/10.1186/s13059-023-03080-y>) for which 10X Genomics provided probe design.

In this regard, we note that we found 5 out of 38 redeployed barcodes that should have been detected by RTL-probes provided with the 10X V1-RTL-probe set to show insufficient signal across all samples investigated (Extended Data Fig. 6) indicating the importance of efficient probe design that we have discussed in our manuscript.

We would very much like to learn about additional use-cases for customized RTL-probes on “real tissue samples” in order to enable robust custom RTL-probe based experiments for both Visium as well as VisiumHD.

Again, we would like to express that we fully agree with Reviewer #1 that our approach will in the future benefit from high-resolution spatial transcriptomics, as well as the use of reliable customized RTL-probes.

Minor comment:

1. Can the authors report the QCs for the spatial assay. For instance, median UMIs, genes and barcode counts/spot.

We report QC information and summary stats for all samples and included this information in ED Fig. 4g.

Sample	Total Spots	Total Genes	Median UMIs per Spot	Median Genes per Spot	Spot Diameter (fullres)	Chemistry Description	Software Version
ML_II_C	3608	19464	32744.0	6856.5	128.9	Visium V1 Slide	spaceranger-2.0.1
ML_II_B_3Cyt	4693	19464	29412.0	6139.0	138.8	Visium V4 Slide	spaceranger-2.0.1
ML_II_B	3956	19464	10649.0	4077.0	128.9	Visium V1 Slide	spaceranger-2.0.1
ML_II_A_3Cyt	4016	19464	22539.0	5732.0	138.7	Visium V4 Slide	spaceranger-2.0.1
ML_III_A	4134	19464	27470.5	6533.5	128.9	Visium V1 Slide	spaceranger-2.0.1
ML_III_A_2Cyt	4284	19464	19479.5	5303.5	138.7	Visium V4 Slide	spaceranger-2.0.1
ML_I	4217	19464	14202.0	4874.0	129.0	Visium V1 Slide	spaceranger-2.0.1
ML_II_A_2	4141	19464	12190.0	4389.0	129.0	Visium V1 Slide	spaceranger-2.0.1
ML_II_B_2Cyt	1817	19464	24984.0	5876.0	138.9	Visium V4 Slide	spaceranger-2.0.1
ML_II_A_1	3870	19464	16214.0	5101.0	129.0	Visium V1 Slide	spaceranger-2.0.1
ML_III_B	2697	19464	30589.0	6729.0	128.9	Visium V1 Slide	spaceranger-2.0.1
ML_I_2	3916	19464	23992.0	6163.0	128.8	Visium V1 Slide	spaceranger-2.0.1

As an example, we further provide summary stats directly from a 10X Space Ranger output (sample ML_III_A_2Cyt) to confirm the quality of our assays for one representative sample for which Visium CytAssist was used (Point-by-point Figure 6).

Point-by-point Figure 6: 10X Space Ranger summary stats for a representative Visium CytAssist sample.

Reviewer #2 (Remarks to the Author):

NATCANCER-TR14236 "Integrated combinatorial functional genomics and spatial transcriptomics of tumors decodes genotype to phenotype relationships", by Breinig et al. In this m/s, the authors present CHOCOLAT-G2P, a strategy to integrate multiplexed in vivo functional genomics and special transcriptomics experiments to unveil genotype-phenotype relationships in liver tumors. More specifically, the authors simultaneously introduce 8 genetic perturbations in a mouse model (via HDTV injection) and study the produced tumors using in vivo functional genomics (barcoded) and 10x Visum special transcriptomics. Overall, the authors identify and study 324 tumors, finding distinct tumor subtypes (e.g. cholangiocarcinoma and zonation-associated hepatocellular carcinomas) as well as several positive and negative epistatic interactions (e.g. cholaginocarcinoma is positively associated with VEGFA and negatively with mtCtnnb1). Overall, I think this is a pretty cool strategy that has potential to advance multiplexed in vivo studies.

We thank Reviewer #2 for recognizing the potential of our strategy to significantly advance multiplexed in vivo studies.

However, in its present first implementation, the obtained results are not particularly novel. As I see it, the main novelty of the strategy is the HDTV injection of several genetic perturbations so that a heterogeneous population of tumors develop. Then, the characterization of these tumors is pretty much standard state-of-the-art (barcoded genetics and spatial transcriptomics).

We appreciate Reviewer #2's opinion and accordingly feel the need to clarify the particular novelty of our approach and highlight its potential advantages in more detail.

One primary innovation of our strategy lies in integrating molecular barcode detection with a commercially available spatial transcriptomics (ST) platform to map complex perturbations alongside complex phenotypes at the tissue level. To our knowledge, there is currently no other standard state-of-the-art strategy that achieves this, rendering our approach an entirely new spatial perturbation mapping technology.

To emphasize this innovation more clearly, we now specifically refer to this technology as PERTURB-CAST (Perturbation Barcode Capture for Spatial Transcriptomics) throughout our revised manuscript and have adjusted the abstract accordingly.

As far as we know, spatially resolved in vivo perturbation mapping is currently only possible using PerturbMap, an appealing approach involving peptide barcoding (ProCode) and cyclic IHC antibody-based detection (Dhainaut et al. Cell, 2022, Volume 185, Issue 7, <https://doi.org/10.1016/j.cell.2022.02.015> as well as a similar approach by Rovira-Clavé et al. Cancer Cell, Volume 40, Issue 11). However, PerturbMap has only been employed in scenarios where ex vivo manipulated cells were injected into animals. In contrast, our model enables tumors to develop directly from hepatocytes in the livers of living animals. Additionally, the linear combinatorial design of ProCodes challenges their ability to handle the detection of higher-order combinatorial perturbations, which our technology effectively addresses. Further, to enable comprehensive phenotyping, the published implementation of PerturbMap performed spatial transcriptomics on a serial section whereas our technology achieves this from the very same sample.

We are currently also aware of several preprints that describe technologies for spatially resolved perturbation mapping via novel molecular barcoding and detection strategies :

- doi: <https://doi.org/10.1101/2023.12.26.573143>
- doi: <https://doi.org/10.1101/2023.12.26.572587>
- doi: <https://doi.org/10.1101/2023.11.30.569494>
- doi: <https://doi.org/10.1101/2023.01.31.525983>
- doi: <https://doi.org/10.1101/2024.01.18.576210>

We also note that two of these preprints have now been published during the review process for our manuscript:

- Kudo, T., *et al.* Multiplexed, image-based pooled screens in primary cells and tissues with PerturbView. *Nat Biotechnol* (2024). <https://doi.org/10.1038/s41587-024-02391-0>
- Gu, J., *et al.* Mapping multimodal phenotypes to perturbations in cells and tissue with CRISPRmap. *Nat Biotechnol* (2024). <https://doi.org/10.1038/s41587-024-02386-x>

However, we would like to highlight the following points:

1. None of these approaches involve direct spatial transcriptomics with whole-transcriptome coverage readouts on the same sample.
2. None of them utilize a commercially available assay platform to map perturbations; instead, they rely on custom probes and readout protocols.
3. None of them have been tested in a setting where tumors develop directly in the organ of an animal, as they use in vitro cell culture experiments or xenograft models where ex vivo manipulated cells are injected into animals.
4. None of them address higher-order combinatorial perturbation experiments. For instance, the manuscript describing the CRISPRmap approach (<https://doi.org/10.1101/2023.12.26.572587>) suggests that combinatorial barcoding could be a compelling future strategy to investigate genetic interactions: “Another opportunity for RNA-based barcoding is to start exploring the effects of combinatorial gene perturbations,...”. In their published manuscript (Gu, J., *et al.*, *Nat Biotechnol* (2024) they state: “Future studies could leverage this feature to study genetic interactions through combinatorial perturbations...”

Our manuscript primarily reports a technological innovation that complements the aforementioned approaches, offering the advantage of spatially mapping perturbations and comprehensive transcriptional signatures (building on approximately 19K transcripts that can theoretically be detected) for phenotyping within the same sample using a mouse model in which genetically diverse tumors develop from normal cells directly in the organ of an animal.

Again, we want to emphasize that our submission reports about a new experimental framework that introduces a new technology, illustrating for the first time how in vivo combinatorial perturbations can be mapped within tissue using the commercially available Visium for FFPE platform.

In our revised manuscript, we now clearly emphasize the distinctive features of our approach:

“Previous in vivo approaches to spatially map perturbations relevant to cancer employed ex vivo manipulated cells that were subsequently injected into animals^{14,15,17}.”

and

„Current approaches to spatially map perturbations within tissue engaged custom protocols and orthogonal readouts to also obtain transcriptomics-based phenotypic profiles, such as sequential antibody-based barcode detection and 10X Visium spatial transcriptomics (ST) on an additional tissue sample¹⁴.“

Regarding the novelty of our biological findings, we acknowledge that the potential biological relevance should be interpreted with caution due to the limited number of samples. However, the substantial number of tumor nodules we analyzed (324 nodules) provided a robust foundation for following conclusions:

1. To our knowledge, our approach provides the first in vivo mouse model-based demonstration that Gp2 upregulation is associated with cholangiocarcinoma.

This novel observation in a mouse model corroborates existing human data linking Gp2 expression to cholangiocarcinoma.

Indeed, it has been suggested that GP2 “can be hypothesized as a novel marker in large bile duct diseases. In particular, in PSC (primary sclerosing cholangitis), anti-GP2 IgA identified a subgroup of patients with severe phenotype and poor survival due to cholangiocarcinoma.” (Jendrek ST, Gotthardt D, Nitzsche T, et al. *Gut* 2017;66:137–144).

In our manuscript, we stated: “Interestingly, CHOCOLAT-G2P pinpointed, among others, expression of ... Gp2 (Pancreatic Glycoprotein 2) as being associated with cholangiocarcinoma (Fig. 3b; Extended Data Fig. 14a). The latter observation harmonizes with earlier research, suggesting that anti-GP2 IgA autoantibodies enable early cholangiocarcinoma detection in subsets of human patients.”

2. Our approach is the first to demonstrate, in an in vivo mouse model, that zonation-associated transcriptional signatures define hepatocellular carcinoma phenotypes in the context of genetic heterogeneity.

This finding aligns very well with human data (Ng, C. K. Y., Piscuoglio, S. & Terracciano, L. M. Molecular classification of hepatocellular carcinoma: The view from metabolic zonation. *Hepatology*. Baltim. Md 66, 1377–1380 (2017)).

In our manuscript, we stated: “Spatial division of metabolic functions is not only central to liver tissue organization under physiological conditions but has been proposed to enable molecular classification of human hepatocellular carcinoma.”

To our knowledge, the relevance of zonation-associated phenotype signatures of liver tumors is only poorly understood, likely due to the lack of spatial tissue-level resolved comprehensive phenotyping approaches to interrogate liver cancer mouse models. Prior studies have relied on bulk RNA-seq, occasionally single-cell RNA-seq, or standard IHC using a limited set of markers.

In fact, a very recent preprint revealed that liver cancer initiation and tumor progression are dependent on zonation. Using elegant mouse models and fate mapping, the authors demonstrate the relevance of zone 3 (central phenotype in our study, which we observed to be the most abundant cancer subtype; Fig.3) and further suggest strategies for cancer prevention that capitalize on knowledge of zonation dependent mechanisms of liver carcinogenesis (doi: <https://doi.org/10.1101/2024.01.10.575013>).

We have now included this information in our revised manuscript.

3. Our approach is the first to demonstrate epistatic regulation of cholangiocarcinoma development by VEGFA and CTNNB1.

Our results, along with additional follow-up studies in larger animal cohorts (see comments for Reviewer#1, Point-By-Point Fig 5, new main Fig.4 and results discussed below), demonstrate that VEGFA and CTNNB1 influences phenotypic decisions and tumorigenesis in heterogeneous liver tumor ecosystems.

However, as it happens in e.g. deep mutational scanning it is very important to control the input so that a really heterogeneous population of tumors is generated. In this case, with this initial implementation, what the authors observe is that the vast majority of tumors are developed from the integration of >5 mutations, with 82% containing Myc and 80% mtCnnb1. On the one hand, I am not sure on how realistic is this scenario (i.e. tumor populations with >5 driver mutations) and, on the other, it severely limits the study of a more heterogeneous population and the interactions of each of these tumor types with the environment.

Related to the complexity of genetic alterations that can be observed in human liver tumors, we found that over 40% have at least 5 co-occurring drivers and over 30% contained at least 8 drivers when focusing only on alterations in known cancer-driving genes defined by COSMIC (Point-by-point Figure 7). Consequently, tumor populations with >5 driver mutations are a very realistic scenario related to liver cancer and most current mouse models to investigate cancer development do not capture this complexity. We have included this information in ED Fig.1 in our revised manuscript.

Point-by-point Figure 7: Median number of cancer driving genes per individual tumor based on a human liver cancer dataset.

a) Frequency of potential driver mutations per sample in the TCGA-LHCC dataset. Potential drivers were defined as either amplification or fusion of known COSMIC oncogenes, or homozygous deletion, nonsense mutation, splice site mutation, or frameshift deletion/insertion in tumour suppressor genes. The median value of 4 co-occurring drivers per sample is indicated by the dashed line.

b). Complementary cumulative distribution plot showing the frequency of samples with at least k driver mutations. The distribution highlights a long tail, with over 40% of samples containing at least 5 co-occurring drivers and over 30% containing at least 8 drivers.

We strongly believe that genetic complexity and heterogeneity is an inherent problem to our understanding of tumor phenotypes and cancer treatment and the lack of model systems to systematically investigate associated phenomena has spurred our interest in developing the approach validated in the current manuscript.

Additionally, we would like to highlight that pooled functional genomic screens using clonal selection and tumor development as a readout have been conducted using the same mouse model on which our approach is based, i.e. injection of plasmids pools and transposon mediated integration of perturbations. Our strategy of generating higher-order combinations that are mapped within spatial tissue context can be seen as an extension of these previously described methods as well as a previously published CRISPR multiplexing approach, which rely on the enrichment of identifiable perturbations driven by competitive tumor growth (see <https://doi.org/10.1038/s41568-020-0275-9> for a review of various prior in vivo functional genomics approaches and <https://doi.org/10.1101/2024.03.07.583774> for a preprint describing the potential of a novel multiplex method to generate combinatorial alterations).

Regarding the high occurrence of Myc and mtCtnnb1 (approximately 80%), we argue that this likely reflects a clonal selective advantage, which is a common readout in forward genetic screens. Indeed, the pairwise combination of Myc and mtCtnnb1 has previously been identified to be one of the strongest cooperating alterations in the mouse model we use when a total of over 20 pairwise combinations were tested in individual cohorts (as measured by time to tumor occurrence; doi:10.1053/j.gastro.2020.08.015), hence aligning with our observation. Despite this strong clonal selection, we were still able to identify—and more importantly, spatially map and phenotypically characterize—a significant portion of the possible combinatorial genotype spectrum (256 combinations), albeit at lower frequencies. We speculate that this scenario mirrors human tumor heterogeneity, where dominant clones are prevalent, and a subset of genetically distinct clones persist in the background, potentially poised for expansion under the right selective pressures.

Additionally, having done the experiment only once, I am really not sure whether the perturbation enrichments/depletions and the epistatic interactions are really driven by the biology or it is just by chance that these tumors developed first and, somehow, hindered the progression of other tumor types. I understand that would be very costly to repeat the experiment a significant number of times, but may be just the injection and the quantification of the genetic perturbations in each tumor could be enough to see if the findings presented are really biology driven.

We understand and appreciate the Reviewer's concern. Overall, we envision our current strategy as primarily a hypothesis-generating screening platform that can form the basis for more detailed follow-up studies. To illustrate: testing all possible combinations for 8 perturbations in individual cohorts with 4 animals for each combination would mean to sacrifice over 1000 animals, with a substantial fraction of combinations likely not inducing tumors at all (particularly single and pairwise combinations, see doi:10.1053/j.gastro.2020.08.015. and manuscript Fig.2). We have now briefly highlighted this combinatorial explosion problem in our revised manuscript.

In this context, our approach that allowed us to test all possible combinations in a single experiment, and investigated a large number of tumor nodules (324 nodules) observed in these samples enabled us to make several predictions that align well with findings from prior liver cancer mouse models using hydrodynamic tail vein injection to establish pairwise

alterations in individual cohorts with larger animal numbers (e.g., <https://doi.org/10.1053/j.gastro.2020.08.015> and doi: 10.1126/sciadv.abn5683).

More importantly, our results indicated an until now unrecognized strong epistatic relationships between VEGFA and Ctnnb1 in relation to cholangiocarcinoma development. To validate that this observation is genuinely driven by biological mechanisms and not simply by chance, we have now conducted follow-up analyses using more animals, the results of which support the predictions made by our CHOCOLAT-G2P experiments.

Specifically, based on our observations (Fig.2, Fig.3 and Extended Data Fig. 18), we hypothesized the following:

- A) Omission of VEGFA overexpression from our combinatorial search space of eight alterations should accelerate tumor occurrence and reduce the fraction of cholangiocarcinoma-like nodules.
- B) Omission of mutant Ctnnb1 overexpression should delay tumor occurrence and increase the fraction of cholangiocarcinoma-like nodules.

The flexibility of our mouse model allowed us to test these hypotheses using traditional methods, such as monitoring time to tumor occurrence in comparative cohorts and performing H&E + Krt19/CK19 IHC to estimate the fraction of cholangiocarcinoma-like nodules. Specifically, we conducted hydrodynamic tail vein (HDTV) injections in three mouse cohorts, each with four animals (Point-by-point Figure 1).

For animals that received the mixture of all 8 perturbation plasmids, we observed that all mice developed tumors within a time frame of 8 weeks. Histopathological investigation based on H&E staining revealed the presence of multifocal tumor nodules, a fraction of which were assigned as cholangiocarcinoma and IHC for CK19 confirmed this observation, hence largely corroborating results from our initial experiments.

For animals that received a mixture of plasmids leaving out VEGFA all mice developed tumors within 5 weeks. Histopathological investigation based on H&E staining revealed the presence of multifocal tumor nodules almost all of which were assigned as hepatocellular carcinoma and IHC for CK19 confirmed this observation due to the absence of CK19 positivity in tumor nodules. Note that CK19 positivity can nonetheless be observed in normal bile duct cells hence providing an internal control for antibody-specificity.

For animals that received a mixture plasmids leaving out mtCTNNB1 all mice developed tumors within 12 weeks. Histopathological investigation based on H&E staining revealed the presence of tumor nodules almost all of which were assigned as cholangiocarcinoma and IHC for CK19 confirmed this observation.

Taken together, these results (Point-by-point Figure 1) validate predictions of C-G2P-based findings related to epistasis through in vivo follow-up experiments: mtCTNNB1 epistatically masks and counteracts the occurrence of cholangiocarcinoma phenotypes whereas VEGFA dictates the development of cholangiocarcinoma in the context of genetically heterogenous tumor ecosystems in our mouse model. These results give us great confidence that the observed epistatic interaction is really driven by the biology.

We display these findings as an additional main Figure 4 in a revised version of the manuscript.

Point-by-point Figure 1; New main Fig 4.: **VEGFA and mutant Ctnnb1 confer epistasis control of cholangiocarcinoma development within heterogeneous tumor ecosystems.**

a, Experimental design. In vivo multiplex perturbation HDTV_i injections were performed using parallel in three different animal cohorts using leave-one-out experimental design.

b, Time to tumor occurrence. Animals were palpated twice weekly to monitor tumor development. N = 4 animals per group.

c, Histological quantification of liver tumor subtypes. H&E images were analyzed and tumor nodules were counted and classified into either hepatocellular carcinoma (HCC) or cholangiocarcinoma (CCA). Median +/- SD, N = 4 animals per group, 2 independent liver tissue sections per animal.

d, Abundance of cholangiocarcinoma. CK19 IHC was used as a cholangiocyte marker. Representative samples are depicted. Scale bar = 5 mm.

Aside from this encouraging validation experiment related to liver tumor biology, our primary goal was to demonstrate the feasibility of our technology, given that our approach to spatially

map perturbations alongside phenotypically informative transcription profiles on the same samples is entirely novel.

As detailed in our response to Reviewer #1, we believe that the multiple strategies we employed to ensure robustness and confidence convincingly establish the feasibility of our novel PERTURB-CAST approach to map combinatorial perturbations within tumor nodules while obtaining tissue-level transcriptional information using the Visium commercial spatial transcriptomics platform.

Very minor point

- Fig. 17a,b in line 500 should be Extended Data Fig. 17a,b

We thank the Reviewer for pointing this out and correct this mistake.

Reviewer #3 (Remarks to the Author):

Strengths

Technically, the CHOCOLAT-G2P system represents a novel in vivo approach for perturbing key oncogenes and tumor suppressor genes in animal models. This system leverages a combination of several cutting-edge technologies, including in vivo gene modulation, spatial transcriptomics, barcoding, and a robust bioinformatics pipeline. The integration of these advanced techniques allows CHOCOLAT-G2P to serve as a powerful tool not only for studying tumor biology but also for broader applications in various fields of biomedical research. The presentation and illustration of the manuscript are clear, and the manuscript itself is well-written. The results presented are robust and compelling, particularly as demonstrated in Figures 1f and 2a. As a technological and methodological paper, it introduces a groundbreaking in vivo method for gene perturbation, which holds significant promise for advancing our understanding of genetic interactions and their effects on biological systems. The CHOCOLAT-G2P system is particularly valuable because it enables the exploration of complex genetic landscapes in a controlled and systematic manner. By allowing simultaneous perturbation of multiple genes within a living organism, this system offers insights into the dynamic interplay between oncogenes and tumor suppressors in the context of the tumor microenvironment. Furthermore, the use of spatial transcriptomics provides a spatially resolved view of gene expression, enhancing the ability to understand the tissue-specific effects of genetic alterations.

We are grateful to Reviewer #3 for acknowledging that our work "...introduces a groundbreaking in vivo method for gene perturbation, offering significant potential to deepen our understanding of genetic interactions and their impact on biological systems."

We also appreciate the recognition that our approach could serve as a "...powerful tool, not only for studying tumor biology but also for broader applications across various fields of biomedical research..."

Concerns

During tumorigenesis, the activation of oncogenes and the inactivation of tumor suppressor genes typically occur in a sequential manner. This sequential process is thought to be crucial for the gradual accumulation of genetic alterations that drive the transformation of normal cells into malignant ones. For example, in colorectal cancer, the inactivation of APC is followed by mutations in KRAS and then by the loss of TP53. These genetic alterations are rarely observed to occur simultaneously in the early lesion, reflecting the stepwise nature of tumorigenesis. However, the methodology presented in this paper introduces a model that allows for the simultaneous occurrence of up to 2^8 combinations of genetic events. While this approach offers a comprehensive exploration of possible genetic interactions, it also raises questions about the biological relevance of such a model. There is concern regarding whether this genotype-to-phenotype modeling accurately reflects the natural progression of human tumorigenesis, where genetic alterations typically occur in a more sequential manner rather than simultaneously. In the context of liver cancer, the findings from the TCGA study provide insights into the patterns of genetic alterations associated with hepatocellular carcinoma. Mutations in TP53 and CTNNB1 are largely mutually exclusive. However, the data presented in this manuscript showed a significant percentage of clones that exhibit co-occurrence of TP53 and CTNNB1 mutations (Figure 2a). This raises further questions about the validity of the genotype-to-phenotype modeling employed in the study.

We fully share Reviewer #3's concern and aimed to briefly address this in our previously submitted manuscript, suggesting that our approach could be adapted to include the sequential introduction of perturbations, such as through inducible expression systems. As Reviewer #3 suggested, we now provide a more detailed discussion of this limitation (see below).

Regarding the concern that our genotype-to-phenotype modeling does not accurately reflect the natural progression of human tumorigenesis, we agree that our strategy of simultaneously establishing combinatorial alterations is a bold approach to tackle the combinatorial explosion problem underlying tumor heterogeneity, which challenges the systematic investigation of all possible genetic alterations observed in human (liver) cancer across different animal cohorts. To illustrate: testing all possible combinations for 8 perturbations in individual cohorts with 4 animals for each combination would mean to sacrifice over 1000 animals, with a substantial fraction of combinations (particularly single and pairwise combinations, see doi:10.1053/j.gastro.2020.08.015. and manuscript Fig.2) likely not inducing tumors at all. Our higher-order combinatorial approach currently serves to complement reductionist approaches using genetically engineered mouse models that generally have only a few defined genetic changes (most often pairwise for liver cancer models, e.g., <https://doi.org/10.1053/j.gastro.2020.08.015>) whereas over 40% of human liver cancer samples present at least 5 co-occurring drivers and over 30% containing at least 8 drivers (Point-by-point Figure 7). We have now included this information in the revised manuscript.

Point-by-point Figure 7: Median number of cancer driving genes per individual tumor based on a human liver cancer dataset.

a) Frequency of potential driver mutations per sample in the TCGA-LHCC dataset. Potential drivers were defined as either amplification or fusion of known COSMIC oncogenes, or homozygous deletion, nonsense mutation, splice site mutation, or frameshift deletion/insertion in tumour suppressor genes. The median value of 4 co-occurring drivers per sample is indicated by the dashed line.

b). Complementary cumulative distribution plot showing the frequency of samples with at least k driver mutations. The distribution highlights a long tail, with over 40% of samples containing at least 5 co-occurring drivers and over 30% containing at least 8 drivers.

Testing combinations sequentially would add an additional layer of combinatorial complexity, i.e. comparing sequence $A \rightarrow B \rightarrow C$ versus $A \rightarrow C \rightarrow B$, etc. Indeed, we are very much interested in these approaches, which have to our knowledge primarily been carried out in cell culture-based settings (e.g. see DOI: 10.1126/science.abi8175, a study we have referenced multiple times in our current manuscript).

Generating higher-order sequential alterations in an autochthonous mouse model in which tumors actually develop from normal cells within their native tissue environment is however more challenging. We are not aware of any reports demonstrating the current feasibility to

robustly carry out such higher-order combinatorial experiments in a systematic fashion with well-defined genetic alterations that are introduced in a stepwise manner.

Related to the validity of the genotype-to-phenotype modeling employed by means of our liver cancer mouse model, we would like to note that pooled functional genomic loss-of-function screens in which alterations were simultaneously introduced (most often pairwise combinations using multiplex testing of loss-of-function alterations in a tumor-prone single perturbation background) and CRISPR multiplexing based on our mouse model system have been previously conducted and resulting phenotypes from combinatorial perturbations were described to match with observations from human liver cancer (e.g. doi:10.1053/j.gastro.2020.08.015).

Nonetheless, we are aware of the limitations of this particular liver cancer mouse model to recapitulate human tumorigenesis. As an example, activating MAPK-signaling by introducing mutant Ras is a frequently used approach to model hepatocellular carcinoma tumor development within this experimental setting (e.g. <https://doi.org/10.1038/s41467-024-46835-2>) even though oncogenic Ras mutations are very uncommon (<3%; Point-by-point Figure 8) in this human liver cancer subtype (<https://doi.org/10.1016/j.cell.2017.05.046>).

Point-by-point Figure 8. Low frequency of oncogenic Ras mutations in human liver cancer. Oncoprint representation; data based on https://www.cbioportal.org/study/summary?id=lihc_tcga.

Along this line and as noted by Reviewer#3, even though human data suggest that TP53 loss of function and oncogenic CTNNB1 mutations might be mutually exclusive, we indeed observed clones that exhibit a co-occurrence for both alterations in our experiments. However, we note that the implementation to infer mutual exclusivity provided by cBioportal does not assign significance for associations between TP53 and CTNNB1 mutations, even though these alterations are amongst the most predominant ones found in hepatocellular carcinoma. In other words, human liver tumors that harbor both, mutations in TP53 alongside mutations in CTNNB1, exist (Point-by-point Figure 9) and the underlying determinants that allow for (or select against) this co-occurrence remain elusive.

Point-by-point Figure 9. Co-occurrence of Trp53 and CTNNB1 alterations in human liver cancer. Oncoprint representation and inference of mutual exclusivity for CTNNB1-TP53; 26 tumor samples present co-occurring alterations in CTNNB1 and TP53. Data based on https://www.cbioportal.org/study/summary?id=lihc_tcga.

Overall, we hope that we and others can in the future further improve our liver cancer mouse model that builds upon the well-established hydrodynamic tail vein injection method to better reflect the natural progression of human liver tumorigenesis where genetic alterations tend to occur in sequential manner, while preserving the opportunity to pre-select alterations of interest and monitor complex genotypes of tumor nodules within their tissue context. The flexibility of our mouse model should in the future allow us to better tailor combinatorial perturbations with the aim to ideally mimic complex alteration spectra observed in human cancer.

The tumor microenvironment in this model system is highly complex, as each tumor clone represents a distinct combination of oncogenic and tumor suppressor alterations. While human tumors also exhibit intertumoral heterogeneity, most tumor clones are derived from and co-evolved with one or very few ancestral clones. Therefore, it remains unclear whether this system accurately represents the tumor microenvironment as it occurs in nature. For example, Myc-driven clones may produce numerous immunosuppressive cytokines, which could systematically or locally affect TP53-mutant clones. However, this system is unable to distinguish the driving events within such a complicated microenvironment.

We again share Reviewer #3's concern and appreciate the reviewer highlighting the potential crosstalk between genetically different tumor nodules.

The ability to observe such interactions without prior knowledge of specific markers initially motivated us to develop a strategy that allows us to generate genetically different co-existing tumor nodules that can be interrogated by the integration of spatial mapping of perturbations with transcriptional profiling from the same tissue slide.

While the current experimental setting (which was chosen as an initial challenging testbed for our technology) may indeed be too complex, we believe our approach offers a valuable opportunity to investigate how different tumor nodules potentially interact and influence one another (see extended discussion below).

Given the flexibility of our approach, we anticipate that reducing combinatorial complexity in future experiments could help investigate these interactions and leverage functional genomics to explore underlying mechanisms given that identifying genotype-specific interactions of tumor subclones in human samples is indeed challenging.

Suggestions

Both of the above limitations should be carefully discussed.

We appreciate the reviewer's feedback and suggestion to better discuss these limitations in our manuscript. In a revised version of our manuscript, we now explicitly highlight these limitations in the discussion.

Revised Discussion:

„Limitations:

First, C-C2P currently generates tumor heterogeneity by establishing combinatorial alterations simultaneously. Although this approach could aid to initially explore a vastly unknown epistatic

interaction space, this contrasts tumor development in humans, where in most cases, cells sequentially acquire alterations¹. To address this, experimental strategies that enable stepwise introduction of genetic alterations require further exploration. Inducible perturbation systems, previously used in the liver cancer mouse model employed here, may provide a potential solution. Alternatively, we envision that serial injections of differentially barcoded plasmid pools could capture spatially resolved in vivo functional genomics data across temporal alteration trajectories. Although the simultaneous introduction of alterations and establishment of hundreds of co-existing tumor nodules could provide a valuable opportunity for studying interclonal interactions and competition, the complexity of our current model may as such not fully recapitulate the ancestral lineage and clonal evolution seen in human tumors.“

If wanted, we can provide a more comprehensive discussion of the limitations of our study.

Additionally, if the authors can demonstrate that this technique can indeed provide “novel” insights into cancer development, it would further strengthen this manuscript. For example, one of the key discoveries is the decoding of relationships between complex genotypes and tumor-intrinsic as well as microenvironmental phenotypes. To bolster the impact of their findings, the authors could select one or two “novel” examples identified by the CHOCOLAT-G2P system and validate these discoveries using low-throughput/traditional approaches. For instance, they might explore the associations between VEGFA and mutant CTNNB1 with cholangiocyte-associated transcripts, such as Krt19 and Cldn7, to provide a deeper validation of the system's capabilities.

We value the suggestions from Reviewer #3 and have indeed focused further on the epistasis-related finding regarding the intriguing novel observations related to VEGFA and mutant CTNNB1 and cholangiocarcinoma development.

As discussed previously (see Reviewer #1), we have proposed the following hypotheses based on our initial combinatorial screening (Fig.2, Fig.3 and Extended Data Fig. 18):

- A) Omission of VEGFA overexpression from our combinatorial search space of eight alterations should accelerate tumor occurrence and reduce the fraction of cholangiocarcinoma-like nodules.
- B) Omission of mutant Ctnnb1 overexpression should delay tumor occurrence and increase the fraction of cholangiocarcinoma-like nodules.

The flexibility of our mouse model allowed us to test these hypotheses using traditional methods, such as monitoring time to tumor occurrence in comparative cohorts and performing H&E + Krt19/CK19 IHC to estimate the fraction of cholangiocarcinoma-like nodules. Specifically, we conducted hydrodynamic tail vein (HDTV) injections in three mouse cohorts, each with four animals (Point-by-point Figure 1).

For animals that received the mixture of all 8 perturbation plasmids we observed that all mice developed tumors within a time frame of 8 weeks. Histopathological investigation based on H&E staining revealed the presence of multifocal tumor nodules, a fraction of which were

assigned as cholangiocarcinoma and IHC for CK19 confirmed this observation, hence largely corroborating results from our initial experiments.

For animals that received a mixture of plasmids leaving out VEGFA all mice developed tumors within 5 weeks. Histopathological investigation based on H&E staining revealed the presence of multifocal tumor nodules almost all of which were assigned as hepatocellular carcinoma and IHC for CK19 confirmed this observation due to the absence of CK19 positivity in tumor nodules. Note that CK19 positivity can nonetheless be observed in normal bile duct cells hence providing an internal control for antibody-specificity.

For animals that received a mixture of plasmids leaving out mtCTNNB1 all mice developed tumors within 12 weeks. Histopathological investigation based on H&E staining revealed the presence of tumor nodules almost all of which were assigned as cholangiocarcinoma and IHC for CK19 confirmed this observation.

Taken together, these results (Point-by-point Figure 1) validate predictions of C-G2P-based findings related to epistasis through follow-up experiments in larger animal cohorts: mtCTNNB1 epistatically masks and counteracts the occurrence of cholangiocarcinoma phenotypes whereas VEGFA dictates the development of cholangiocarcinoma in the context of genetically heterogenous tumor ecosystems in our mouse model.

The fact that these results are in perfect alignment with the hypotheses derived from C-G2P experiments gives us great confidence in the potential of our approach to robustly map combinatorial perturbations, infer complex epistatic relationships and to thereby enable new discoveries.

We display these findings as an additional Figure 4 in a revised version of the manuscript.

Point-by-point Figure 1; New main Fig 4.: **VEGFA and mutant Ctnnb1 confer epistasis control of cholangiocarcinoma development within heterogeneous tumor ecosystems.**

a, Experimental design. In vivo multiplex perturbation HDTV_i injections were performed using parallel in three different animal cohorts using leave-one-out experimental design.

b, Time to tumor occurrence. Animals were palpated twice weekly to monitor tumor development. N = 4 animals per group.

c, Histological quantification of liver tumor subtypes. H&E images were analyzed and tumor nodules were counted and classified into either hepatocellular carcinoma (HCC) or cholangiocarcinoma (CCA). Median +/- SD, N = 4 animals per group, 2 independent liver tissue sections per animal. Significant relations are indicated as *** pval < 0.001, ** pval < 0.01, * pval < 0.05.

d, Abundance of cholangiocarcinoma. CK19 IHC was used as a cholangiocyte marker. Representative samples are depicted. Scale bar = 5 mm.

There are numerous multi-omic profiles available for liver cancer derived from human specimens. A critical step would be to assess whether the relationships between complex

genotypes and tumor-intrinsic as well as microenvironmental phenotypes, as illustrated in Figure 3, can be reproduced using these “human” datasets. If the “novel” discoveries made through the CHOCOLAT-G2P system in animal models can be corroborated with existing “human” data, it would significantly enhance the credibility and relevance of this approach. Such validation would not only demonstrate that this system can effectively model human cancer but also underscore its potential as a powerful tool for uncovering new insights into cancer biology that are directly translatable to human disease.

We agree with Reviewer #3 that the credibility and relevance of our approach can be strengthened if novel discoveries made using the CHOCOLAT-G2P animal model were corroborated with existing human data.

In our manuscript, we already highlighted several findings supported by observations in human liver cancer. However, as noted in response to concerns raised by Reviewer #2, we acknowledge that the biological relevance of our findings should be interpreted with caution due to the limited number of samples used.

The strongest genotype-phenotype predictions derived from our experiments were not associated with microenvironmental/immune-cell related phenotypes but rather related to the epistasis regulation of cholangiocarcinoma by VEGFA and CTNNB1 for which we were now able to provide validation by additional in vivo experiments as outlined above (Point-by-point Figure 1).

Nonetheless, we would again like to point out several findings that align with existing “human” data that have to our knowledge never been reported in other liver cancer mouse models:

1. Our approach is the first to demonstrate, in an in vivo mouse model, that zonation-associated transcriptional signatures define hepatocellular carcinoma phenotypes despite genetic complexity and can be linked to mutant CTNNB1 expression.

This finding aligns very well with human data (as shown in Désert et al. "Human hepatocellular carcinomas with a periportal phenotype have the lowest potential for early recurrence after curative resection." *Hepatology* 66.5 (2017) and conceptually discussed in Ng, C. K. Y., Piscuoglio, S. & Terracciano, L. M. Molecular classification of hepatocellular carcinoma: The view from metabolic zonation. *Hepatology*. Baltim. Md 66, 1377–1380 (2017)). Analyses based on single-cell RNA sequencing data from human liver cancer lend further support to a zonation-based classification (doi: 10.3389/fimmu.2023.1140201).

In our manuscript, we stated: “Spatial division of metabolic functions is not only central to liver tissue organization under physiological conditions but has been proposed to enable molecular classification of human hepatocellular carcinoma.”

We have now included this information in the revised manuscript:

„Notably, this genotype-phenotype observations align well with aforementioned zonation-based classification of human HCCs, and single-cell RNA sequencing data from human liver cancer, which revealed that the central-like HCC subtype is associated with Ctnnb1 mutations^{33–35}“

To our knowledge, the relevance of zonation-associated phenotype signatures of liver tumors is only poorly understood, likely due to the lack of spatial tissue-level resolved comprehensive phenotyping approaches to interrogate liver cancer mouse models.

In fact, a very recent preprint revealed that liver cancer initiation is dependent on zonation. Using elegant transgenic mouse models and fate mapping, alongside hydrodynamic tail vein injection-based models including pairwise mtCTNNB1 and MYC perturbation, the authors demonstrate the relevance of zone 3 (central phenotype in our study, which we observed to

be the most abundant cancer subtype) and further suggest strategies for cancer prevention that capitalize on knowledge of zonation dependent mechanisms of liver carcinogenesis. (doi: <https://doi.org/10.1101/2024.01.10.575013>).

We have now included this information in our revised manuscript.

“..recent findings from zonation fate-mapping animal models suggest liver cancer prevention strategies that leverage central-zonation-dependent mechanisms, particularly targeting *Gstm3*, which we also identified as a central-like tumor marker (Fig 3b)..”

2. To our knowledge, our approach provides the first in vivo mouse model-based demonstration that Gp2 upregulation is associated with cholangiocarcinoma.

This novel observation in a mouse model corroborates existing human data linking Gp2 expression to cholangiocarcinoma. Indeed, it has been suggested that GP2 “can be hypothesized as a novel marker in large bile duct diseases... anti-GP2 IgA identified a subgroup of patients with severe phenotype and poor survival due to cholangiocarcinoma.” (Jendrek ST, Gotthardt D, Nitzsche T, et al. *Gut* 2017;66:137–144).

In our manuscript, we stated: “Interestingly, CHOCOLAT-G2P pinpointed, among others, expression of ... Gp2 (Pancreatic Glycoprotein 2) as being associated with cholangiocarcinoma (Fig. 3b; Extended Data Fig. 14a). The latter observation harmonizes with earlier research, suggesting that anti-GP2 IgA autoantibodies enable early cholangiocarcinoma detection in subsets of human patients.”

3. Related to the observed tendency towards mutual exclusivity between VEGFA and mtCTNNB1 (Fig. 2e,f) that we further investigated in the context of cholangiocarcinoma development (Fig.3, and 4 and Point-by-point Figure 1) we note that such a tendency can also be observed in human liver cancer samples (Point-by-point Figure 10).

However, inferring mutual exclusivity largely depends on the overall frequency of individual alterations and sample size. Along this line we note that the implementation to infer mutual exclusivity provided by cBioportal does not assign mutual exclusivity between VEGFA amplifications and mutant CTNNB1 to be significant in human liver cancer cohorts, which also holds true for the mutual exclusivity between TP53 and CTNNB1 mutations mentioned by Reviewer #3 (Point-by-point Figure 9). In other words, human liver tumors that harbor VEGFA amplifications alongside mutations in CTNNB1 exist (Point-by-point Figure 10). As such, we did not specifically point out that the tendency for mutual exclusivity between VEGFA and mtCTNNB1 that we observed in our experiments mirrors observations in human liver cancer.

A	B	Neither	A Not B	B Not A	Both	Log2 Odds Ratio
CTNNB1	TP53	173	73	94	26	-0.609
VEGFA	CTNNB1	243	24	93	6	-0.614

Point-by-point Figure 10. Tendency towards mutual exclusivity between VEGFA and mtCTNNB1 in human liver cancer. Oncoprint representation and inference of mutual exclusivity for CTNNB1-VEGFA and CTNNB1-TP53 for comparison. 6 tumor samples present co-occurring alterations in CTNNB1 and VEGFA. Data based on https://www.cbioportal.org/study/summary?id=lihc_tcga.

Statistical analysis should be performed for some of the analyses, for example, those presented in Figures 1f and 3e. Some results lacked statistical significance, such as Figure 2f.

We provide this information in our revised manuscript.

We report p-values for the results shown in Figures 1f and 3e and the new main Fig.4.

Details of the statistical analysis are described in the Methods section, and the code is available at <https://github.com/gerstung-lab/CHOCOLAT-G2P>.

Although the interactions between Myc/NICD and VEGFA/mtCtnnb1 in Figure 2f did not reach the significance threshold, they show trends that prompted further investigation. This is explored in the subsequent section (association of VEGFA with the cholangiocyte phenotype) and confirmed through additional experiments (Fig. 4).

Text in figures (such as Figures 3b, d, and e) is too small.

Thank you for pointing this out. We adjusted text in Fig 3.

The term CHOCOLAT-G2P is too long and could be made shorter.

We agree with the reviewer that the term "CHOCOLAT-G2P" is quite long and can be cumbersome to use frequently.

To address this, we propose using the shortened term "C-G2P" throughout the text. However, we will retain the full name "CHOCOLAT-G2P" at the first mention and in the figure legends to maintain clarity and memorability. This approach would allow us to simplify usage while preserving the distinctiveness of the original term.

Point-by-Point: nBME-24-3483-T

Reviewer #2 (Remarks to the Author):

In this revised version on the m/s, the authors have addressed most of the points I raised during the first review although, from their answers, I get the impression that I was not able to clearly communicate my global impression of the article: It is a novel and cool strategy, with good potential to advance multiplexed in vivo studies. However, I was not convinced of the biological relevance of the findings in cancer research. In their rebuttal, the authors now explain the novelty of their findings, but this was never an issue. In any case, they have addressed some of my concerns and have clarified others and, more importantly, the article is now in a journal more suited for the methodological advance that PERTURB-CAST represents. Overall, I am happy to support its publication in nBME.

Thank you for your positive feedback on our revised manuscript.

We appreciate your concern regarding the biological relevance of our findings, particularly given the limited number of samples analyzed using spatial transcriptomics. While this study primarily focuses on establishing the methodology, we hope that the approaches outlined here will enable future studies to uncover meaningful genotype-phenotype relationships that have the potential to enhance our understanding of tumor heterogeneity and ultimately inform strategies for improving patient care.

Thank you once again for your comments and your support for the publication of our manuscript in nBME.

Reviewer #3 (Remarks to the Author):

The authors have thoughtfully addressed my questions and incorporated additional results and discussion. The manuscript has been substantially enhanced in this revised version.

Thank you for your positive feedback on our revised manuscript.

We are delighted that you found the revisions and additional results to be valuable and that they have enhanced the manuscript. We are particularly grateful for your suggestion to perform additional experiments focusing on cholangiocarcinoma and the role of VEGFA and mtCTNNB1, as it provided valuable direction and has strengthened the study.

Reviewer #4 (Remarks to the Author):

In this manuscript Breinig, Lomakin, Heidari et al describe a new method to link cancer genotype in mice with gene expression state and the tumor microenvironment. Determining the impact of diverse and complex cancer genotypes in tumors that develop entirely within the natural in vivo context is of great importance. The use of triplets of RTL probes for non-expressed gene as barcodes is clever, the triplets end up being critical. Validation of the BC detection is multifold and suggest that this overall approach is accurate. As Review 4, I have tried to consider both the updated manuscript as well as the previous Reviewers' comments (and the authors' very wordy response).

Thank you for your detailed and positive feedback on our revised manuscript.

We appreciate your thorough review of our manuscript and your careful attention to the information and data provided in the supplementary files.

Thoughts:

1. I actually do not find Figure 1 very clear at all. The text is too small and the figures are too dense. This does the paper (and the reader) a disservice. They should consider splitting Figure

1 into at least 2 figures and incorporating aspects of Supplementary Figure 2 and 3. For example, from Figure 1 it's not even clear what genes that are perturbing. Also, there is quite a lot going on in the vectors in Supplementary Figure 3 (protein tags that seem not to have mostly worked and BCs driven by the pol2 and pol3 promoters). This would be more educational if they made small technical conclusions in the Results about all of these components. Were the Pol2 or Pol3 BCs detected better? It seems that the BC worked better than protein tags, so that seems like a win for their approach, otherwise this really could have been down with peptide tags. In my opinion, not need to do it on adjacent section is not a big advantage of their approach.

2. Perhaps there are other things of importance in the vectors that I missed because I didn't have my magnifying glass. My PhD advisor told me that papers can be reviewed by middle-aged people who might need reading glasses, and you don't want to be the one to remind them of that. Also Figure 3, so small, perhaps split into two if there are two main points in there.

We appreciate the reviewer's suggestion and adjusted the figures.

To reduce complexity and present individual figures with a clearer and less dense layout, we have updated the layout of the manuscript's four main figures to a seven-figure layout in the revised version.

Figure 1: This figure has been split into three revised figures and incorporates aspects of Extended Data Figures 2 and 3.

- New Fig. 1: Now focuses on the concept of C-G2P and PERTURB-CAST, without delving into the specific genes that were perturbed.
- New Fig. 2: This figure now emphasizes the combinatorial complexity of human liver cancer. It illustrates the genes perturbed in this context and highlights the novel barcoding strategy implemented in the plasmids we designed.
- New Fig. 3: This figure now concentrates on spatial barcode mapping using PERTURB-CAST for C-G2P samples.

Figure 3: This figure has been split into two revised figures:

- New Fig. 5: This figure highlights spatial phenotype signatures and provides clearer visualization of these signatures in the revised manuscript.
- New Fig. 6: This figure focuses solely on genotype-phenotype relationships, particularly those related to cholangiocytic features and the results for VEGFA and mtCTNNB1.

Finally, Figure 4 has been further modified:

- New Fig. 7: This figure now includes information from Extended Data Figure 18, illustrating the presence of VEGFA and the absence of mtCTNNB1 in cholangiocarcinoma. This provides background for the additional in vivo experiments shown.

The font size has been adjusted throughout, and where applicable, we now use a minimal font size of Arial 8 for all main figures.

Related to the previous illustration of Vectors, the decision to depict various functional elements in the vectors (ED Fig.3) stemmed from my enthusiasm to integrate multiple features that could facilitate future experiments. These elements included T7 promoter sequences for zombie assays (now incorporated into PerturbView), long barcodes for multiplexFISH with at least 10 hybridization sites, protein tags for alignment with PerturbMap readouts, U6-driven

barcodes for potential guideRNA capture, and 10X capture sequences to enable 10X scRNA-seq applications, among others.

We recognize that illustrating excessive details detract from the key findings presented in the manuscript. To enhance clarity, we adjusted and simplified vector display in the revised Fig.2 and Extended Data Fig.3 accordingly to only focus on aspects presented in this study.

We have now further included information related to barcode detection and the use of pol3 promoters in the results section and also adjusted ED Fig.6c to indicate pol3-driven barcodes more explicitly:

“Notably, redeployed barcodes expressed using a Pol III promoter (hU6; Extended Data Fig. 3) were detectable, but we noticed a trend where detection became weaker as the barcode was positioned farther from the 5' end (Extended Data Fig. 6c)“.

3. While the totally random combination of genetic alterations is presented as a strength in their current study this will not always be the case. Even in their study while there are theoretically 256 combinations, it seems that most initial cells get multiple integrations thus the number of cells that gets only 1 or 2 genetic alterations is likely very low. If one wanted to study many defined three gene combinations they would need a different approach to create those manipulations. Presumably the Authors can imagine ways to do that which could be added to the Discussion.

We thank the reviewer for this comment.

We agree that the random combination we generate with our system represents a rather bold approach to address the combinatorial explosion problem associated with the multitude of alterations that can be detected in tumors. Indeed, more systematic approaches to probe combinatorial combinations could be advisable to better disentangle genetic interactions.

We have now included a statement in the Discussion that describes alternative strategies to achieve this goal:

„Further, to more systematically investigate genetic interactions we speculate that the multiplexing capabilities offered by MultiMir combinatorial RNAi as well as CRISPR/Cas12a warrant further investigation.“

4. I acknowledge their logic of using shRNAs over Cas9/sgRNAs. However most loss-of-function somatic cancer modeling is with Cas9 so it seems bizarre to not have that as a future direction or mention at all in the Discussion.

We agree. We have now included CRISPR approaches as a future direction in the Discussion.

„Further, to more systematically investigate genetic interactions we speculate that the multiplexing capabilities offered by MultiMir combinatorial RNAi as well as CRISPR/Cas12a warrant further investigation. For multiplex CRISPR perturbations, it may be necessary to employ modifications that avoid double-stranded breaks, such as CRISPR interference (CRISPRi), to minimize the risk of unwanted chromosomal rearrangements and other detrimental effects.“

5. In retrospect it seems that they should have include other “inert’ vectors. Their shRen show up in almost half the tumors and they don’t have an inert cDNA, but it also would have shown up in a lot of tumors. So there are lots of vectors that just get carried along in the tumors. Not sure if the Authors would want to add some discussion of how to do this better in the future (perhaps they disagree with me, which is fine)

We thank the reviewer for this comment.

Indeed, that the fraction of 'by-stander' events is quite substantial and we considered using the presence of shRen as an estimate for this effect. As we cannot fully rule out that shRen expression can indeed serve as a reliable "inert" control, we felt that a more rigorous assessment of this by-stander measure would be needed to justify its use as an underlying ground-truth for modeling importance of perturbations that drive tumorigenesis by deviation from shRen occurrence, similar to approaches used in large-scale functional genomics screens that rely on the use of hundreds of inert control perturbations. We therefore 'simply' stated marginal frequencies of individual perturbations (Fig. 2d → revised Fig.4d) illustrating that e.g. MYC or CTNNB1 are more abundant than shRen.

Given that our phenotypic readout extends beyond tumor occurrence, we recognize that the presence of a perturbation—even if it is merely carried along without contributing to tumor development—can still result in a detectable (and potentially disease relevant) phenotypic effect captured by spatial transcriptomics.

6. I am not sure that they get across as clearly as they should that many others have generated tumors of defined genotypes or random combination of genotypes (Sidi Chen, Shramek, Roland Rab, Winslow Labs and others) but their approach is way better because of the transcription read out of the cancer cells and the TME. Roland Rad did something extremely similar in pancreas many many years ago and essentially had the same problem in that many tumors had most if not all of the genetic alterations.

We thank the reviewer for the kind feedback.

We have cited the majority of previous approaches conducted by e.g. the Chen, Rad, and Winslow labs with the aim to effectively highlight both the similarities and differences in our work.

We currently feel that it remains to be seen if our PERTURB-CAST „...is way better because of the transcription read out of the cancer cells and the TME“.

While we are very enthusiastic about the potential of PERTURB-CAST to offer more comprehensive insights into genotype-phenotype relationships, we recognize the need for tempered optimism. Currently, traditional approaches may offer certain benefits in terms of throughput, despite their limitations.

7. Pro-codes and other methods that tag cell lines prior to injection are not at all like this approach. Conflating genetically engineered cancer models in which tumors grow from single cells within the correct environment with cell line transplants will ultimately be acknowledge as a major contributor to confusion in cancer biology.

While we recognize that our mouse model clearly differs from approaches that involve tagging cell lines prior to injection, we consider PERTURB-CAST to be complementary to Pro-Codes and other emerging molecular barcoding techniques for spatially mapping perturbations.

From this perspective, we believe that highlighting the distinctions and parallels between PERTURB-CAST and, for example, Pro-Codes is educational.

8. They only look at the number of tumors with each perturbation combination. Are there combination that lead to larger tumors?

Thank you for your thoughtful comment.

In our current study, we initially focused on quantifying the number of tumors associated with each perturbation combination as an initial measure of perturbation effects. We recognize the importance of evaluating not only the incidence but also the size of tumors arising from different perturbation combinations.

However, we note that when assessing tumor size from a single plane of sectioning, the observed size may be misleading. A spherical or irregularly shaped tumor may appear to have varying nodule sizes depending on the plane in which it is cut. Larger nodules may appear smaller if they are sectioned near the edge, while smaller nodules may appear disproportionately large if the section is made through their central portion (Point-by-point Figure 1).

Point-by-point Figure 1: Limitations of single z-plane section analyses of tumor size. Schematic illustrating how spherical tumors may appear in varying sizes depending on the plane of sectioning.

Using our current dataset, we did not find relevant associations between certain combinations of perturbations and tumor size (data not shown). This emphasizes the need for e.g. 3D imaging to accurately assess volumetric tumor size and heterogeneity. We are actively working on implementing strategies to more thoroughly investigate this aspect of our mouse model with the aim to refine our readout.

9. The new leave-one-out data is nice and solidifies their ~new biology.

We thank the reviewer for the kind feedback.

Minor points.

1. Fig. 2b, I am not sure why the top like is a rainbow. I would seem to be more useful if it indicated singles, doubles, triples, etc

Given the 256 possible combinations tested, we chose to use a continuous color palette (rainbow) on top of Fig. 2b to spatially represent the genotypes shown in Fig. 2a. This approach allows us to visually link the spatial distribution of genotypes with the specific combinations of perturbations, whose occurrences were depicted in Fig. 2b.

We believe this color palette most effectively illustrates the complexity of the dataset and therefore kept it in the revised Fig. 4a.

2. Fig 4b, seem backwards from how most people would plot this.

We thank the reviewer for highlighting this point.

Indeed, it is common practice to plot this data differently and label the y-axis as "survival" in similar plots. However, I am not particularly inclined toward this visualization approach, as our data represent tumor occurrence, and animals are actively sacrificed before succumbing to tumor burden.

To align with common visualization strategies for similar experiments, we can adjust the plot presented in revised Fig.7c if explicitly wanted.

3. Fig S1C. They should show the oncoprint for the genomic alterations that their NICD models (would that be all amplification in Notch genes?)

We thank the reviewer for highlighting this point. We've updated the oncoprint (see below) and have now included it in the revised main Fig.2.

Related to NICD models we therefore focused on Notch1 in human HCC, since our engineered construct is derived from the NOTCH1 intracellular domain. In contrast to the Oncoprint depicted in the former ED Fig.1c we have now included data based on amplifications, gains, deep deletions, increased mRNA expression and putative driving mutations. As such the frequencies for alterations differ from the data depicted in the former ED Fig.1c.

For now, we can only speculate that this model also captures aspects of amplifications of other components of the Notch signaling pathway for which overexpression in human HCCs was shown. Two recent papers from the Lujambio lab specifically address the role of Notch1 in liver cancer stating that "...NOTCH1 is frequently upregulated in HCC patients (overexpressed in around one-third of HCCs) and its expression significantly correlates with genes in the NOTCH signaling pathway, indicating activation..." (<https://doi.org/10.1158/2159-8290.CD-24-1215>; Point-by-point Figure 2.)

Point-by-point Figure 2 from Lindblad et al., 2024: Expression of NOTCH receptors and NOTCH pathway genes in the TCGA LIHC cohort. Each column represents one patient. The tumors with the corresponding gene upregulated (defined as 0.2 SD above the mean) are highlighted in red. In the right columns, the co-occurrence of each NOTCH receptor with the rest of the genes is calculated by Fisher Exact Test. P values corrected by Benjamini-Hochberg at cBioPortal.

4. Sup Figures, fonts are pretty small.

We thank the reviewer for pointing this out.

Where feasible, we have increased the font size to improve readability.

However, due to the complexity of the data, such as in Extended Data Fig. 16 in which we plot expression of marker genes for all tumor nodules detected in a total of 11 Visium samples, it was occasionally not possible to increase the font size without compromising overall presentation of the data in one figure.

5. From the Reviewers comments, while it is unarguable that human tumors acquire mutations sequentially, it is arguable whether that is actually important. There are very few examples to suggest that mutations order is actually important (some modeling of genomic alterations sequentially has been done in GEMM (using Cre/Flp system)) but this hasn't been done in high throughput (as the Author note). The don't agree that the inability to do this sequentially is a major problem for this manuscript.

We thank the reviewer for this feedback.

Raising awareness for this potential limitation of our approach seems justified and we hope that we have sufficiently addressed this topic in our discussion.

Personally, I believe that it is challenging to determine whether the phenotypic effects of sequential acquisition of alterations differ from those of simultaneous acquisition, given the current lack of functional data addressing this question. Nonetheless, I hold the view that gaining a deeper understanding of this distinction could be important and I hope that future research strategies will effectively address this gap.

6. I also don't entirely agree with the review that call this "standard state-of-the art (barcoded genetics...)" I am not sure of many other papers of "barcoded genetics" with ST readouts etc. Yes this isn't all new stuff but it innovative and new.

We thank the reviewer for this feedback.